# Integration of hunger and hormonal state gates infant-directed aggression

Mingran Cao[1], Rachida Ammari[1], Maxwell X. Chen[1], Patty Wai[1], Bradley B. Jamieson[1], Swang Liang[1], Basma F. A. Husain[1], Aashna Sahni[1], Nathalie Legrave[2,4], Irene Salgarella[1], James MacRae[2], Molly Strom[3] & Johannes Kohl[1]✉

Social behaviour is substantially shaped by internal physiological states. Although progress has been made in understanding how individual states such as hunger, stress or arousal modulate behaviour[1–9], animals experience multiple states at any given time[10]. The neural mechanisms that integrate such orthogonal states—and how this integration affects behaviour—remain poorly understood. Here we report how hunger and oestrous state converge on neurons in the medial preoptic area (MPOA) to shape infant-directed behaviour. We find that hunger promotes pup-directed aggression in normally non-aggressive virgin female mice. This behavioural switch occurs through the inhibition of MPOA neurons, driven by the release of neuropeptide Y from Agouti-related peptide-expressing neurons in the arcuate nucleus (Arc^AgRP neurons). The propensity for hunger-induced aggression is set by reproductive state, with MPOA neurons detecting changes in the progesterone to oestradiol ratio across the oestrous cycle. Hunger and oestrous state converge on hyperpolarization-activated cyclic nucleotide-gated (HCN) channels, which sets the baseline activity and excitability of MPOA neurons. Using microendoscopy imaging, we confirm these findings in vivo, revealing that MPOA neurons encode a state for pup-directed aggression. This work provides a mechanistic understanding of how multiple physiological states are integrated to flexibly control social behaviour.

When encountering conspecifics, animals must decide on how to behave. Such social decisions are typically seen as the result of accumulating external sensory information about the target (for example, sex, age or status). However, internal states—such as hunger, stress or arousal—substantially affect social behaviour. Although the effects of individual physiological states on behaviour are increasingly well understood, organisms must integrate multiple states at any given time to make behavioural decisions. However, how this state integration occurs in the brain remains largely unknown. We address this question using a simple paradigm in which female mice are presented with pups and exhibit pup-directed care or aggression. We first establish how two state variables, hunger and oestrous state, affect pup interactions. Then we uncover the cellular and neural mechanisms by which these orthogonal states are integrated to shape social behaviour.

## Arc^AgRP→MPOA pathway drives pup attack

Virgin female laboratory mice typically either ignore pups or exhibit spontaneous parental behaviour. Food deprivation induced a shift towards pup-directed aggression in these animals (Fig. 1a and Extended Data Fig. 1n–s). The percentage of aggressive mice (Agg^+) increased, and attack latency decreased, with food deprivation duration, which plateaued after 3 h (Fig. 1b and Extended Data Fig. 1a). Restoring food access increased feeding and reduced pup-directed aggression

(Fig. 1b,c). Notably, food deprivation triggered aggression regardless of whether mice had previously shown parental behaviour or ignored pups, with similar attack latencies in both groups (Extended Data Fig. 1b,c). This aggression was specifically directed at pups, as the proportion of mice that attacked prey or adult intruders of either sex was unaffected by food deprivation (Extended Data Fig. 1f). Moreover, this behavioural shift was not stress-related. Food-deprived mice did not show changes in performance in elevated-plus maze and open-field tests (Extended Data Fig. 1h,i) and did not respond similarly to other stressors (Extended Data Fig. 1j). Hunger therefore triggers a switch to infant-directed aggression in virgin female mice.

We next investigated the neural mechanisms that underlie this switch. Arc^AgRP neurons have a central role in the regulation of hunger-driven behaviours[11,12]. We therefore tested whether they mediate the effects of food deprivation on pup-directed behaviour. Chemogenetic activation of Arc^AgRP neurons increased food consumption, as previously reported[12,13] (Fig. 1d,e). Notably, this manipulation also induced pup-directed aggression in sated mice, whereas no effects were observed in animals injected with a control virus (Fig. 1d–f). Activation of Arc^AgRP neurons is therefore sufficient to induce pup-directed aggression in female mice. Conversely, when Arc^AgRP neurons were chemogenetically inhibited through an ivermectin-responsive human glycine receptor (hGlyAG)[14], food-deprivation-induced pup aggression was strongly reduced (Fig. 1g–i).

[1]State-dependent Neural Processing Laboratory, The Francis Crick Institute, London, UK. [2]Metabolomics Science Technology Platform, The Francis Crick Institute, London, UK. [3]Vector Core, The Francis Crick Institute, London, UK. [4]Present address: Metabolomics Platform, Luxembourg Institute of Health, Strassen, Luxembourg. ✉e-mail: jonny.kohl@crick.ac.uk

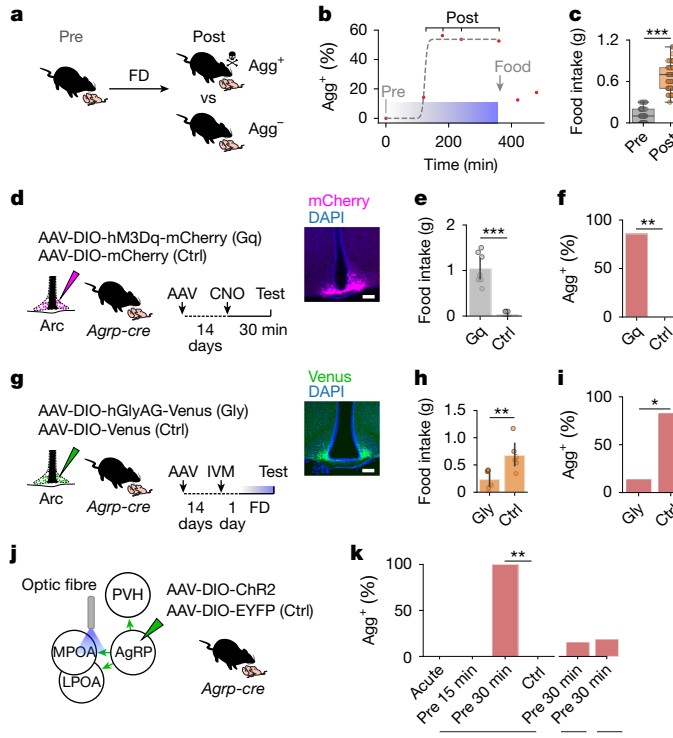

**Fig. 1 | Arc^AgRP→MPOA projections mediate hunger-induced, pup-directed aggression. a**, Schematic of the switch to pup-directed aggression induced by food deprivation (FD). **b**, Percentage of Agg⁺ mice as a function of FD duration (refed for 1 or 2 h; each point represents a cohort of $n$ = 9–10 mice). The blue bar indicates the deprivation period. Logistic regression fitted to aggression outcomes, $R^2$ = 0.365, $P$ = 1.11 × 10⁻¹⁶. **c**, Effects of 1 h of food intake before and after 6 h of FD ($n$ = 40). **d**, Left, schematic of chemogenetic activation of Arc^AgRP neurons (Gq) compared with controls (Ctrl). Right, example brain section of mCherry fluorescence in Arc^AgRP neurons. Scale bar, 100 μm. **e**, Effect of Arc^AgRP neuron activation on 1 h of food consumption in sated mice compared with controls ($n$ = 7 per group). **f**, Percentage of sated mice injected with clozapine-*N*-oxide (CNO) showing aggression compared with controls ($n$ = 7 per group). **g**, Left, schematic of chemogenetic inhibition of Arc^AgRP neurons with ivermectin (IVM)-sensitive hGlyAG versus controls after 6 h of FD. Right, example brain section. Scale bar, 100 μm. **h**, Effect of Arc^AgRP neuron inhibition on 1 h of food consumption in food-deprived mice compared with controls ($n$ = 6, 6). **i**, Percentage of food-deprived mice injected with IVM showing aggression ($n$ = 7) and controls ($n$ = 6). **j**, Schematic of optogenetic activation of Arc^AgRP projections. **k**, Percentage of Agg⁺ mice after activation of Arc^AgRP projections. Stimulation during (acute) or for 15 or 30 min before (pre) pup interactions ($n$ = 5 (MPOA), 6 (LPOA), 5 (PVH) and 6 (Ctrl)). Controls received 30 min of pre-stimulation. Statistics: paired *t*-test (**c**); *U*-test (**e**,**h**); or Fisher's exact test (**f**,**i**,**k** (Benjamini–Hochberg adjustment in **k**)). All tests were two-sided. Data are the mean ± s.e.m. Box plots show the median (line) and interquartile range (IQR; box), and whiskers are 1.5× the IQR. *$P$ < 0.05, **$P$ < 0.01, ***$P$ < 0.001. See Supplementary Table 3 for further details of statistical analyses.

To address whether hungry mice attack pups because they perceive them as food, we recorded bulk Arc^AgRP activity in food-deprived mice that showed aggression to pups using fibre photometry (Extended Data Fig. 1k). Arc^AgRP activity has been shown to increase during food deprivation and to rapidly decrease in response to food-related cues[15,16]. However, we observed that it increased in mice during pup investigation (Extended Data Fig. 1l,m), similar to previously reported responses to adult conspecifics[17].

Arc^AgRP neurons might exert these effects by directly targeting circuits that mediate pup-directed behaviour. We used the immediate-early gene *Fos* to assess neuronal recruitment in aggressive (Agg⁺) and non-aggressive (Agg⁻) mice, focusing on brain regions crucial for pup-directed behaviours, including the hypothalamus, the septal and amygdaloid nuclei[18] (Methods and Extended Data Fig. 2a,b). Of the 53 assessed areas, 5 showed significantly lower FOS⁺ cell densities in Agg⁺ mice than Agg⁻ mice (Extended Data Fig. 2c), which suggests that Arc^AgRP neurons may drive pup-directed aggression by inhibiting parenting circuits. The absence of FOS differences in areas implicated in female infanticide (BNST, PA, PeFA and MeA)[19,20] may result from the masking of bidirectional activity changes in neuronal subsets by population averaging. Alternatively, pup-directed aggression—driven by inhibitory Arc^AgRP neurons—may rely on disinhibition, with key excitatory neurons located elsewhere, a result consistent with the reduced FOS⁺ densities observed in Agg⁺ mice (Extended Data Fig. 2c).

Arc^AgRP neurons send largely non-collateralized projections to more than a dozen targets[7,21–23], including to two of these five candidate areas: the MPOA and the lateral preoptic area (LPOA) (Extended Data Fig. 2d). To address whether these candidate projections mediate pup-directed aggression, we induced viral-mediated expression of channelrhodopsin-2 (ChR2) in Arc^AgRP neurons and implanted optical fibres above their projection targets (Fig. 1j and Extended Data Fig. 2e). Acute optogenetic stimulation of Arc^AgR→MPOA projections during pup interactions, or 15 min of stimulation before behavioural testing, did not affect pup-directed behaviour (Fig. 1k). However, stimulating MPOA projections for 30 min before pup interactions switched all sated mice to pup-directed aggression (Fig. 1k). Prolonged stimulation of MPOA projections for 1 h (see ref. 11) also increased food intake. However, this increase was correlated with longer attack latencies (Extended Data Fig. 2f–j), which indicates that Arc^AgRP→MPOA projections influence feeding and pup-directed aggression through dissociable mechanisms. By contrast, optogenetic stimulation of nearby LPOA projections did not affect pup-directed behaviour or food intake (Fig. 1k and Extended Data Fig. 2k). We also confirmed that activation of projections to the paraventricular nucleus of the hypothalamus (PVH) increased food intake, as previously shown[24], without affecting social behaviour (Fig. 1k and Extended Data Fig. 2l). Optogenetic and chemogenetic Arc^AgRP manipulations resulted in pup-attack latencies comparable to those observed after food deprivation (Extended Data Fig. 2m). This result suggests that engaging this circuit is sufficient to replicate the behavioural switch induced by metabolic state changes. These findings establish that Arc^AgRP→MPOA projections mediate hunger-induced pup-directed aggression.

## Oestrous state sets switching rate

We next asked why hunger induces pup-directed aggression in only around 60% of females (Fig. 1b). Agg⁺ mice were not hungrier than Agg⁻ mice: food consumption and plasma levels of the hunger hormone ghrelin was not significantly different between the two groups (Fig. 2a–c). We therefore proposed that Agg⁺ females are in a reproductive state permissive to aggression. In female rodents, the oestrous cycle lasts 4–5 days and is linked with substantial behavioural and neurophysiological changes[25,26] (Fig. 2d). The percentage of mice switching to pup-directed aggression (switching rate) fluctuated across oestrous cycle stages, being highest in metestrus (70%) and lowest in oestrus (32%; Fig. 2e). Oestradiol (E2) and progesterone (P4) are the main effectors of the oestrous cycle[27] (Fig. 2d), but the switching rate was not correlated with individual levels of E2 or P4 (Extended Data Fig. 3a,b). Instead, it tracked the P4/E2 ratio, which suggests that relative levels of both hormones are integrated in feeding and/or parenting circuits (Fig. 2f). In support of this hypothesis, the switching rates of female mice in mid-pregnancy or late pregnancy—which have higher P4 and E2 levels than virgins, but comparable P4/E2 ratios (Fig. 2g and Supplementary Table 2)—closely matched our predictions (Fig. 2h). By contrast, the oestrous state did not affect baseline pup-directed behaviour or attack latency (Extended Data Fig. 3c,d).

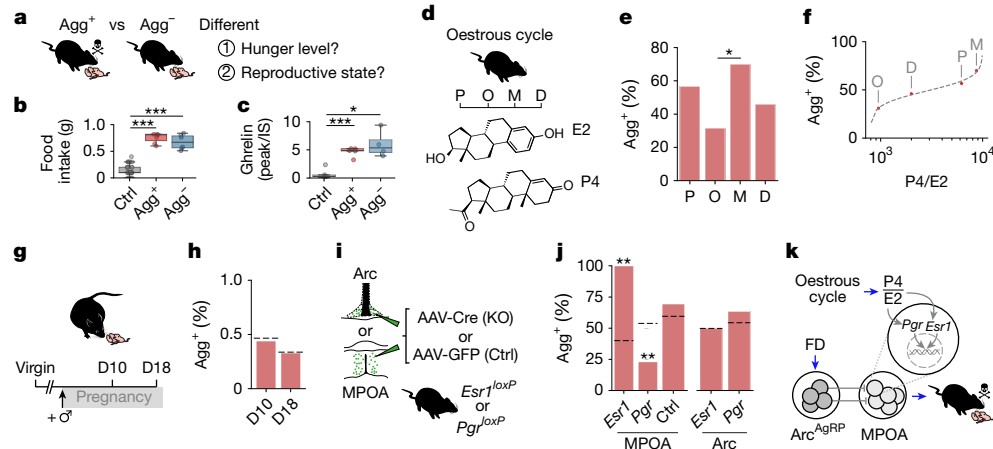

**Fig. 2 | Oestrous state sets behavioural switching probability in the MPOA.** **a**, Potential factors that contribute to different pup-directed behaviours after FD. **b**, Effects of 1 h of food intake at baseline (Ctrl) and after 6 h of FD ($n$ (left to right) = 32, 7 and 6 mice). **c**, Relative plasma ghrelin levels (ratio of peak area to internal standard (IS)) at baseline (Ctrl) and after 6 h of FD ($n$ (left to right) = 5, 5 and 4 mice). **d**, E2 and P4 levels vary across the oestrous cycle. P, prooestrus; O, oestrus; M, metestrus; D, diestrus. **e**, Percentage of Agg+ mice across oestrous stages ($n$ (left to right) = 30, 19, 40 and 37 mice). **f**, Switching rate as a function of the P4/E2 ratio (logit function fitted to binary aggression data. $R^2$ = 0.071, $P$ = 0.03 ($n$ (left to right) = 30, 19, 40 and 37 mice; Supplementary Table 2). **g**, Schematic of testing at mid-pregnancy (day 10 (D10)) and late (D18) pregnancy. **h**, Switching rates in pregnant females ($n$ = 9 (D10) and 12 (D18) mice). Dashed lines indicate predicted values (Methods). **i**, Schematic of targeted *Esr1* or *Pgr*

ablation in the MPOA or the Arc, and controls. **j**, Percentage of Agg+ mice after *Esr1* or *Pgr*, KO ($n$ (left to right) = 10, 13, 8, 10 and 11 mice). Dashed lines indicate predicted baseline switching rates of each cohort with intact receptors based on the measured distribution of oestrous stages (Methods). **k**, Model of how integration of oestrous and hunger state in the MPOA sets the switching rate. Statistics: one-way analysis of variance (ANOVA) with Tukey post hoc test (**b**,**c**); Fisher's exact test (two-sided, Benjamini–Hochberg adjustment) (**e**); binomial tests (**h**); or Poisson binomial distribution for expected aggression rates based on oestrous state (**j**). Observed rates outside the 99% confidence interval (CI) are indicated. Box plots show the median (line) and the interquartile range (box), and whiskers are 1.5× the IQR. *$P$ < 0.05, **$P$ < 0.01, ***$P$ < 0.001. See Supplementary Table 3 for further details of statistical analyses. The schematic in **k** was created using BioRender (https://www.biorender.com).

We next tested this model and assessed where hormonal state is sensed in this context. E2 and P4 can influence neuronal function through membrane-bound receptors and through their intracellular receptors ESR1 and PR, which act as transcription factors[28] and are highly enriched in the MPOA[3]. Mice with floxed *Esr1* or *Pgr* alleles were injected with an adeno-associated virus (AAV) expressing Cre recombinase into the arcuate nucleus (Arc) or MPOA (Fig. 2i). This resulted in local receptor knockout (KO), whereas injection of a control AAV did not affect receptor expression[3]. KO of *Esr1* or *Pgr* in the Arc did not alter pup-directed behaviour, but receptor ablation in the MPOA significantly affected the switching rate. Notably, 100% of *Esr1*-ablated mice became aggressive after food deprivation compared with a predicted 40% baseline rate for a group of mice with intact receptors based on the measured oestrous stage distribution (Fig. 2j and Methods). This effect probably occurs because E2 insensitivity increases the relative P4/E2 ratio sensed by MPOA neurons. By contrast, only 23% of *Pgr*-ablated mice became aggressive (53% predicted), a result in accordance with a low P4/E2 ratio acting on MPOA neurons (Fig. 2j). Levels of parental behaviour were positively correlated with *Pgr* ablation efficiency (Extended Data Fig. 3g), and injection of a control AAV did not affect switching rate (Fig. 2j). Neither *Esr1* nor *Pgr* KO resulted in spontaneous pup-directed aggression before food deprivation, which suggests that hormonal modulation alone is insufficient to trigger aggression in the absence of hunger. Notably, food intake was not affected by oestrous state, a finding that supports the conclusion that ovarian hormones modulate parenting rather than feeding centres in this context (Extended Data Fig. 3e,f). Integration of hunger and oestrous state in the MPOA therefore controls a switch in pup-directed behaviour (Fig. 2k).

## State integration in MPOA neurons

To address how MPOA neurons perform this integration, we performed patch-clamp recordings in brain slices from female mice before (Pre)

and after (Post) food deprivation (Fig. 3a). MPOA neurons from Agg+ mice exhibited reduced spontaneous firing, a twofold increase in the proportion of silent neurons and a strong reduction in intrinsic excitability (Fig. 3b–d and Extended Data Fig. 4a–c). Other biophysical parameters were unchanged (Extended Data Fig. 4f–r). These changes also occurred in galanin (Gal)-expressing MPOA neurons, which have a well-defined role in parental behaviour[19,29] (Extended Data Fig. 4u–z). The reduced spontaneous activity and excitability of MPOA neurons in Agg+ mice were not due to overt changes in the resting membrane potential or synaptic inputs (Extended Data Fig. 4f, m–p). However, membrane input resistance was increased in Agg+ mice, which hints at a closure or downregulation of ion channels (Extended Data Fig. 4d). Indeed, negative current injection revealed a depolarizing voltage sag in Pre and Agg− mice, which was strongly reduced in Agg+ mice (Fig. 3e,f). The sag amplitude was inversely correlated with input resistance (Extended Data Fig. 4e) and positively correlated with neuronal excitability (Extended Data Fig. 4s,t). This sag was mediated by HCN channels and was abolished by the HCN blocker ZD-7288 (ref. 30) (Fig. 3f). HCN blockade in brain slices from sated mice also silenced MPOA neurons (Fig. 3c and Extended Data Fig. 5a) and decreased their excitability (Extended Data Fig. 5d). Modulation of HCN channel function therefore reproduces the Agg+ neuronal phenotype. To test whether food deprivation alters MPOA neuron properties independently of behavioural outcome or oestrous cycle stage, we performed two comparisons: (1) a Pre group matched to the weighted Post group (60% Agg+, 40% Agg−) for overall oestrous cycle composition (that is, proportional representation of each cycle stage); and (2) Pre and Post groups in the same oestrous stage (diestrus). In both cases, food deprivation was associated with a reduced voltage sag amplitude, an increased input resistance and a trend towards lower neuronal excitability (Extended Data Fig. 5j–w). These findings indicate that food deprivation alone affects MPOA neuronal physiology.

In addition to the neuropeptide AgRP itself, Arc^AgRP neurons release GABA and neuropeptide Y (NPY), which mediate feeding in a partially

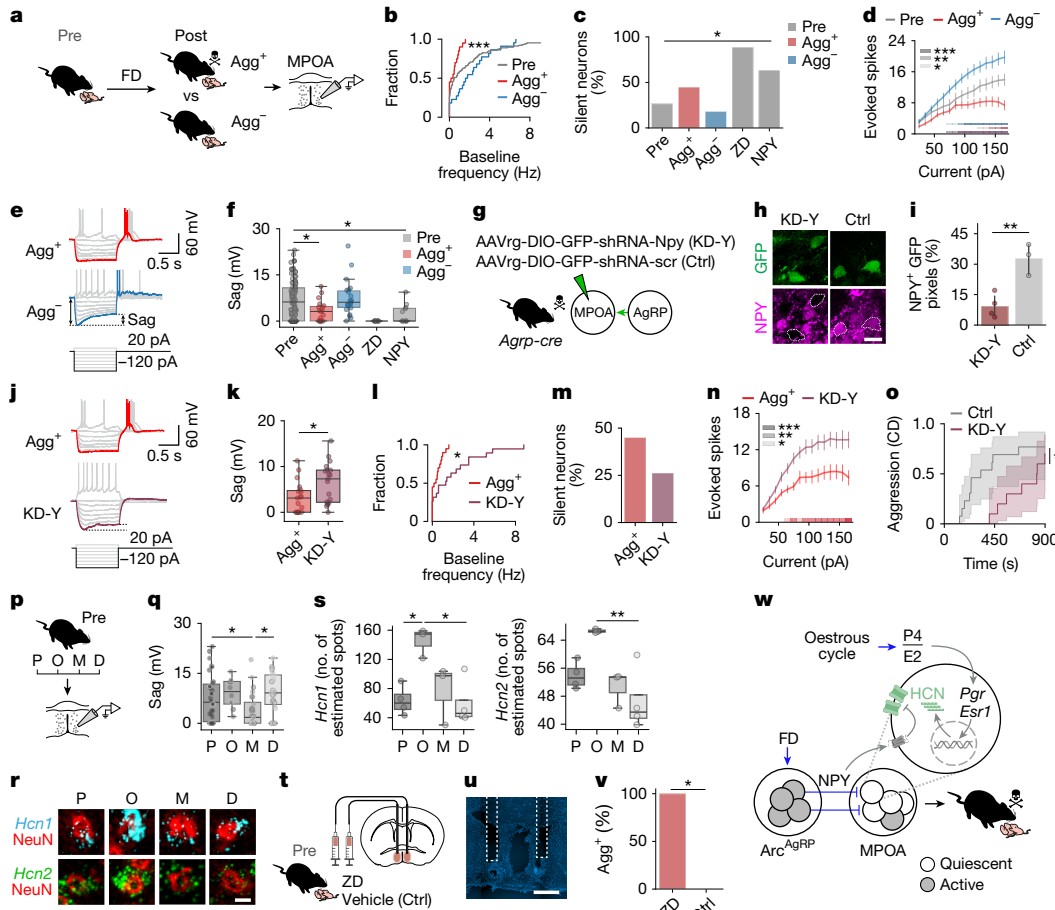

**Fig. 3 | Integration of hunger and oestrous state in MPOA neurons.**
**a**, Schematic of whole-cell recordings. **b**, Baseline firing for the indicated mice (n = 126 (Pre), 20 (Agg⁺) and 22 (Agg⁻) cells, N = 24 (Pre), 6 (Agg⁺) and 3 (Agg⁻) mice). **c**, Silent neurons at resting potential for the indicated mice (n (left to right) = 105, 20, 22, 9 and 11 cells, N (left to right) = 17, 6, 3, 1 and 4 mice). ZD, ZD-7288. **d**, Evoked spikes for the indicated mice (n = 23 (Pre), 22 (Agg⁺), 21 (Agg⁻) cells, N = 7 (Pre), 5 (Agg⁺) and 3 (Agg⁻) mice). **e,f**, Voltage sag (**e**) and amplitude (**f**) for the indicated mice (n (left to right) = 103, 19, 22, 9 and 11 cells, N (left to right) = 22, 6, 3, 1 and 4 mice). **g**, Schematic of *Npy* knockdown (KD-Y) in Arc^AgRP→MPOA projections, and the scrambled control. **h**, Example images (dashed outlines indicate GFP⁺ cells). **i**, Knockdown efficiency (N = 4 (KD-Y) and 3 (Ctrl) mice). **j,k**, Sag (**j**) and amplitude (**k**) for the indicated mice (n = 19 (Agg⁺) and 19 (KD-Y) cells, N = 6 (Agg⁺) and 3 (KD-Y) mice). **l,m**, Baseline firing (**l**) and silent neurons (**m**) for the indicated mice (n = 20 (Agg⁺) and 19 (KD-Y) cells, N = 6 (Agg⁺) and 3 (KD-Y) mice). **n**, Evoked spikes for the indicated mice (n = 22 (Agg⁺) and 17 (KD-Y) cells, N = 5 (Agg⁺) and 3 (KD-Y) mice). **o**, Cumulative incidence of aggression for the indicated mice (n = 8 (Ctrl) and 13 (KD-Y)). Shading, 95% CI. **p**, Schematic of MPOA recordings across the oestrous cycle. **q**, Sag amplitude for the indicated mice (n (left to right) = 30, 8, 28 and 27 neurons, N (left to right) = 9, 3, 8 and 4 mice). **r,s**, Images of *Hcn* expression in the MPOA (**r**) and estimated spots (**s**) (n (left to right) = 4, 3, 3 and 4). **t**, Schematic of ZD infusion into the MPOA. **u**, Image of cannula placement. **v**, Percentage of Agg⁺ mice (n = 4 (ZD) and 5 (Ctrl)). **w**, State integration model. Statistics: *U*-test (**b,i,k,l** (**b**, Pre vs Agg⁺)); Fisher's exact test (**c,m,v** (two-sided, Benjamini–Hochberg in **c**); mixed linear model (**d,n** (**d**, significant periods: blue, Pre/Agg⁻; red, Pre/Agg⁺; purple, Agg⁺/Agg⁻)); *U*-test with Benjamini–Hochberg (Pre, Agg⁺, Agg⁻; Pre, ZD, NPY) (**f**); log-rank test (one-sided) (**o**); or one-way ANOVA with Tukey post hoc (**q,s**). Data are the mean ± s.e.m. Box plots show the median (line) and the interquartile range (box), and whiskers are 1.5× the IQR. *P < 0.05, **P < 0.01, ***P < 0.001. See Supplementary Table 3 for details of statistical analyses. Scale bars, 1 mm (**t**), 20 µm (**h**) or 10 µm (**r**). The schematic in **w** was created using BioRender (https://www.biorender.com).

redundant manner[13,31,32]. We therefore asked which of these neuromediators control the effect of food deprivation on pup-directed behaviour. Food deprivation affected pup interactions within around 2 h (Fig. 1b), and optogenetic activation of Arc^AgR→MPOA projections for 30 min resulted in pup-directed aggression (Fig. 1k). By contrast, AgRP mediates a delayed, chronic feeding response[13,33] and its application to brain slices from sated mice did not reproduce the MPOA neuronal silencing observed in Agg⁺ animals (Extended Data Fig. 6a–f). GABA and NPY modulate feeding more rapidly[13,33], but we did not find evidence of extensive direct GABAergic Arc^AgRP→MPOA connectivity (Extended Data Fig. 6m–p and Supplementary Note). Moreover, food deprivation did not significantly hyperpolarize MPOA neurons, as expected from increased GABAergic transmission (Extended Data Fig. 4f). We therefore reasoned that NPY release from Arc^AgRP→MPOA projections during

food deprivation mediates hunger-induced aggression. Indeed, bath application of NPY to brain slices from sated animals partially reproduced the neural phenotype of Agg⁺ mice, reducing MPOA neuronal activity and the HCN-mediated voltage sag (Fig. 3c,f and Extended Data Fig. 5a). Consistent with this mechanism, single-cell transcriptomic data showed that around 55% of MPOA neurons coexpress *Npy* receptor genes and *Hcn* transcripts[34] (Extended Data Fig. 7g,h).

To directly test the role of NPY release from Arc^AgRP→MPOA projections, we injected a retrograde, Cre-dependent AAV expressing a short hairpin RNA (shRNA) against *Npy* into the MPOA of *Agrp-cre* mice (Fig. 3g). This led to *Npy* knockdown in Arc^AgRP→MPOA projections, whereas a control virus did not affect *Npy* expression (Fig. 3g–i). Projection-specific *Npy* knockdown in Agg⁺ mice increased the sag amplitude (that is, HCN function), reduced neuronal silencing and

increased the excitability of MPOA neurons (Fig. 3j–n). It also delayed the onset of pup-directed aggression (Fig. 3o), with attack latencies scaling with the number of transduced neurons (Extended Data Fig. 6aa), but had no effect on food intake after deprivation (Extended Data Fig. 6ab). To test whether AgRP contributes to this effect, we performed projection-specific *Agrp* knockdown. This manipulation did not alter pup-directed aggression or key MPOA properties (sag amplitude, baseline activity or excitability; Extended Data Fig. 6q–z). This finding indicates that NPY—rather than AgRP—release from Arc$^{AgRP}$→MPOA projections promotes the hunger-evoked switch to aggression.

We next examined how the reproductive state affects this system to set the switching rate. Our receptor KO experiments suggested that the oestrous state is sensed in the MPOA (Fig. 2i,j). We therefore performed whole-cell recordings from MPOA neurons of female mice across oestrous stages (Fig. 3p and Extended Data Fig. 5). Voltage sag amplitude and the proportion of neurons exhibiting voltage sag fluctuated during the oestrous cycle (Fig. 3q and Extended Data Fig. 5ac), being lowest in metestrus—when the switching rate is maximal—and highest in oestrus, when the switching rate is minimal (Fig. 2e). The switching rate was also inversely correlated with the proportion of sag-exhibiting neurons (Extended Data Fig. 5ad). We therefore proposed that a fluctuating P4/E2 ratio tunes HCN expression in MPOA neurons through the transcription factor receptors PR and ESR1. HCN channels comprise four subunits (HCN1–HCN4), all of which are expressed in the MPOA[35,36] (Extended Data Fig. 7). Using single-molecule fluorescence in situ hybridization, *Hcn1*, *Hcn2* and *Hcn4* transcript levels in MPOA neurons indeed fluctuated across the oestrous cycle, with *Hcn1* and *Hcn2* showing a substantial peak in oestrus (Fig. 3r,s and Extended Data Fig. 7c–e).

Hunger and oestrous state therefore converge on HCN channels to regulate MPOA neuron activity and excitability. The oestrous stage modulates HCN channel abundance, whereas NPY release during food deprivation inhibits available HCN channels. Neither signal alone substantially alters neuronal excitability (Extended Data Fig. 5); rather, excitability is gated by their integration. In oestrus, a low P4/E2 ratio results in a high density of HCN channels, which are only partially inhibited by NPY. As a result, MPOA neurons remain active and excitable even after food deprivation. By contrast, the high P4/E2 ratio during metestrus reduces HCN channel number, which enables more effective inhibition by NPY. This in turn leads to quiescent MPOA neurons with low activity and excitability, thereby promoting aggression (Fig. 3w). To test this model, we administered a HCN channel blocker into the MPOA of non-food-deprived mice before behavioural testing (Fig. 3t,u). Application of the blocker, but not vehicle, induced pup-directed aggression with a short latency in sated mice (Fig. 3v and Extended Data Fig. 6ac), without affecting feeding (Extended Data Fig. 6ad). HCN-mediated inhibition of MPOA neurons is therefore sufficient to switch females to pup-directed aggression.

## MPOA neurons encode an aggressive state

These results suggest that quiescent MPOA neurons promote aggression towards pups. To better understand how the biophysical changes associated with hunger and oestrous state affect neural function in vivo, we performed cellular-resolution calcium imaging during pup interactions (Fig. 4a,b). Using a head-mounted miniature microscope, we tracked the activity of individual MPOA neurons before and after food deprivation (Extended Data Fig. 8). Among the six recorded female mice, one was in proestrus, two in oestrus, two in metestrus and one in diestrus. Both mice in oestrus were non-aggressive (Agg$^-$), whereas all others were Agg$^+$, consistent with our model in which oestrous stage interacts with food deprivation to shape behavioural outcomes. This pattern suggests that the low P4/E2 ratio characteristic of oestrus biases animals towards an Agg$^-$ phenotype, whereas the increased ratio during metestrus promotes pup-directed aggression. Similar to our findings in

brain slices, baseline activity (Methods) was significantly lower in Agg$^+$ than in Agg$^-$ mice (Fig. 4c). This difference was already present before food deprivation (Pre$^+$ versus Pre$^-$), and therefore probably reflects an influence of oestrous state. Although the slice electrophysiology results suggested that food deprivation decreases the baseline activity of MPOA neurons, we did not detect this effect in vivo (Pre$^+$ versus Agg$^+$, Pre$^-$ versus Agg$^-$), which may be due to the limited sensitivity of one-photon calcium imaging. In Agg$^+$ mice, however, MPOA responses to pup chemoinvestigation and grooming were suppressed, which may reflect reduced neuronal excitability (Fig. 4d and Extended Data Fig. 9a,f). Moreover, the absolute tuning of MPOA neurons—defined as the magnitude of activation or inhibition—was reduced in Agg$^+$ mice (Fig. 4e, Methods and Extended Data Fig. 9e).

Notably, MPOA neurons remained responsive during pup-directed aggression, with both increases and decreases in activity observed across the population (Fig. 4f). This pattern resulted in a near-zero population average (Fig. 4g), consistent with previous population-level fibre photometry recordings[20]. Many MPOA neurons responded during individual pup-directed aggression episodes (Fig. 4f and Extended Data Figs. 8d and 9q), but their activity showed sustained changes—primarily inhibition (Fig. 4h)—after aggression onset and correlated more strongly with a prolonged aggressive state (from aggression onset to the end of the assay) than with specific behavioural episodes (Fig. 4h and Extended Data Fig. 9s). To test whether a persistent neural state emerged after aggression, we quantified aggression selectivity across post-aggression activity epochs using a receiver operating characteristic (ROC)-based approach (Methods). In contrast to grooming and sniffing—which showed strongly skewed selectivity distributions consistent with transient, event-linked encoding—aggression-related selectivity values were centred around 0.5. This pattern indicates the presence of a sustained, population-level activity state rather than time-locked responses (Extended Data Fig. 9r). Projecting MPOA population activity onto its first two principal components (PCs) revealed a distinct state along PC2 in Agg$^+$ mice (Fig. 4i). This state was reliably inferred in an unsupervised manner using a hidden Markov model (HMM), which detected the majority (94.9 ± 11.3%) of aggression-associated neural activity episodes (Fig. 4j,k). Here too, the inferred HMM state more strongly tracked a sustained aggressive state than discrete attack events (Extended Data Fig. 9w). By contrast, HMM states associated with pup sniffing and grooming showed weaker correspondence to those behaviours (Extended Data Fig. 9x), which suggests that aggression is linked to a distinct and persistent neural state in MPOA neurons. In support of this interpretation, baseline MPOA activity progressively declined across repeated aggression episodes, and the extent of this inhibition predicted the latency to the next attack (Extended Data Fig. 9y–aa). This finding implies that the aggressive state is self-reinforcing, with MPOA neurons becoming increasingly suppressed as aggression escalates.

To assess how well this state could be identified from population activity, we trained a linear support vector machine (SVM) on the first two PCs, which successfully decoded aggression at a level comparable to a SVM trained on the full neural dataset (Fig. 4l). This finding indicates that PC1 and PC2 capture a robust and low-dimensional signature of pup-directed aggression. The contribution of individual neurons to this aggression state—as reflected in their PC2 loading—was correlated with their capacity to predict pup aggression (Extended Data Fig. 9u). As male mice are spontaneously infanticidal, even when sated, we also examined MPOA activity patterns associated with pup-directed aggression in male mice and observed a similar state (Extended Data Fig. 10g–k). Of note, MPOA neurons tuned (that is, activated or inhibited) to aggression were often also responsive to pup grooming, both before and after food deprivation (Extended Data Fig. 10a–f), which suggests that affiliative and aggressive behaviours recruit overlapping neuronal populations. Thus, in addition to their role in parental behaviour, MPOA neurons encode a distinct state for pup-directed aggression.

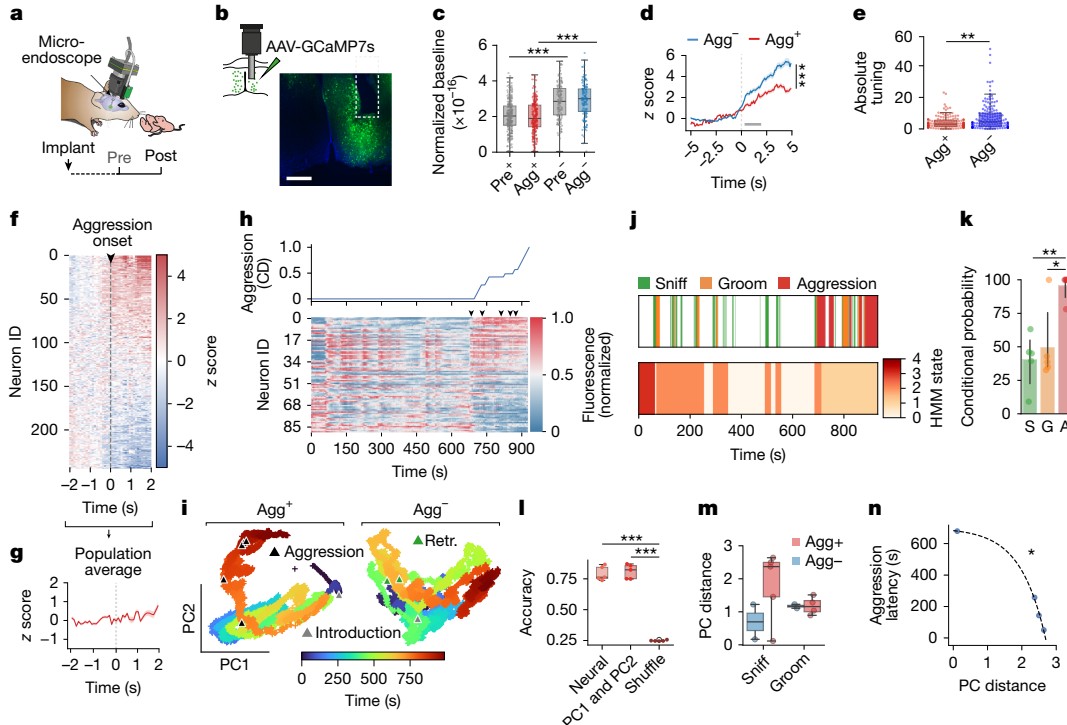

**Fig. 4 | MPOA neurons encode an aggressive state. a**, Schematic of miniature microscopy recordings. **b**, Image of lens placement in the MPOA. Scale bar, 500 μm. **c**, Normalized baseline activity (raw fluorescence; Pre$^+$ and Pre$^-$ are mice later classified as Agg$^+$ or Agg$^-$, respectively; $n$ = 243 (Agg$^+$) and 148 (Agg$^-$) neurons from $N$ = 5 (Agg$^+$) and 2 (Agg$^-$) mice). **d,e**, $z$ scored neuronal responses (**d**) and absolute tuning index (**e**) during pup chemoinvestigation ($n$ = 37 (**d**, Agg$^+$), 46 (**d**, Agg$^-$), 154 (**e**, Agg$^+$) and 148 (**e**, Agg$^-$) neurons from $N$ = 4 (Agg$^+$) and 2 (Agg$^-$) mice). Dashed line, sniffing onset; bar, mean bout duration (2.1 s). **f**, Responses during pup-directed aggression, with hierarchical clustering based on the mean onset (first episode, $n$ = 243 neurons from $N$ = 5 mice). ID, identifier. **g**, Averaged, $z$ scored responses during aggression onset. **h**, Neuronal responses from Agg$^+$ animals, sorted by the correlation with the cumulative distribution (CD) of aggression (67 neurons). Arrowheads indicate aggression episodes. **i**, Population activity projected onto PC1 and PC2. Retr., pup retrieval. **j**, Ethogram

and HMM state segmentation using Agg$^+$ neural data. **k**, Conditional probability of behaviours when in the HMM state most frequently aligned with them. S, sniff; G, groom; A, aggression ($N$ (left to right) = 5, 5 and 5 mice). Data are the mean ± s.e.m. **l**, Behavioural prediction accuracy from the SVM classifier trained on neural data, PCs or shuffled data ($N$ = 5 mice). **m**, Pre versus Post PC distances during pup chemoinvestigation and grooming ($N$ = 5 (Agg$^+$) and 2 (Agg$^-$) mice). **n**, Exponential fit of PC distance versus aggression latency ($n$ = 4). 'Aggressive state' = onset to assay end. Statistics: linear mixed-effects model with mouse ID as random effect (**c**); two-way ANOVA with Tukey post hoc test (**d**); $U$-test (two-sided) (**e,m**); one-way ANOVA with Tukey post hoc test (**k,l**). Data are mean the ± s.e.m. Box plots show the median (line) and the interquartile range (box), and whiskers are 1.5× the IQR. Shaded areas (**d,g**) represent 95% CI. *$P$ < 0.05, **$P$ < 0.01, ***$P$ < 0.001. See Supplementary Table 3 for further details of statistical analyses.

This finding raises the question of what drives the transition of MPOA population dynamics into this aggression state. Hunger and oestrous state may alter pup representations in the MPOA by modulating neuronal excitability. To test for changes in neural responses during pup chemoinvestigation before and after food deprivation, we used PC distance as a measure of representational similarity. Agg$^+$ mice exhibited increased shifts in pup representations (Fig. 4m), and PC distance was inversely correlated with aggression latency, which indicates that greater changes in pup representation were associated with a faster onset of attack (Fig. 4n). These results suggest that hunger and oestrous state promote an aggression state by altering pup representations in the MPOA.

## Discussion

Through the combination of behavioural, circuit-level and cellular approaches, we demonstrated how hypothalamic neurons integrate hunger and oestrous state to drive a switch in social behaviour. We identified HCN channels as molecular integrators of these states in MPOA neurons, whereby baseline channel expression is dynamically set across the oestrous cycle. Notably, the behavioural switch is a function of the P4/E2 ratio rather than individual hormone levels. Genome-wide targets of ESR1 were recently identified in the brain, including *Hcn1* and *Pgr*[37]. Consistent with this finding, administration of E2 increases

*Hcn1* expression[37] (Extended Data Fig. 7f), and the chromatin accessibility of *Hcn* fluctuates across oestrous stages[38]. Although the targets of PR remain less well characterized, it has been shown to inhibit *Esr1* (refs. 39,40). Reciprocal interactions between ESR1 and PR, as well as coordinated DNA binding of both receptors[38], therefore probably contribute to hormone ratio sensing. Such sensing may occur in individual MPOA neurons that express both ESR1 and PR[3], and/or across distinct neuronal populations with differing sensitivities to each hormone. Notably, a large proportion of MPOA neurons coexpresses the key components central to our model (*Esr1*, *Pgr*, *Npyr* and *Hcn*), which therefore enables state integration in individual neurons (Extended Data Fig. 7g,h). Behavioural differences between Agg$^+$ and Agg$^-$ mice in the same oestrous stage may reflect individual variability in hormone levels or receptor expression. For example, variable *Esr1* expression in the MPOA has been linked to parental performance in lactating females[41].

The oestrous state modulates *Hcn* expression, whereas food deprivation inhibits HCN channel function through NPY (Fig. 3w). The underlying NPY receptor subtypes and downstream signalling pathways remain to be identified, but approximately 57% of MPOA neurons express either of the NPY receptors Y1 or Y2, both of which inhibit HCN channels by reducing cAMP levels through $G_{i/o}$-protein-coupled mechanisms[42–45]. Although *Hcn* transcript levels were relatively low during diestrus, the sag amplitude remained high (Fig. 3q,s). This discrepancy may arise from differences in the timing of data collection in the prolonged

(around 2 days) diestrus phase, during which the P4/E2 ratio gradually declines, or from post-transcriptional modulation. For instance, hypothalamic cAMP levels fluctuate across the oestrous cycle[46], and increasing cAMP levels during diestrus may enhance HCN channel function despite reduced *Hcn* expression[47].

Knockdown of *Npy* in Arc[AgRP]→MPOA projections delayed, but did not completely abolish, pup-directed aggression (Fig. 3o). This effect might result from incomplete AAV transduction or *Npy* knockdown efficiency (Fig. 3i). Along with shorter aggression latencies after prolonged food deprivation (Extended Data Fig. 1a), this result suggests that NPY release progressively increases during food deprivation. Consistent with this finding, the addition of NPY receptor antagonists after food deprivation led to variable biophysical effects on MPOA neurons (Extended Data Fig. 5e–i). The observation that 30 min of Arc[AgR]→MPOA pre-stimulation triggered the transition to pup-directed aggression (Fig. 1k) suggests that the behavioural switch requires sustained NPY release and/or slow integration of the neuropeptidergic signal in the MPOA. This result aligns with previous work showing that prolonged Arc[AgRP] activation is necessary for maximal NPY-dependent feeding responses[11,21,33]. As NPY is released from dense-core vesicles and may act through volume transmission and slow-acting GPCR pathways[48], extended stimulation may be needed to reach effective levels of neuromodulation. These effects were detectable across the MPOA, including in parenting-relevant MPOA[Gal] neurons (Extended Data Fig. 4u–z).

HCN channels have a well-established role in neuronal excitability[35,44] and have a substantial impact on states such as sexual satiety and anxiety[45,49,50]. Reduced HCN function shifts MPOA neurons into a quiescent state with reduced baseline activity and excitability (Fig. 3b–d). Aspects of this reduced excitability are also seen in vivo, in which pup-induced activity in MPOA neurons was significantly weaker in Agg[+] mice (Fig. 4c,d). Previous studies have shown that MPOA lesions and optogenetic inhibition induce pup-directed aggression[19,20,51], and bulk calcium imaging suggests that MPOA neurons are largely silent during pup attacks in virgin females[20]. These findings support a model in which aggression primarily results from disinhibition of aggression-promoting neurons downstream of the MPOA[20]. By contrast, our cellular-resolution recordings revealed that most MPOA neurons are either activated or inhibited during aggression, resulting in a minimal net response (Fig. 4f,g). Although MPOA neurons exhibit behavioural tuning—confirming previous work[20]—their activity was even more correlated with an aggression state, which was reliably decoded using both supervised and unsupervised approaches (Fig. 4j,l and Extended Data Fig. 9). Future studies will investigate whether the neurons encoding this state have specific molecular signatures and/or connectivity profiles.

This state-dependent switch may provide behavioural flexibility to enable adaptive responses to pups during periods of food scarcity, as observed in male gerbils after prolonged food deprivation[52]. Food-deprived mice are more likely to consume prey (Extended Data Fig. 1g), but do not seem to perceive pups as food because pup interactions increase Arc[AgRP] activity (Extended Data Fig. 1k–m), in contrast to the suppression of this population observed in response to food cues[15,16]. Although ethical and legal constraints prevent us from assessing whether Agg[+] females cannibalize pups, this interpretation is supported by two additional observations: first, pup and food representations in the MPOA differed after food deprivation (Extended Data Fig. 9p); and second, a similar state occurred in males during pup-directed aggression (Extended Data Fig. 10). Beyond regulating feeding, Arc[AgRP] neurons coordinate numerous behavioural adaptations to food deprivation through different projections[7,17,21–24,53–56]. Our findings, along with a recent study[57], extend their role to the modulation of pup interactions. Notably, repeated pup exposure (sensitization) seems to prevent the hunger-induced switch to pup-directed aggression through an unknown mechanism[57]. Future work will explore how social experience modulates the Arc[AgRP]→MPOA circuit to shape infant-directed behaviour.

A central question in neuroscience and physiology is how internal states drive adaptive behavioural change[10]. Recent work has begun to uncover how hunger and thirst jointly regulate ingestive behaviours[58,59]; however, far less is known about how multiple physiological states interact to shape social behaviour. Our work identifies a neural mechanism by which internal states impart flexibility to pup interactions and provides a conceptual framework for exploring how other states are integrated in the brain.

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

## Methods

### Ethical compliance

All animal procedures performed in this study were approved by the UK Government (Home Office) and by the Crick Institutional Animal Welfare Ethical Review Panel.

### Mice

Animals were housed in individually ventilated cages on a 12/12-h light–dark cycle (lights on: 22:00–10:00) at 21 °C and 32% humidity with food and water available ad libitum. Standard mouse chow (2018S Teklad Global 18% Protein Rodent Diet) was used in all experiments. Baseline (Pre) behavioural testing was performed in the first 4 h of the dark phase, and testing after food deprivation (Post) was performed 6 h after the start of the Pre phase, unless stated otherwise.

C57BL/6J mice (*Mus musculus*) from the Crick breeding colonies were used at age 8–14 weeks for all behavioural experiments. *Agrp-cre* mice[32] (The Jackson Laboratory, JAX 012899) were used to target Arc[AgRP] neurons. For slice physiology experiments, this line was crossed to Cre-dependent *Rosa26 Tomato* mice (Ai9, The Jackson Laboratory, JAX 007909). For hormone receptor KO experiments, *Esr1[loxP]* (oestrogen receptor α conditional KO, imported from EMMA, EM:11179)[60] or *Pr[loxP]* (progesterone receptor conditional KO, made in-house)[3] were used. All lines were maintained in a C57BL/6J background. Unless otherwise noted in the figure legends, all experiments were performed in female mice.

### Behavioural profiling

Virgin females without previous pup exposure were used in all experiments. For experiments in pregnant females, virgin females were paired up with an experienced stud male until a vaginal plug was detected, which marked pregnancy day 1 (D1). Behavioural scoring and analysis were performed by an individual blind to the experimental condition of the animal (for example, Pre versus Post, manipulation versus control).

**Pup-directed behaviour assay.** Animals were individually housed for 4 days before behavioural testing. Experiments were performed in the home cage and were preceded by a 10-min habituation period. Two C57BL/6J pups 1–3 days of age were placed in different corners opposite the nest, and pup interactions were recorded for 15 min with a Basler Ace GigE, acA1300-60gmNIR camera. Videos were acquired at a frame rate of 30 Hz using a custom protocol written in Bonsai (NeuroGEARS, https://bonsai-rx.org/) and behaviours were scored using behavioural observation research interactive software (BORIS)[61]. Pup-directed behaviours were classified as follows: contact latency was defined as the time elapsed until the first contact of the test animal's nose with a pup; pup grooming was defined as physical contact with pups involving licking, pup displacement and rhythmic head movements; and pup chemoinvestigation was defined as close interaction with the nose of the animal touching the pup but no additional physical contact. The onset of pup retrieval was defined by the time elapsed until a pup was picked up and retrieved to a nest. Time in nest was defined as the time the female mouse stayed in the nest with at least one pup. Crouching was defined as the female mouse stationarily positioned over pups in the nest. Total parenting time was calculated as the sum of time spent grooming pups, retrieving pups and time spent in the nest with at least one pup. Nest building was defined as collecting bedding or nesting material and bringing it to the nest and shaping it into a new nest. Food-deprived mice were classified as aggressive (Agg⁺) or non-aggressive (Agg⁻) as follows: after an initial chemoinvestigation and grooming phase, Agg⁻ animals exhibit non-aggressive behaviours such as pup retrieval, nest building, rearing and digging (Extended Data Fig. 1). Agg⁻ animals were further classified as 'parental' or 'ignoring' based on whether initial chemoinvestigation and grooming were followed by parental behaviour components. Parental animals retrieved pups after a brief grooming period. Once in the nest, they remained with the pups, crouched above them and engaged in grooming and occasionally nest building (Extended Data Fig. 1p,q). By contrast, ignoring animals only performed non-pup-related behaviours—such as rearing and digging—after initial chemoinvestigation and grooming (Extended Data Fig. 1r,s). Aggressive contact was defined as close interactions with pups involving rapid, rhythmic head movement, biting or aggressive carrying of pups around the cage[62,63]. In behavioural experiments, if a pup was attacked, all pups were immediately removed, and the trial was terminated. During in vivo imaging experiments, if any pup was attacked, attacked pups were promptly replaced with new pups to enable the observation of multiple aggression episodes. In the rare event of injury, affected pups were immediately euthanized.

**Prey assay.** House crickets (*Gryllus domesticus*, 12–20 mm in length, purchased from the Northampton Reptile Centre) were used as targets. Immediately after pup-directed behaviour assays, a cricket was placed in the cage for 15 min. Capturing, biting or biting with forepaw assistance was classified as prey-directed aggression.

**Residence intruder assay.** Male or female adult mice 8–14 weeks of age were introduced into the resident's cage immediately after the pup-directed behaviour assays in randomized order, and resident mice were allowed to interact with the intruder for 15 min. Trials in which the intruder exhibited aggression towards the resident were excluded. Mice were categorized as aggressive towards the intruder if biting and fighting occurred.

**Elevated-plus maze test.** A standard elevated-plus maze with two closed and two open arms, elevated 90 cm above ground, was used[64]. The assay was initiated by placing the mouse in the open arm of the plus maze, and animal trajectories were recorded for 10 min. Videos were captured and analysed using EthoVision XT 14 (Noldus).

**Open-field test.** A white behaviour test box (60 × 60 × 30 cm, length × width × height) was virtually divided into a centre (30 × 30 cm) and a periphery. A mouse was placed in the periphery and recorded for 10 min to measure the time spent in the centre or peripheral area. Videos were captured by a top camera and analysed using EthoVision XT 14 (Noldus). Custom detection profiles were set for each mouse, and the detection threshold was adjusted so that the mouse could be detected in >95% of video frames. The time spent in the closed versus open arm, and centre versus periphery, average speed and total time spent moving were quantified using the EthoVision animal tracking pipeline.

**Food intake.** Mice were single-housed for 4 days before food intake was measured. On the day of measurement, animals were provided with fresh bedding to avoid leftover food crumbs in the cage. A Petri dish with food pellets was provided and 1 h of food intake was quantified by calculating the weight difference of the Petri dish. Food intake on behavioural testing days was measured immediately after pup interactions. Baseline food intake was quantified on the day before behavioural testing during the same circadian time (4 h before the end of the dark phase) in sated mice. For refeeding, Agg⁺ mice were provided with food ad libitum for 1 or 2 h before pup interactions were assessed.

**FOS mapping.** To identify brain areas that are differentially recruited between aggressive and non-aggressive pup interactions in food-deprived mice, pup-directed behavioural assays were performed as described above (see the section 'Pup-directed behaviour assay'). At 90 min after the first pup contact, mice were deeply anaesthetized and rapidly transcardially perfused with 30 ml ice-cold PBS, followed by 30 ml ice-cold paraformaldehyde (PFA) (4% in PBS). Brains were dissected and post-fixed in PFA (4% in PBS) at 4 °C for 16 h. The next day, brains were rinsed with cold PBS and 60 μm coronal sections

were prepared with a vibratome (Leica VT1000 S). Sections were further post-fixed in PFA (4% in PBS) at room temperature for 10 min and immunostaining against FOS was performed (see the section 'Immunohistochemistry'). Brain sections were imaged on a slide scanner, and FOS$^+$ cell densities were quantified between sections from Agg$^+$ and Agg$^-$ mice using QuPath software (see the section 'Imaging').

## Mass spectrometry

Trunk blood was collected into EDTA tubes and samples were centrifuged at 2,000$g$ for 10 min at 4 °C using a microcentrifuge. The supernatant (serum) was pipetted into a fresh 1.5-ml tube and samples stored at −80 °C. Next, 10 μl of serum was mixed with 30 μl ice-cold methanol to induce protein precipitation. Samples were briefly vortexed, placed on ice for 5 min and centrifuged at 4 °C for 10 min. Next, 30 μl of extract was mixed with 270 μl methanol, and 30 μl of the diluted extract was transferred to a vial equipped with an insert, followed by the addition of 1 nmol Scyllo-inositol (Sigma). Samples were dried and derivatized with 20 μl freshly prepared methoxyamine (20 mg ml$^{-1}$, in pyridine) (both Sigma) at room temperature for >10 h, followed by a second step of derivatization with 20 μl BSTFA + 1% TMCS (Sigma) performed at room temperature for 1 h. Data acquisition was performed largely as previously described[65] using an Agilent 7890B-7000C GC-MSD in EI mode. GC−MS parameters were as follows: carrier gas, helium; flow rate, 0.9 ml min$^{-1}$; column, DB-5MS (Agilent); inlet temperature, 270 °C; temperature gradient, 70 °C (2 min), ramp to 295 °C (12.5 °C min$^{-1}$), ramp to 320 °C (25 °C min$^{-1}$, 3 min hold). The scan range was $m/z$ = 50–550. Data analysis was performed using MANIC software (v.3.0.20)[66]. Metabolites were identified and quantified by comparing to the authentic standard of ghrelin (Anaspec AS-24160).

## Oestrous cycle staging

Vaginal smears were taken immediately after pup interaction assays. Animals were scruffed and 20 μl of PBS was gently pipetted several times at the surface of vagina. Samples were air-dried and stained with 10 μl crystal violet (C.I. 42555, Merck). Mouse identifiers were shuffled, and the oestrous cycle was assessed by an individual blind to aggression phenotype (see ref. 67).

## Histology and immunostaining

**Perfusion and tissue sectioning.** Animals were transcardially perfused with PBS followed by 4% PFA in PBS. Brains were dissected and post-fixed in 4% PFA overnight at 4 °C then washed in PBS. After embedding in 4% low-melting point agarose (Thermo Fisher, 16520-050) in PBS, 60-μm coronal sections were cut on a vibratome (Leica) and mounted on Superfrost Plus slides (VWR, 48311-703) with DAPI-containing Vectashield mounting medium (Vector Laboratories, H-1200). Acute, 250-μm-thick brain sections from electrophysiological recordings were post-fixed in 4% PFA in PBS with 200 mM sucrose (Sigma-Aldrich, S5016) and 0.1 M HEPES (Sigma-Aldrich, H3375) at 4 °C on a nutator overnight, rinsed in PBS and washed in PBS-T (0.3% Triton X-100 in PBS) for 1 h.

**Immunohistochemistry.** Immunostaining was performed in 48-well tissue culture plates. Brain sections were permeabilized for 30 min in PBS-T (0.3% Triton X-100 in PBS), post-fixed with 4% PFA in PBS for 10 min and washed in PBS (3× 20 min). Blocking was carried out for 3 h at room temperature in blocking buffer (3% BSA, 2% normal donkey serum in PBS). Incubation with primary antibodies (in PBS) was performed for 24–48 h on a nutator at 4 °C. After washing in PBS (3× 20 min), secondary antibodies were added in PBS-T for 48 h at 4 °C. After final washes in PBS-T (3× 20 min), sections were mounted. The following primary antibodies were used: rabbit anti-FOS (Synaptic Systems, 226003, 1:2,000); rabbit anti-NPY (Abcam, ab30914, 1:500); and rabbit anti-AgRP (Abcam, ab254558, 1:500). The following secondary antibodies were used: donkey anti-rabbit Alexa Fluor-568 (Thermo Fisher, A-11057, 1:2,000); donkey anti-rabbit Alexa Fluor-647 (Thermo

Fisher, A-21245, 1:2,000); and goat anti-rabbit Alexa Fluor-647 (Thermo Fisher, A-21244, 1:1,000).

**In situ hybridization.** Animals were transcardially perfused with ice-cold PBS, and freshly dissected brains were embedded in OCT (Tissue-Tek, 4583), frozen on dry ice and stored at −80 °C. Subsequently, 18-μm cryosections were cut on a Leica CM1950 cryostat and collected on Superfrost Plus slides (VWR, 48311-703) in three series, only one of which was stained and imaged. Slides were fixed in 10% neutral buffered formalin, followed by a series of dehydration steps in ethanol (5 min each of 50%, 70%, 100% and 100% v/v ethanol). Slides were pretreated with RNAscope protease III reagent for 30 min at 40 °C. Single-molecule fluorescent in situ hybridization was performed on slides using a RNAscope LS Multiplex Reagent kit (Advanced Cell Diagnostics), a LS 4-Plex Ancillary kit and a Multiplex Reagent kit on a robotic staining system (Leica BOND-III). RNAscope probes were *Hcn1* (ACD, 423658), *Hcn2* (427009), *Hcn3* (551528) and *Hcn4* (421278). Immunostainings against the neuronal marker NeuN were subsequently performed (Millipore, MAB377, 1:500).

**Imaging.** Images were acquired on a Vectra Polaris Automated Quantitative Pathology Imaging system (Akoya Biosciences) at ×20 magnification. Regions of interest (ROIs) were selected using Phenochart software (Akoya Biosciences) and image tiles were spectrally unmixed using inForm Tissue Analysis software (Akoya Biosciences). Stitching of spectrally unmixed image tiles and image analyses were performed in QuPath software[68]. FOS-positive nuclei (or NeuN-positive neuronal cell bodies) were first detected using custom QuPath scripts. Detection of *Esr1*, *Pgr* and *Hcn* transcripts was subsequently performed on cell body detections. Thick brain sections (250 μm) were imaged on an upright confocal microscope (Zeiss LSM 710) using a ×63 (NA 1.4) oil-immersion objective and a $z$ step size of 0.5 μm.

## Surgical and recording procedures

Analgesia was provided 1 day before surgery (0.15 ml carprofen in 200 ml drinking water). Mice were anaesthetized using isoflurane (4% for induction, 1.5% for maintenance) in oxygen-enriched air and head-fixed in a stereotactic frame (Model 940, Kopf Instruments). Meloxicam (10 mg kg$^{-1}$ body weight) and buprenorphine (0.1 mg kg$^{-1}$ body weight) were given subcutaneously before craniotomy. The surgery site was closed using Vicryl sutures (Ethicon) or Vetbond surgical glue (3M). Carprofen was provided in drinking water for 2 days after surgery for postoperative pain management. Eyes were protected with ophthalmic ointment (Viscotears, Alcon). The rectal body temperature was maintained at 37 °C during surgery using a heating pad (Harvard Apparatus) and animals were kept in a heated recovery chamber until fully mobile. Animals were allowed to recover for at least 2 weeks before behavioural testing.

**Brain coordinates.** See Supplementary Table 1 for injection, implantation and recording coordinates. Coordinates are anteroposterior/mediolateral/dorsoventral and in mm. Dorsoventral coordinates are measured from the brain surface. Chemogenetic effectors were injected into two rostrocaudal Arc coordinates (−1.4/±0.25/−5.90 and −1.6/±0.25/−5.90 mm) to maximize the number of transduced neurons. For projection-specific *Npy* and *AgRP* knockdown, MPOA coordinates were adjusted to 0.0/±0.3/−5.05 mm to maximize the number of retrogradely labelled Arc$^{AgRP}$ neurons.

**Chemogenetics.** For chemogenetic activation, 200 nl AAV5-hSyn-DIO-hM3Dq(Gq)-mCherry (Addgene, 44361, 2.5 × 10$^{13}$ genome copies (GC) per ml) or AAV5-hSyn-DIO-mCherry (Addgene, 50459, 1.8 × 10$^{13}$ GC per ml) was injected into the Arc (see Supplementary Table 1 for coordinates). After assessment of spontaneous pup-directed behaviours 3 weeks after viral injection, CNO (Bio-Techne

12352200, 3 mg kg$^{-1}$) was intraperitoneally injected, and pup-directed behaviour was assessed 30 min later. For chemogenetic inhibition, 250 nl AAV5-loxP-hGlyAG-2A-nlsVenus (1.6 × 10$^{13}$ GC per ml, Crick Vector Core) was prepared from a pAAV-loxP-hGlyAG-2A-nlsVenus plasmid[14,69] (a gift from H. Fenselau) and injected into the Arc (see the section 'Brain coordinates'). Ivermectin (5 mg kg$^{-1}$, dissolved in 7:3 propylene glycol and glycerol) was injected 24 h before the start of food deprivation, and behaviour was assessed after 6 h of food deprivation.

**Optogenetics.** To optogenetically activate Arc$^{AgRP}$ projections, 250 nl AAV5-EF1a-DIO-ChR2-EYFP or AAV1-EF1a-DIO-ChR2-EYFP (Addgene, 20298, 0.7 × 10$^{13}$ GC per ml) or AAV1-EF1a-DIO-YFP (Addgene, 27056, 2.5 × 10$^{13}$ GC per ml) was injected into the Arc (see Supplementary Table 1 for coordinates). During the same surgery, optic fibres (Doric Lenses) were implanted 200–400 μm above the target area (MPOA: dual fibre cannula 200/245 μm, 0.37 NA, GS1.0; LPOA: dual fibre cannula 200/245 μm, 0.37 NA, GS2.0; PVH: mono fibre cannula 400/470 μm, 0.37 NA). After 2–3 weeks of recovery, animals were connected to matching patch cords connected to a laser (Stradus 473–80 nm, Vortran) through a commutator (RJ1, Thorlabs). Four distinct protocols for optogenetic stimulation were used: acute stimulation whenever animals were close to a pup; or 15, 30 or 60 min of pre-stimulation followed by a 15-min pup-directed behaviour assay. A period of 3–4 days was allowed between two consecutive optogenetic experiments to prevent sensitization to pups. The light power exiting the fibre tip corresponded to an irradiance of 4.68 mW mm$^{-2}$ at the target region (http://www.stanford.edu/group/dlab/cgi-bin/graph/chart.php). For acute stimulation, blue light was delivered in 20-ms pulses at 20 Hz for 1–4 s whenever the animal contacted a pup with its snout. In the pre-stimulation protocols, cycles of 1 s of 20 Hz stimulation followed by 4 s without stimulation were delivered for the indicated duration[15].

**Hormone receptor KO.** AAV2/5-CMV-EGFP-Cre (250 nl, Addgene, 105545, 2 × 10$^{13}$ GC per ml) was injected into the MPOA (see the section 'Brain coordinates') of $Esr1^{loxP}$ or $Pr^{loxP}$ mice. Animals were tested 3 weeks after injection, and brain slices were subsequently prepared for histological analyses. The efficiency of viral-genetic receptor KO was established in a separate experimental cohort of $Esr1^{loxP}$ or $Pr^{loxP}$ animals that received unilateral MPOA injections of either AAV2/5-CMV-EGFP-Cre or AAV2/5-CMV-EGFP (250 nl, Addgene 105530, 2 × 10$^{13}$ GC per ml), and which has since been published[3].

**Gene knockdown.** Constructs for shRNA-mediated knockdown of $Npy$ and $Agrp$ were developed using the Broad Institute's hairpin design tool (https://portals.broadinstitute.org/gpp/public/seq/search) on the $Npy$ (NM_023456.3, position: 3728–3748) and $Agrp$ (NM_007427.3, position: 187–648) coding sequences. The following sequences were used: (1) Npy_817 CACTGATTTCAGACCTCTTAACTCGAGTTAAGAGGTCTG AAATCAGTG TTTTT; (2) Npy_818 GCTCTGCGACACTACATCAATCTCG AGATTGATGTAGTGTCGCAGAGCTT TTT; (3) Agrp_50 GTTCCCAGG TCTAAGTCTGAACTCGAGTTCAGACTTAGACCTGGGAACTT TTT; (4) Agrp_51 GGCAGGGGATGAGAATAAACTCGAGTTTATTCTCATC CCCTGCCTTTTT; (5) Agrp_4 GGCAAAGATCAGCAAGCAACTCG AGTTGCTTGCTGATCTTTGCCTTTTT (where TTTTT indicates the termination signal). Using NEBuilder, these oligonucleotides were cloned into the HpaI/SpeI sites of pAAV-G-Creon shRNA[Control] plasmid (Addgene, 181824)[70], which generated the constructs pAAV-G-CreON-shRNA_817-NPY-GFP, pAAV-G-CreON-shRNA_818-NPY-GFP, pAAV-G-CreON-AGRPshRNA-GFP-50, pAAV-G-CreON-AGRPshRNA-GFP-51 and pAAV-G-CreON-AGRPshRNA-GFP-4, respectively. As a negative control, a scrambled sequence (CCTAAGGTTAAGTCGCCC TCGCTC GAGCGAGGGCGACTTAACCTTAGGTTTTTT) was designed using VectorBuilder.

pAAV-G-CreON-shRNA_817-NPY-GFP, pAAV-G-CreON-shRNA_818-NPY-GFP, pAAV-G-CreON-AGRPshRNA-GFP-50, pAAV-G-CreON-AGRPshRNA-GFP-51 and pAAV-G-CreON-AGRPshRNA-GFP-4 (see above) were packaged as rAAV2-retro capsids and the titre was measured by qPCR. For projection-specific knockdown of $Npy$, 400 nl of a 1:1 mix of AAV-retro-G-CreON-shRNA_817-NPY-GFP (3.8 × 10$^{13}$ GC per ml) and AAV-retro-G-CreON-shRNA_818-NPY-GFP (2.3 × 10$^{13}$ GC per ml) was bilaterally injected into the MPOA (see the section 'Brain coordinates'). For projection-specific knockdown of $Agrp$, 400 nl of a 1:1:1 mix of AAV-retro-G-CreON-AGRPshRNA-GFP-50 (1.0 × 10$^{13}$ GC per ml), AAV-retro-G-CreON-AGRPshRNA-GFP-51 (1.6 × 10$^{13}$ GC per ml) and AAV-retro-G-CreON-AGRPshRNA-GFP-4 (1.3 × 10$^{13}$ GC per ml) was bilaterally injected into the same coordinates. As control, AAV-retro-CreON-shRNA-scr expressing a scrambled shRNA (400 nl, 1.78 × 10$^{13}$ GC per ml) was injected. Behavioural testing and/or electrophysiological recordings were performed 3 weeks after injection.

**Cannulation experiments.** Mice were implanted with stainless-steel bilateral guide cannulas (C235GS-5- 1.0/SPC, Protech International) 0.2 mm above the MPOA. Cannulas were fixed to the skull with dental cement. Dummy cannulas (C235DCS-5/SPC, Protech International) were inserted into guide cannulas to prevent clogging and closed with a dust cap. Mice were allowed to recover for 4 days. One hour before behavioural testing (see the section 'Pup-directed behaviour assay'), 1 μl of ZD-7288 (Tocris 1000; 1 mM, in sterile artificial cerebrospinal fluid (ACSF)) or ACSF alone (vehicle) was administered to each side of the cannula at a rate of 0.5 μl min$^{-1}$.

**Fibre photometry.** AAV-hsyn-DIO-GCaMP7s (Addgene, 104491-AAV1, 300 nl, 1.5 × 10$^{13}$ GC per ml) was injected into the Arc of $Agrp$-cre mice and a 200 μm fibre-optic cannula (MFC_200/230-0.37_6mm_MF1.25_FLT, Doric Lenses) was implanted into the MPOA (see Supplementary Table 1 for coordinates). The cannula was fixed to the skull using UV light-curable glue (RelyX Unicem, 3M) and Superbond cement (Prestige Dental). Recordings were performed 3 weeks after surgery using a FP3001 fibre photometry system (Neurophotometrics). In brief, two LEDs (415 nm and 470 nm, light power of about 50 μW) were pulsed at 20 Hz in an interleaved manner to obtain an isosbestic motion signal (415 nm) and GCaMP activity (470 nm). A FLIR 277 BlackFly CMOS camera was used to detect fluorescent signals, and acquisition was controlled (and synchronized to the acquisition of behavioural video recordings) using Bonsai.

**Miniature microscopy imaging.** AAV2/1-syn-GCaMP7s (Addgene, 104487, 100–200 nl, 2 × 10$^{13}$ GC per ml) was unilaterally injected into the MPOA of C57BL/6J mice using a Nanoject II or Nanoject III injector (Drummond Scientific) and pulled glass capillaries (3-000-203-G/X, Drummond Scientific). See Supplementary Table 1 for injection and implantation coordinates. After letting the virus diffuse for 5 min, the injection needle was slowly retracted and an integrated gradient-index lens (0.6 × 7.3 mm, 1050-002177, Inscopix) was slowly implanted and fixed to the skull using UV light-curable glue (RelyX Unicem, 3M) and Superbond cement (Prestige Dental).

Recordings started 6–8 weeks after surgery. Mice were connected to a miniature microscope (nVista, Inscopix) to check for sufficient expression of GCaMP7s. Imaging data were acquired using nVista HD software (Inscopix) at a frame rate of 20 Hz with 475 nm LED power of 0.1–0.2 mW mm$^{-2}$, an analog gain of 5–8 and an image resolution of 800 × 1,280 pixels. Imaging parameters and focal depth were kept identical across sessions. Imaging and behavioural video collection were synchronized using Bonsai. Mice were connected to the microscope and allowed 20 min of habituation before recordings were performed in their home cage. A 1-min baseline was acquired before pups were introduced, which was used to calculate the relative fluorescence change for each ROI in the field of view.

## Ex vivo electrophysiology

C57BL/6J mice were deeply anaesthetized with 3% isoflurane in oxygen and decapitated. The brain was quickly dissected and placed in ice-cold slicing solution containing (in mM): sucrose (214), KCl (2), NaH$_2$PO$_4$ (1.2), NaHCO$_3$ (26), MgCl$_2$ (2), CaCl$_2$ (2) and D-glucose (10), equilibrated with carbogen (95% O$_2$/5% CO$_2$). Coronal brain slices (250 μm thick) containing the MPOA were cut on a vibratome (Leica VT1200S) in ice-cold slicing solution and transferred to an incubation chamber with ACSF containing (in mM): NaCl (127), KCl (2), NaH$_2$PO$_4$ (1.2), NaHCO$_3$ (26), MgCl$_2$ (1.3), CaCl$_2$ (2.4) and D-glucose (10), which was continuously oxygenated with carbogen. After at least 1 h of recovery at 35 °C, slices were transferred to a submersion chamber under an upright microscope with infrared Nomarski differential interference contrast optics (Slicescope, Scientifica). During recordings, slices were submerged in, and continuously perfused (1–2 ml min$^{-1}$) with, ACSF at near physiological temperature (33 °C) and continuously oxygenated with carbogen. Glass micropipettes (3–6 MΩ resistance) were pulled from borosilicate capillaries (World Precision Instruments) on a P-97 Flaming/Brown micropipette puller (Sutter) and filled with internal solution containing (in mM): potassium gluconate (140), KCl (10), KOH (1), EGTA (1), Na$_2$ATP (2), Mg$_2$ATP (2) and HEPES (10), pH 7.3, 280–290 mOsm. Access resistance was monitored throughout the experiment, and neurons in which it exceeded 25 MΩ or changed by ≥20% were excluded. The liquid junction potential was 16.4 mV and was not compensated. We characterized the intrinsic electrophysiological properties of cells using a standardized current-clamp protocol that consists of $I/V$ curves, ramps and current injections. HCN-mediated voltage sag amplitudes were measured in response to hyperpolarizing 1-s direct-current steps[71]. T-type calcium currents were assessed using a standard current-clamp protocol in which cells were hyperpolarized to −120 mV and then stepped back to −60 mV[72]. The amplitude of the resulting rebound was then quantified. To assess excitability, ramping depolarizing currents (10 pA s$^{-1}$) from +25 to +165 pA were injected. Spontaneous postsynaptic currents (sPSCs) were detected using a threshold-based detector (WinEDR v.4, template mode). The rise time was defined as the time needed for sPSC amplitudes to reach 1-e$^{-1}$ (≈63%) of its maximal value, and the time constant of decay was defined as the time needed for the sPSC amplitude to return to 1/e (≈37%) of the resting state. The HCN channel blocker ZD-7288 (Tocris 1000) was added at a concentration of 50 μM 1 h before recordings. NPY (Phoenix Pharmaceuticals 049-03) was added at a concentration of 100 μM 1 h before recordings. NPY receptor antagonists (NPY1R: 10 μM BIBP 3226, Tocris, 2707; NPY2R: 100 nM BIIE 0246, Merck, SML2450) were added 1 h before recording[73,74]. Recordings were acquired using a Multiclamp 700B amplifier (Molecular Devices), low-pass filtered at 10 kHz and digitized using a Digidata 1550B digitizer (Molecular Devices). Slow and fast capacitive components were semiautomatically compensated. Offline data analysis was performed with Clampfit 10 software (Molecular Devices), WinEDR (v.4), WinWCP (v.5; http://spider. science.strath.ac.uk/sipbs/software_ses.htm) and custom routines written in Python (v.3.7).

**Channelrhodopsin-assisted connectivity mapping.** For channelrhodopsin-assisted connectivity mapping[75], 200 nl AAV1-E F1a-FLEx-hChR2(H134R)-EYFP (Addgene, 20296, 7 × 10$^{12}$ GC per ml) was bilaterally injected into the MPOA of *Agrp-cre* mice. Acute brain sections were prepared 3 weeks after viral injection. We used a CsCl-based internal solution containing (in mM): CsCl (140), EGTA (1), Na$_2$ATP (2) and HEPES (10), pH 7.3, 280–290 mOsm. Spontaneous inhibitory postsynaptic currents were recorded in voltage-clamp configuration at −70 mV in the presence of 1 μM TTX (Alomone T-550) and 100 μM 4-AP (Sigma 275875). Drugs were washed in at least 10 min before recordings. Photostimulation was delivered from a 490 nm LED (pE-100, CoolLED) through a ×60 objective and consisted of 2–10 ms of light pulses at a light intensity of about 2.6 mW mm$^{-2}$.

## Quantification and data analysis

Error bars, exact *n* values and statistical tests are described in the figure legends. No statistical methods were used to predetermine sample sizes. Sample sizes were estimated on the basis of previous experiments performed in our group and are consistent with those generally used in the field. Animals were only excluded if viral transduction was unsuccessful or off-target or if the fibre, cannula or lens tip placement was off-target. For electrophysiological recordings, only cells with a stable series resistance of <30 MΩ were analysed. These criteria were determined before statistical tests were performed. The following experiments were replicated twice by different experimenters: switch to pup-directed aggression induced by 6 h of food deprivation and slice physiology recordings across the oestrous cycle. All attempts at replication were successful. Animals were randomly assigned to treatment and control groups. Experimental groups consisted of multiple cohorts to avoid litter and cage effects. Data acquisition was not performed blind. Behavioural data were scored by an individual blind to the experimental design, and analyses of behavioural, histological, electrophysiological and in vivo imaging data were conducted under blind conditions. Exact *P* values, *t* values, *F* values and degrees of freedom are provided in the source data.

## Calculation of behavioural transition probabilities

To calculate the behavioural transition probabilities shown in Extended Data Fig. 1, we first created temporally ordered lists of scored behaviours for individuals classified as Agg$^+$, parental Agg$^-$ or ignoring Agg$^-$. We filtered these lists to include only relevant behaviours, then parsed them into sequential behaviour pairs (for example, behaviour 1→behaviour 2). For each unique pair, we calculated the transition probability by dividing the number of occurrences of that pair by the total number of transitions originating from behaviour 1. This produced a behavioural transition matrix for each individual mouse, whereby each entry represents the conditional probability of transitioning from one behaviour to another. Rows were normalized such that each value reflects the likelihood of transitioning to a new state given the current behaviour. For visualization, we constructed directed graphs in which nodes represent individual behaviours, arrows denote transitions and arrow thickness corresponds to the transition probability.

## Calculation of predicted baseline switching rates

The observed switching rates of animals with hormone receptor ablation were compared with the predicted baseline switching rate, which would be expected for each cohort if receptors were intact. These baseline rates were determined using hypothesis testing on Poisson binomial distributions, which were constructed on the basis of the oestrous cycle distribution of each cohort using the poibin package (https://github.com/tsakim/poibin). The predicted switching rate corresponds to the mean of each custom Poisson binomial distribution.

## Image analysis and registration

The ImageJ plugin ABBA[76] was used to register coronal brain sections to the Allen Brain Atlas (CCFv3)[77]. In brief, *x* and *y* rotations were adjusted across all sections from a given brain, and two rounds of affine registration using Elastix were performed. Samples then underwent non-rigid registration using the BigWarp tool (sample channel: DAPI; atlas channel: Nissl). Positive cell detection was performed on the transformed samples using QuPath, followed by subcellular detection of *Hcn* transcript spots and clusters. Spot counts in clusters were estimated by dividing the cluster area by the expected size of individual spots. Transformed cell detections were exported from QuPath, visualized using a custom Python app (https://github.com/nickdelgrosso/ABBA-QuPath-RegistrationAnalysis) and analysed using custom scripts in Python (v.3.7).

## Quantification of *Npy* and *Agrp* knockdown efficiency

Brain sections from *Agrp-cre* mice injected with conditional AAVs expressing GFP and shRNA targeting either *Npy* or *Agrp* (or a negative control, see the section 'Gene knockdown') were immunostained for NPY or AgRP, respectively (see the section 'Immunohistochemistry') and imaged using a Zeiss LSM 710 confocal microscope. Quantification was performed using pixel-based analysis, as NPY and AgRP immunoreactivity was primarily localized to fibres rather than cell bodies. Image stacks were imported into ImageJ, and the JaCoP plugin[78] was used to calculate the percentage of pixels in the NPY or AgRP channels that colocalized with GFP-positive pixels.

## Coexpression analysis

To assess coexpression of *Hcn* subunits (*Hcn1* and *Hcn2*), *Npy* receptor genes (*Npy1r* and *Npy2r*) and *Esr1* and *Pgr* in MPOA neurons, we analysed a previously published single-cell RNA sequencing dataset[79]. We queried the adult hypothalamus dataset (WMB-10Xv3-HY-log2.h5ad) and filtered for neurons assigned to the MPOA. Coexpression was assessed by calculating the proportion of cells expressing each gene above a defined threshold (1 copy; Extended Data Fig. 7h), and the overlap across marker-defined neuronal clusters were examined.

## Processing and analysis of fibre photometry data

The recorded interleaved trace was separated into isosbestic (415 nm) and calcium-dependent (470 nm) channels using custom Python routines. To correct for motion artefacts and baseline drift, a linear fit of the 415 nm signal was computed and subtracted from the 470 nm signal. To further correct for slow fluctuations such as photobleaching, a moving minimum baseline (20-s sliding window) was subtracted from the resulting trace. The relative fluorescence change was then calculated as $\frac{\Delta F}{F_{mean}} = \frac{F - F_{mean}}{F_{mean}}$ and normalized using min–max scaling. Manually scored behaviours were aligned to the activity traces through timestamps acquired in Bonsai.

## Processing and analysis of in vivo imaging data

**Pre-processing.** Image frames were spatially downsampled to 400 × 540 pixels. Drift of the baseline signal over time was removed using a spatial bandpass filter with lower and upper cut-off spatial frequencies of 0.005 and 0.5 oscillations per pixel, respectively. Motion artefact correction was performed, and the relative fluorescence change $\Delta F/F_0$ for each pixel compared with the baseline was calculated as $\frac{\Delta F}{F_0} = \frac{F - F_0}{F_0}$, where $F_0$ is the mean fluorescence value of each pixel during the baseline period). Cell detection based on princicpal component analysis (PCA) or independent component analysis was performed using a mean ROI radius of 7–9 pixels in Inscopix Data Processing software. All automatically identified cells were manually verified to exclude false-positive detections, and cells not detected by the algorithm were manually added. Cell traces were deconvolved using OASIS[80] with a model order of 1 and a spike SNR threshold of 3.0.

**Longitudinal registration.** Longitudinal registration of pre and post field-of-views was performed using Inscopix Data Processing software without session correlation for thresholding. The resulting aligned traces were manually quality-controlled. ROIs with irregular shapes or without activity transients were discarded.

**Evoked activity and absolute tuning index.** For population-averaged neural activity and absolute tuning indices, we analysed the first behavioural bout of each specified action per session. To reduce potential confounds from previous behaviour occurrences, or cumulative social experience, and to ensure that neural activity reflected the response to the behaviour of interest, we selected bouts in which no other overt behaviours occurred during the baseline period. This was straightforward for isolated chemoinvestigation events (for example, pup

or intruder sniffing), but more challenging for behaviours typically embedded in behavioural sequences, such as pup grooming and aggressive contact. These behaviours are often preceded by pup-directed sniffing or grooming, and only a very small number of episodes occurred in complete isolation. Grooming-related and aggression-related traces were therefore not excluded based on baseline contamination but were still limited to the first bout per session to minimize experience-dependent effects.

The absolute tuning index measures how strongly the activity for each detected cell deviates from baseline during a behavioural event, incorporating both positive and negative activity changes. This index accounts for variability by considering the standard deviation of both the baseline and activity period. The baseline and activity windows used for z score and tuning index calculations were adapted for each behaviour based on the average duration of behavioural bouts (that is, ±2 s for pup sniffing and attacks, ±4 s for pup grooming, ±5 s for male intruder sniffing, and ±3 s for female intruder sniffing). Tuning indexes were calculated on the basis of these behaviour-specific windows, using the pre-event period as baseline and the post-event period as the activity window. The z scores were calculated using ±5 s from behaviour onset as $z = x - \mu\sigma$, where $x = \Delta F/F$ of the current timestamp, $\mu$ is the mean $\Delta F/F$ of the baseline period and $\sigma$ is the standard deviation of the baseline period. Significant responses were called when the z scored $\Delta F/F$ of the baseline and activity periods were significantly different (using unpaired t-tests). Cells were thereafter categorized as exhibiting increased, decreased or unchanged evoked activity. The single neuron tuning index was derived from performing an unpaired t-test between the activity and baseline periods and represents the absolute t value. Only neurons exhibiting increased activity during behaviours were used for z score plots.

**PCA.** PCA was used to reduce the dimensionality of the neural data and to identify the primary sources of variance in each recorded pup interaction session. Before applying PCA, the activity of each recorded neuron was standardized to ensure comparability across different neurons. The standardized activity for each neuron in each session was computed as follows:

$$x_i' = \frac{x_i - \mu_i}{\sigma_i}$$

where $x_i$ represents the activity of the $i$-th neuron, $\mu_i$ is its mean activity across the entire dataset and $\sigma_i$ is its standard deviation. To quantify relationships between neurons, we computed the covariance matrix $C$ of the standardized data as follows:

$$C = \frac{1}{n-1} \sum_{i=1}^{n} (x_i - \bar{x})(x_i - \bar{x})^T$$

where $n$ is the total number of neurons and $\bar{x}$ is the mean activity vector across all neurons. The eigenvalues and eigenvectors of $C$ were then computed, with each eigenvector $v_j$ representing a PC and its corresponding eigenvalue $\lambda_j$ indicating the variance explained by that component. The PCs were ranked by their eigenvalues, with higher-ranked components capturing more variance. Recording sessions were included in the analysis only if the first two PCs accounted for at least 70% of the total variance. These two PCs were then used to project the neural population activity into a lower-dimensional space for visualization.

**PC distance calculation.** For the calculation of PC distances, activity episodes of 5 s after behavioural onset were extracted from each neuron. All episodes were standardized before analysis. To quantify the similarity between two neural activity episodes, $a$ and $b$, in the $k$-dimensional PC space ($k = 2$), we computed the pointwise Euclidean distance at each time point $t$ as follows:

$$d(a_t, b_t) = \sqrt{\sum_{i=1}^{k} (a_{t,i} - b_{t,i})^2}$$

where $a_t = (a_{t,1}, a_{t,2}, ..., a_{t,k})$ and $b_t = (b_{t,1}, b_{t,2}, ..., bt_k)$ are the projections of the episodes onto the reduced $k$-dimensional PC space at time $t$, $a_{t,1}$ and $b_{t,1}$ represent the $i$-th PC of each episode at time $t$, and $k$ is the number of retained PCs (chosen to account for 90% of the variance).

To obtain the total distance between the two episodes, the pointwise Euclidean distances were summed across all time points as follows:

$$D(a,b) = \sqrt{\sum_{t=1}^{T} \sum_{i=1}^{k} (a_{t,i} - b_{t,i})^2}$$

where $T$ is the total number of time points in the episode. This total distance, $D(a,b)$, serves as a similarity measure of neural population activity between conditions (for example, pre and post), with larger values indicating greater divergence in activity patterns.

**Multiclass SVM classification.** A SVM classifier was used to categorize behavioural states (pup sniff, pup groom, lick aggressive and late aggression) based on the following data: (1) raw neural data, (2) PCA-reduced neural data and (3) shuffled neural data (control). All behavioural episodes of >2 s were selected. Behavioural labels were assigned unique numeric identifiers using LabelEncoder, whereas the neural activity data served as the feature matrix ($X$) and the behavioural labels as the target variable ($y$). To address class imbalances, the synthetic minority oversampling technique (SMOTE) was applied. The $k$-neighbour parameter was dynamically adjusted based on the size of the smallest class. If a class contained fewer than two samples, SMOTE was not applied and the dataset was excluded from the analysis. Classifier performance was evaluated over 50 iterations. The dataset was split into training and test sets (split by episode number rather than by frame number) using stratification to preserve class distributions. To avoid data leakage, splits were made based on episode numbers rather than frame numbers. The SVM classifier was implemented using SVC from scikit-learn with default parameters. After training, predictions were generated for the test set, and accuracy was computed as follows:

$$\text{Accuracy} = \frac{\sum_{i=1}^{n} \mathbb{I}(y_{\text{pred},i} = y_{\text{test},i})}{n}$$

where $y_{\text{pred}}$ and $y_{\text{test}}$ are the predicted and true labels, respectively, $\mathbb{I}(\cdot)$ is the indicator function, which returns 1 if the predicted and true labels match, and 0 otherwise, and $n$ is the total number of test samples.

The accuracy of the SVM trained on raw neural data, PCA-reduced neural data and shuffled data was then compared. To assess classification performance across behavioural states, a confusion matrix was computed. The average accuracy and average confusion matrix were obtained by summing the confusion matrices over all iterations and then normalizing them row-wise as percentages as follows:

$$\text{Normalized CM}_{ij} = \frac{\text{CM}_{ij}}{\sum_{j=1}^{m} \text{CM}_{ij}} \times 100$$

where $\text{CM}_{ij}$ represents the number of times a sample from class $i$ was classified as class $j$, and $m$ is the total number of classes.

**HMM analysis.** To determine the optimal number of states for the HMM, we first explored a range of state values and assessed model performance using the evidence lower bound (ELBO). ELBO, a standard metric in variational inference, measures model fit, with higher values indicating better performance. The optimal state number was identified by locating the turning point in the ELBO curve, beyond which additional states provided diminishing improvements.

This optimal value (here 5; Extended Data Fig. 9v) was then used for all subsequent analyses. The HMM was implemented using the ssm package (https://github.com/lindermanlab/ssm) with a Gaussian observation model. This framework applies Bayesian learning and inference to state-space models and is well suited for analysing sequential neural data[9]. The HMM was trained on neural recordings using the expectation-maximization algorithm for 50 iterations, with initial state assignments determined using $k$-means clustering. To assess the relationship between neural states and behaviour, we constructed a binary matrix for each behaviour, marking timestamps where the behaviour occurred as 1 and all other timestamps as 0. We then identified the state most frequently associated with each behaviour and computed the conditional probability of that behaviour occurring in the inferred state. This probability quantifies how strongly a given neural state is linked to specific behavioural patterns.

**Selectivity analysis.** To assess the selectivity of individual neurons for a particular stimulus, social (pups, male intruder or female intruder) and non-social stimuli (Lego brick, bedding or food) were presented sequentially in randomized order. Selectivity for pups versus other stimuli was quantified using a choice probability approach[4,81]. For each neuron, fluorescence signals ($\Delta F/F_0$) recorded during pairs of chemosensory investigation behaviours (for example, pup versus intruder investigation) were used to estimate how reliably the two behaviours could be distinguished based on their $\Delta F/F_0$ distributions. Specifically, $\Delta F/F_0$ values were extracted for each neuron's activity during behaviour $\alpha$ (for example, attack) and behaviour $\beta$ (for example, sniff). These distributions were plotted as paired histograms and cumulative distribution functions (CDFs). A ROC curve was then generated, with the CDF of pup sniffing on the $x$ axis. The selectivity index was computed as: Selectivity index $= 1 - \text{AUC}_{\text{ROC}}$.

Here $\text{AUC}_{\text{ROC}}$ is the area under the ROC curve. Neurons exclusively active during pup sniffing have a selectivity index of 1, those exclusively active during investigation of another stimulus an index of 0, and non-selective neurons an index of 0.5. To minimize the effects of gradual desensitization, only the first chemosensory investigation episode for each stimulus was used.

For aggression-related analyses (Extended Data Fig. 9r), we adapted this approach to test for persistent neural encoding. Following the first pup-directed aggression episode, we extracted the $\Delta F/F_0$ values for each neuron during aggression periods and non-aggression periods. Selectivity indices were computed as above. To assess population-level structure, we analysed the distribution of selectivity indices across neurons: a normal distribution centred at 0.5 indicates persistent state encoding, whereas skewed distributions suggest selective responses to discrete behavioural events. For comparison, we applied the same analysis to pup sniffing and pup grooming behaviours using activity recorded before the first aggression episode. Selectivity distributions for sniffing and grooming were significantly skewed, which suggested event-related encoding by discrete neuronal subpopulations.

### Reporting summary

Further information on research design is available in the Nature Portfolio Reporting Summary linked to this article.

## Data availability

The data that support the findings of this study are available from the corresponding author upon request. The previously published adult hypothalamus single-cell RNA sequencing dataset[79] (WMB-10Xv3-HY-log2.h5ad) was downloaded from https://allen-brain-cell-atlas.s3.us-west-2.amazonaws.com/index.html#expression_matrices/WMB-10Xv3/20230630/. Source data are provided with this paper.

## Code availability

Code created for this study is available from GitHub (https://github.com/FrancisCrickInstitute/negative_parental_switch).

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

**Acknowledgements** We thank A. Schaefer, V. Stempel and members of the State-dependent Neural Processing Laboratory for discussions and comments on the manuscript; staff at the Biological Research Facility at the Francis Crick Institute for animal care and technical assistance; staff at the Crick Light Microscopy, Experimental Histopathology, Mass Spectrometry and Bioinformatics Science Technology Platforms; staff at the Making Lab, Mechanical Workshop and Vector Core; A. Strosche and V. Stempel for advice on cannulation; H. Fenselau for sharing the hGlyAG construct; S. Wood and A. Resasco for assistance with surgeries; and N. Borak for sharing implanted mice for miniature microscopy recordings. This study received support from the Francis Crick Institute (core funding FC001153). The Francis Crick Institute receives its funding from Cancer Research UK, the UK Medical Research Council and the Wellcome Trust (to J.K.). This study was also supported by a European Research Council starting grant (ERC-2019-STG847873 to J.K.), a NARSAD Young Investigator Award (BB/V016946/1 to J.K.) and a Boehringer Ingelheim PhD fellowship (to M.C.).

**Author contributions** M.C., R.A. and J.K. conceptualized the study. M.C., R.A., M.X.C., P.W., B.F.A.H. and A.S. performed behavioural experiments. M.C., P.W. and B.F.A.H. performed stereotaxic surgeries. R.A. and B.B.J. performed and analysed slice electrophysiology recordings. M.C. and S.L. performed histology and antibody staining. P.W. performed RNAscope single-molecule fluorescence in situ hybridization experiments. M.C., M.X.C. and A.S. performed optogenetic stimulation experiments. M.C. and I.S. performed fibre photometry and miniature microscopy recordings. N.L. and J.M. performed mass spectrometry data acquisition and analysis. M.S. designed and prepared custom viruses. M.C. and J.K. analysed behavioural and in vivo imaging data. M.C. and J.K. acquired funding. M.C. and J.K. supervised the project. M.C. and J.K. prepared the figures. M.C. and J.K. wrote the original draft of the manuscript. M.C., R.A. and J.K. reviewed and edited the manuscript.

**Funding** Open Access funding provided by The Francis Crick Institute.

**Competing interests** The authors declare no competing interests.

**Additional information**
**Correspondence and requests for materials** should be addressed to Johannes Kohl.

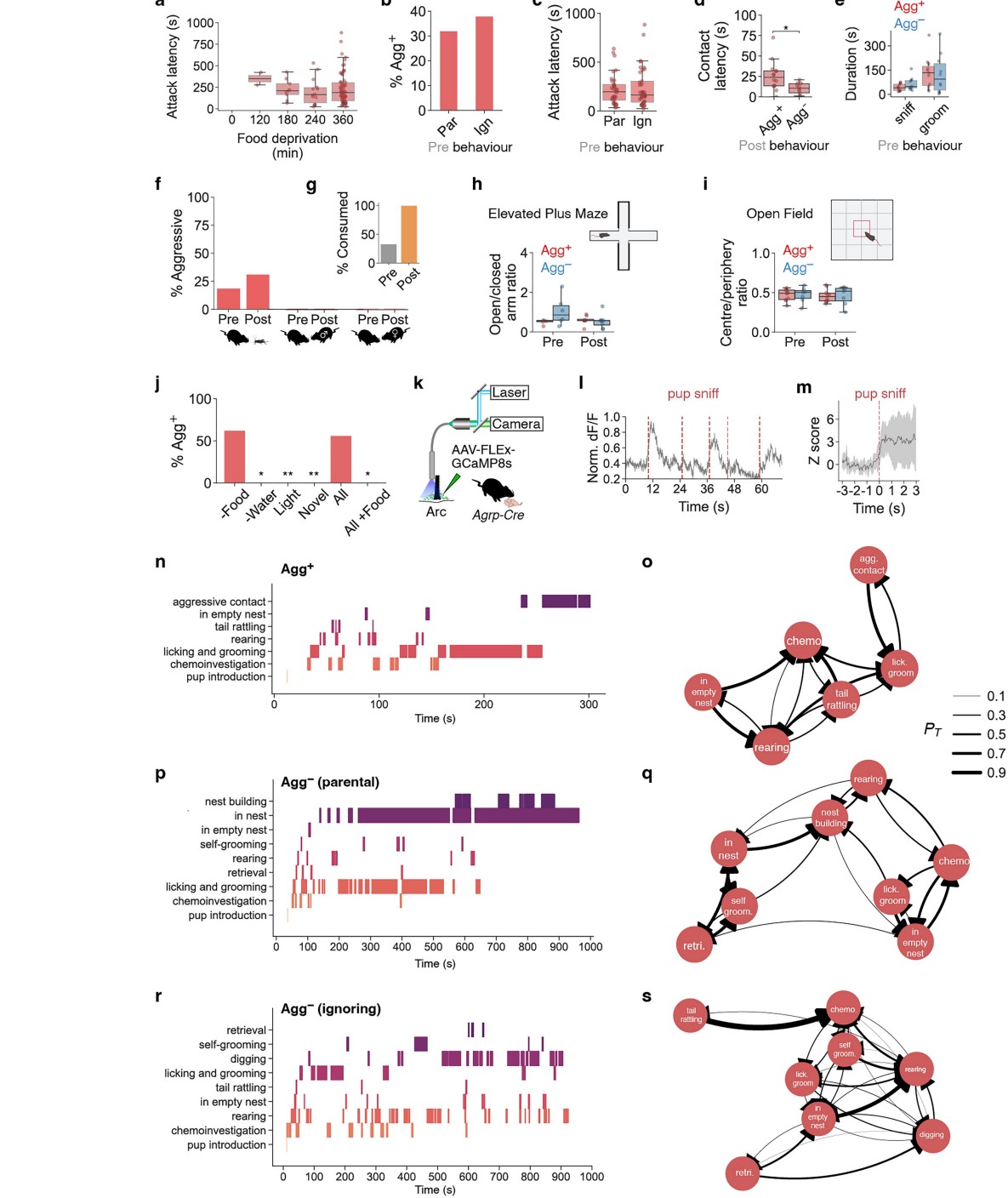

**Extended Data Fig. 1** | See next page for caption.

**Extended Data Fig. 1 | State- and target-specificity of the negative parental switch. a**, Attack latency as a function of food restriction duration ($n$ (left to right) = 2, 9, 14 and 70; $P = 6.5 \times 10^{-20}$). **b**, Percentage of Agg$^+$ animals depending on behaviour before food deprivation (Par, parental, Ign, ignoring; $n = 67$ (Par), 66 (Ign) mice). **c**, Attack latency depending on behaviour before food deprivation ($n = 33$ (Par), 37 (Ign)). **d**, Pup contact latency of Agg$^+$ and Agg$^-$ mice after food deprivation ($n = 14$ (Agg$^+$), 10 (Agg$^-$)). **e**, Duration of pup sniffing and pup grooming in Agg$^+$ and Agg$^-$ mice (sniff: $n = 12$ (Agg$^+$), 10 (Agg$^-$); groom: $n = 11$ (Agg$^+$), 10 (Agg$^-$)). **f**, Percentage of mice exhibiting aggression towards prey (cricket) or adult intruders in Pre and Post period ($n$ (left to right) = 16, 16 and 16 mice). **g**, Percentage of mice that consume prey after initiating hunting behaviour ($n = 6$ (Pre), 5 (Post)). **h**, Performance of Agg$^+$ and Agg$^-$ mice in Elevated Plus Maze before and after food deprivation ($n = 6$ (Pre), 8 (Post)). **i**, Performance of Agg$^+$ and Agg$^-$ mice in Open Field before and after food deprivation ($n = 7$ (Pre), 10 (Post)). **j**, Percentage of mice switching to pup-directed aggression after 6 h of either food restriction, water restriction, light cycle inversion, or housing in novel, empty cage. All, all stressors, All+Food, all stressors in non-food-deprived mice ($n$ (left to right) = 16, 8, 11, 10, 16 and 8 mice). Significance levels are between '–Food' and all other groups. **k**, Fibre photometry recordings from Arc$^{AgRP}$ neurons during pup interactions. **l**, Example recording trace of Arc$^{AgRP}$ population activity during pup chemoinvestigation in 6-h food-deprived mice. **m**, Averaged, Z-scored Arc$^{AgRP}$ activity during pup chemoinvestigation ($n = 5$ traces, $N = 1$ mouse). Data are mean ± s.e.m. **n–s**, Representative behavioural raster plots (**n,p,r**) and corresponding behavioural state transition diagrams (**o,q,s**; see Methods) from individual mice classified as Agg$^+$, parental Agg$^-$, or pup-ignoring Agg$^-$ during pup interactions. Chemo, chemoinvestigation, retri., pup retrieval, in nest: in nest with pups, in empty nest: in next without pups, $P_T$, transition probability. Statistics: One-way ANOVA in **a**. Chi-Square test (two-sided) in **b,g**. Fisher's exact test in **f,j** (two-sided, Benjamini-Hochberg adjustment in **j**). $U$ test (two-sided) in **c,d**. Two-way ANOVA in **e,h,i**. Box plots: median (line), interquartile range (box), whiskers, 1.5× IQR. *$P < 0.05$, **$P < 0.01$. See Supplementary Table 3 for further details of the statistical analyses.

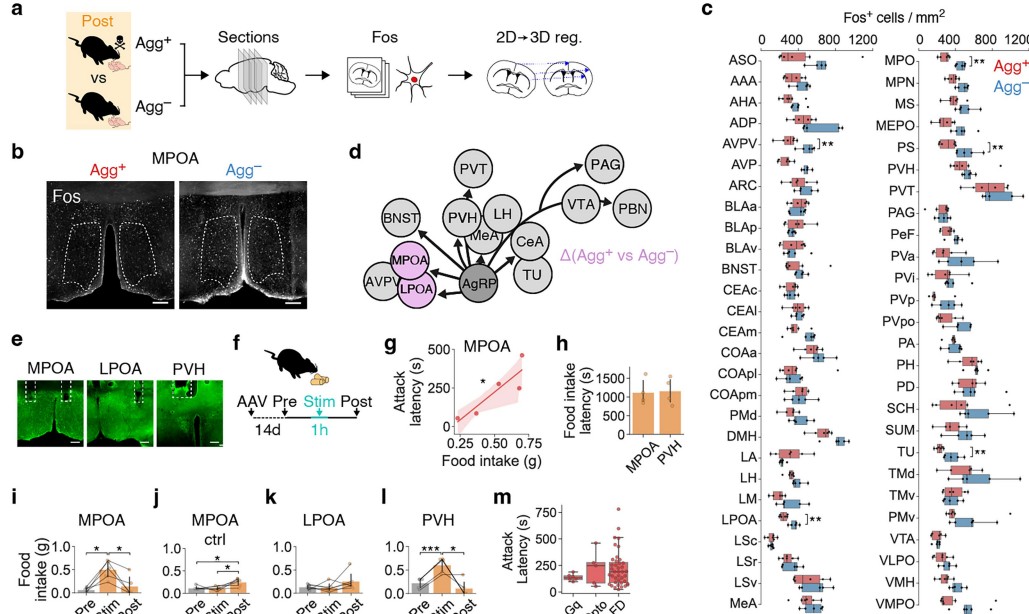

**Extended Data Fig. 2 | Identification of candidate Arc^AgRP targets and optogenetic activation of projections. a**, Immunostainings against c-Fos in brain sections from Agg⁺ and Agg⁻ mice, and registration to Allen Brain Atlas (see Methods). **b**, Example FOS⁺ cell densities in MPOA of Agg⁺ and Agg⁻ mice in MPOA sections. Scale bars, 200 μm. **c**, Density of FOS⁺ cells in hypothalamic brain areas of Agg⁺ and Agg⁻ mice (*n* = 6 (Agg⁺), 5 (Agg⁻) mice). **d**, Arc^AgRP projections. Areas with significantly different FOS⁺ cell numbers between Agg⁺ and Agg⁻ groups, and receiving direct Arc^AgRP projections, are highlighted. Note that ADP, MPN, MPO, PD and PS are MPOA subregions. **e**, Implantation sites of optical fibres in MPOA, LPOA and PVH of *Agrp-Cre* mice injected with AAV-DIO-ChR2-EYFP. Scale bars, 200 μm. **f**, Optogenetic stimulation paradigm to assess food intake. **g**, Correlation between food intake and pup attack latency in mice with optogenetic stimulation of Arc^AgRP→MPOA projections (linear regression, R² = 0.784; *P* = 0.046, *n* = 5 mice). **h**, Latency of food intake during continuous stimulation of Arc^AgRP→MPOA and Arc^AgRP→PVH projections (*n* = 4 (MPOA),

4 (PVH) mice). **i,j**, Effect of optogenetically activating Arc^AgRP→MPOA projections (**i**, 1-h pre-stim, *n* = 5 mice) on 1-h food consumption in sated mice, and negative controls (**j**, *Agrp-Cre* mice injected with AAV-DIO-EYFP, *n* = 6 mice). **k,l**, Effect of optogenetically activating Arc^AgRP→LPOA (**k**, *n* = 6 mice) or Arc^AgRP→PVH (**l**, *n* = 5 mice) projections on 1-h food consumption in sated mice. **m**, Pup attack latency after chemogenetic activation of Arc^AgRP neurons (Gq), optogenetic stimulation of Arc^AgRP→MPOA projections (Opto, 30-min pre stimulation), and food deprivation (*n* (left to right) = 7, 5 and 39 mice). Statistics: *U* test in **c,h** (two-sided, Benjamini-Hochberg adjustment in **c**). Repeated measures ANOVA in **i–m**, with pairwise *t*-tests and Benjamini-Hochberg adjustment in **i,j,l**. Data are mean ± s.e.m. Box plots: median (line), interquartile range (box), whiskers, 1.5× IQR. See Supplementary Table 1 for acronyms of brain areas. *P < 0.05, **P < 0.01, ***P < 0.001. See Supplementary Table 3 for further details of the statistical analyses.

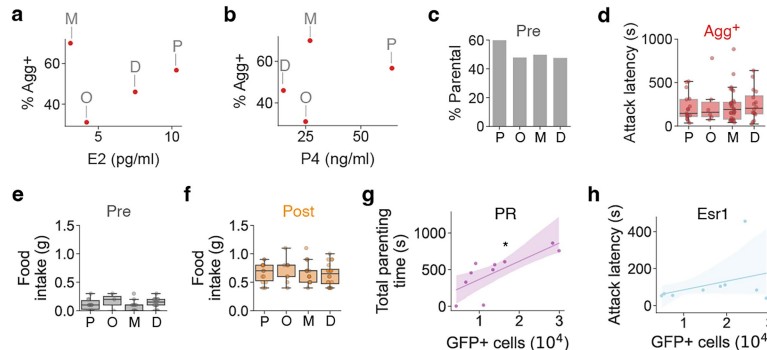

**Extended Data Fig. 3 | Effects of oestrous state on parental interactions and food intake. a, b,** Switching rate does not correlate with P4 (**a**) or E2 (**b**) plasma concentration (P, O, M, D; $n$ = 30, 19, 40 and 37 mice). **c,** Percentage of spontaneously parental virgin female mice in different estrous states before food deprivation ($n$ (left to right) = 35, 26, 52 and 45 mice). **d,** Attack latency of Agg⁺ mice in different estrous states ($n$ (left to right) = 17, 6, 29 and 16 mice). **e,f,** 1-h food consumption in sated mice before (**e**) and after (**f**) food deprivation depending on estrous state (Pre: $n$ (left to right) = 10, 3, 11 and 14; Post: $n$ (left to right) = 14, 9, 17 and 20). **g,** Correlation between total parenting time (see

Methods) and number of GFP-labelled MPOA neurons in $Pgr^{loxP}$ mice injected with an AAV co-expressing GFP and Cre (linear regression, $R^2$ = 0.581; $P$ = 0.01, $n$ = 10). **h,** Correlation between attack latency and number of GFP-labelled MPOA neurons in $Esr1^{loxP}$ mice injected with an AAV co-expressing GFP and Cre (linear regression, $R^2$ = 0.113; $P$ = 0.377, $n$ = 9). P, proestrus, O, oestrus, M, metestrus, D, diestrus. Statistics: Fisher's exact test in **c** (two-sided, Benjamini-Hochberg adjustment). One-way ANOVA in **d–f**. Data are mean ± s.e.m. Box plots: median (line), interquartile range (box), whiskers, 1.5× IQR. *$P$ < 0.05. See Supplementary Table 3 for further details of the statistical analyses.

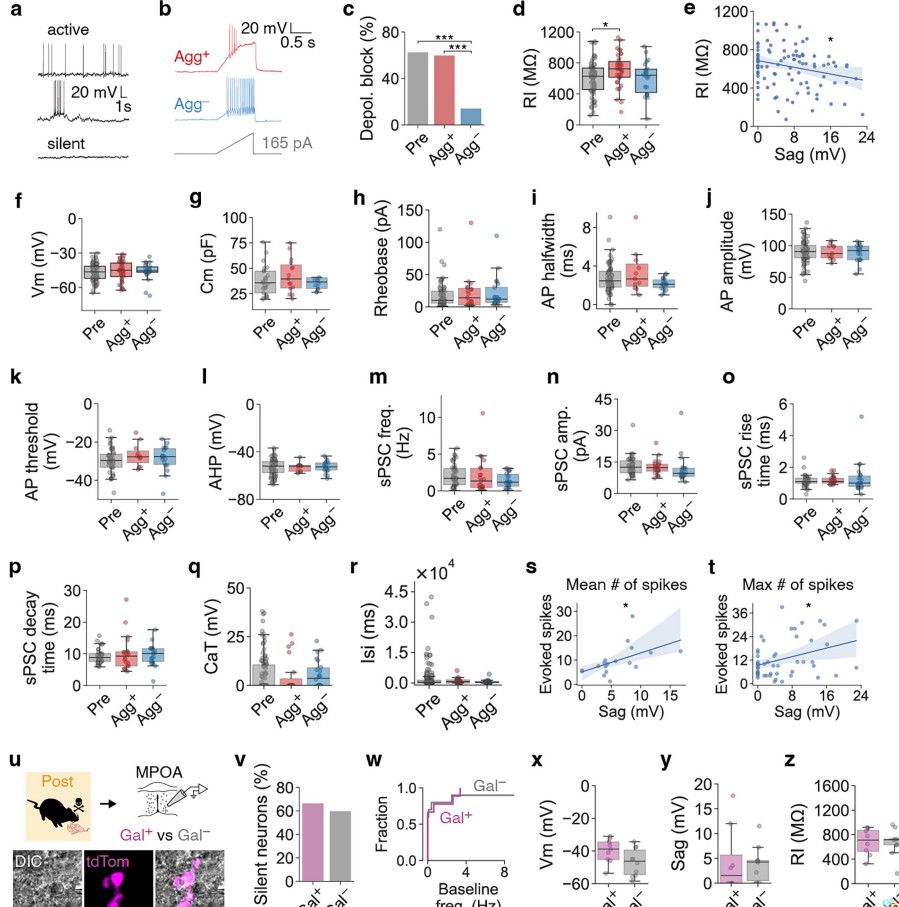

**Extended Data Fig. 4 | Biophysical effects of food deprivation on MPOA neurons. a**, Example current-clamp traces of different activity patterns of MPOA neurons at resting potential. **b**, Example current clamp recording traces of cells with (Agg⁺) and without (Agg⁻) depolarization block. **c,d,f–r,** Whole-cell recordings from MPOA neurons in mice before (Pre) and after (Agg⁺ and Agg⁻) 6-h food deprivation): percentage of neurons exhibiting depolarization block (**c**, *n* (left to right) = 62, 20 and 21 cells, from *N* (left to right) = 28, 6 and 3 mice), input resistance (**d**, *n* (left to right) = 103, 19 and 22 cells, *N* (left to right) = 23, 6 and 3 mice), resting membrane potential (**f**, *n* (left to right) = 125, 20 and 22 cells, *N* (left to right) = 24, 6 and 3 mice), membrane capacitance (**g**, *n* (left to right) = 25, 15 and 14 cells, *N* (left to right) = 10, 5 and 2 mice), rheobase (**h**, *n* (left to right) = 72, 19 and 21 cells, *N* (left to right) = 17, 6 and 3 mice), action potential half-width (**i**, *n* (left to right) = 74, 11 and 18 cells, *N* (left to right) = 22, 6 and 3 mice), action potential amplitude (**j**, *n* (left to right) = 76, 11 and 18 cells, *N* (left to right) = 22, 6 and 3 mice), action potential threshold (**k**, *n* (left to right) = 45, 18 and 11 cells, *N* (left to right) = 36, 6 and 3 mice), afterhyperpolarization (**l**, *n* (left to right) = 75, 11 and 18 cells, *N* (left to right) = 22, 6 and 3 mice), sPSC frequency (**m**, *n* (left to right) = 51, 22, and 21 cells, *N* (left to right) = 23, 6 and 3 mice), sPSC amplitude (**n**, *n* (left to right) = 50, 21 and 22 cells, *N* (left to right) = 23, 6 and 3 mice), sPSC rise time (**o**, *n* (left to right) = 47, 21 and 22 cells,

*N* (left to right) = 21, 6 and 3 mice), sPSC decay time (**p**, *n* (left to right) = 41, 21 and 21 cells, *N* (left to right) = 17, 6 and 3 mice), T-type calcium channel-mediated rebound depolarisation (**q**, *n* (left to right) = 102, 19 and 22 cells, *N* = 22, 6 and 3 mice), and inter-spike interval (**r**, *n* (left to right) = 91, 11 and 18 cells, *N* (left to right) = 22, 6 and 3 mice). **e**, Correlation between input resistance and voltage sag amplitude (linear regression, $R^2$ = 0.045; *P* = 0.0185, *n* = 102 cells). **s,t**, Correlation between mean voltage sag amplitude and mean number of evoked action potentials per animal (**s**, linear regression, $R^2$ = 0.272; *P* = 0.026, *n* = 18), and correlation between voltage sag amplitude and number of evoked action potentials at the maximal injected current (**t**, *P* = 0.011, mixed linear model with mouse ID as random effect, *n* = 50 cells, *N* = 18 mice). **u**, Whole-cell recordings from Galanin-positive (Gal⁺) and -negative (Gal⁻) MPOA neurons in Agg⁺ mice (*n* = 11, 10 cells, *N* = 3, 4 mice). Scale bar, 20 μm. **v–z**, Percentage of silent neurons at resting potential (**v**), baseline firing frequency (**w**), resting membrane potential (**x**), voltage sag amplitude (**y**) and input resistance (**z**). Statistics: Chi-Square test in **c** (two-sided, Benjamini-Hochberg adjustment). One-way ANOVA with Tukey *post hoc* test in **d,f–r**. Fisher's exact test (two-sided) in **v**, *U* test (two-sided) in **w–z**. Box plots: median (line), interquartile range (box), whiskers, 1.5× IQR. **P* < 0.05, ***P* < 0.01. See Supplementary Table 3 for further details of the statistical analyses.

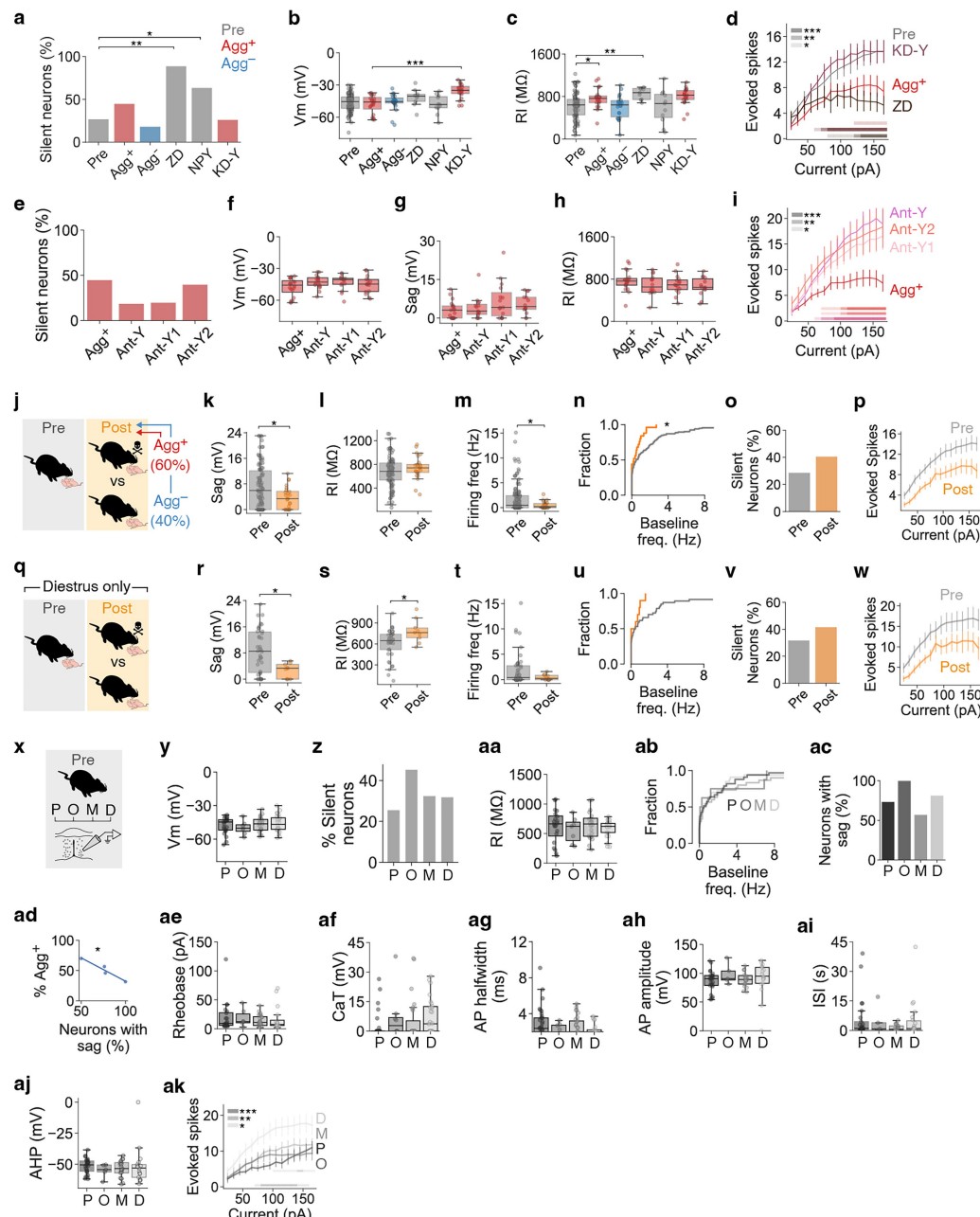

**Extended Data Fig. 5** | See next page for caption.

**Extended Data Fig. 5 | Biophysical effects of NPY, HCN channel function and oestrous state on MPOA neurons. a–c**, Biophysical parameters of MPOA neurons before and after 6-h food deprivation: silent neurons at resting potential (**a**, $n$ (left to right) = 126, 20, 22, 9, 11 and 19 cells, $N$ (left to right) = 24, 6, 3, 1, 4 and 3 mice), resting potential (**b**, $n$ (left to right) = 125, 20, 22, 9, 11 and 19 cells, $N$ (left to right) = 24, 6, 3, 1, 4 and 3 mice), and input resistance (**c**, $n$ (left to right) = 103, 19, 22, 9, 11 and 19 cells, $N$ (left to right) = 23, 6, 3, 1, 4 and 3 mice). **d**, Action potentials per injected current (Pre, Agg$^+$, ZD, KD-Y; $n$ = 36, 22, 11 and 17 cells, $N$ = 17, 7, 3 and 3 mice). **e–i**, Biophysical parameters in Agg$^+$ mice after combined (Ant-Y) and separate administration of Y1 and Y2 receptor antagonists (Agg$^+$, Ant-Y, Ant-Y1, Ant-Y2): silent neurons at resting potential (**e**, $n$ (left to right) = 20, 16, 15 and 15 cells, $N$ (left to right) = 6, 3, 2 and 3 mice), resting potential (**f**, $n$ (left to right) = 20, 16, 15 and 15 cells, $N$ (left to right) = 6, 3, 2 and 3 mice), voltage sag amplitude (**g**, $n$ (left to right) = 19, 15, 15 and 14 cells, $N$ (left to right) = 6, 3, 2 and 3 mice), input resistance (**h**, $n$ (left to right) = 19, 14, 15 and 14 cells, $N$ (left to right) = 6, 3, 2 and 3 mice) and action potentials per injected current (**i**, Agg$^+$, Ant-Y, Ant-Y1, Ant-Y2; $n$ = 10, 15 and 14 cells, $N$ (left to right) = 7, 3, 2 and 3 mice). **j–w**, MPOA neuronal properties between a Pre group (sampled to match oestrous cycle distribution of Post group) and a weighted Post group (60% Agg$^+$, 40% Agg$^-$; **j–p**), and between Pre and Post groups restricted to mice in dioestrus (**q–w**). Voltage sag amplitude (**k, r**; $n$ = 108 (Pre), 24 (Post) cells, $N$ = 39 (Pre), 9 (Post) mice; and $n$ = 38 (Pre), 9 (Post) cells, $N$ = 11 (Pre), 3 (Post) mice), input resistance (**l, s**; $n$ = 109 (Pre), 24 (Post) cells, $N$ = 39 (Pre), 9 (Post) mice; and $n$ = 38 (Pre), 9 (Post) cells, $N$ = 11 (Pre), 3 (Post) mice), baseline firing frequency (**m, t** and **n, u**; $n$ = 132 (Pre), 25 (Post) cells, $N$ = 43 (Pre), 9 (Post) mice; and $n$ = 47 (Pre), 10 (Post) cells, $N$ = 12 (Pre), 3 (Post) mice), silent neurons at resting membrane potential (**o, v**; $n$ = 136 (Pre), 27 (Post) cells, $N$ = 43 (Pre), 9 (Post) mice; and $n$ = 47 (Pre), 12 (Post) cells, $N$ = 12 (Pre), 3 (Post) mice), number of action potentials evoked by injected somatic current (**p, w**; $n$ = 36 (Pre), 21 (Post) cells, $N$ = 19 (Pre), 8 (Post) mice; and $n$ = 14 (Pre), 9 (Post) cells, $N$ = 4 (Pre), 3 (Post) mice). **x–ak**, Recordings from MPOA neurons at different oestrous stages before food deprivation (**x**): resting potential (**y**, $n$ (left to right) = 33, 8, 30 and 32 cells, $N$ (left to right) = 10, 3, 8 and 5 mice), silent neurons at resting potential (**z**, $n$ (left to right) = 33, 8, 30 and 33 cells, $N$ (left to right) = 10, 3, 8 and 5 mice), input resistance (**aa**, $n$ (left to right) = 30, 8, 28 and 27 cells, $N$ (left to right) = 9, 3, 8 and 4 mice), baseline firing frequency (**ab**, $n$ = 33 (P), 8 (O), 30 (M) and 33 (D) cells, $N$ = 10 (P), 3 (O), 8 (M) and 5 (D) mice), neurons with voltage sag (**ac**, $n$ (left to right) = 30, 8, 28, 27 cells, $N$ (left to right) = 9, 3, 8 and 4 mice), correlation between percentage of neurons with voltage sag and switching rate (**ad**, linear regression, $R^2$ = 0.035; $P$ = 0.006; data points are oestrous stages), rheobase (**ae**, $n$ (left to right) = 20, 6, 24 and 16 cells, $N$ (left to right) = 6, 3, 7 and 3 mice), T-type calcium channel-mediated rebound depolarisation (**af**, $n$ (left to right) 29, 8, 28 and 27 cells, $N$ (left to right) = 9, 3, 8 and 4 mice), action potential half-width (**ag**, $n$ (left to right) = 25, 5, 21 and 16 cells, $N$ (left to right) = 9, 3, 6 and 5 mice), action potential amplitude (**ah**, $n$ (left to right) = 25, 5, 21 and 18 cells, $N$ (left to right) = 9, 3, 6 and 5 mice), inter-spike interval (**ai**, $n$ (left to right) = 25, 5, 21 and 23 cells, $N$ (left to right) = 9, 3, 6 and 5 mice), and afterhyperpolarisation (**aj**, $n$ (left to right) = 25, 5, 21 and 17 cells, $N$ (left to right) = 9, 3, 6 and 5 mice). **ak**, Action potentials per injected current ($n$ = 10 (P), 11 (O), 13 (M) and 16 (D) cells, $N$ = 3 (P), 5 (O), 6 (M) and 4 (D) mice). Statistics: Fisher's exact test in **a,e,o,v,z,ac** (two-sided, Benjamini-Hochberg in **a,e**). $U$ test (between Pre, Agg$^+$ and Agg$^-$; Pre, ZD and NPY; Agg$^+$ and KD-Y) in **b,c,k–n,r–u** (two-sided, Benjamini-Hochberg adjustment in **b,c**). Mixed linear model with cell ID as random effect in **d,i,p,w,ak**. Bars in **d,i,ak** indicate periods of significant difference (**d**, top to bottom: mauve, Agg$^+$ vs ZD; dark brown, ZD vs KD-Y; gray, ZD vs Pre; **i**, top to bottom: red, Agg$^+$ vs Ant-Y2; salmon, Agg$^+$ vs Ant-Y1; pink, Agg$^+$ vs Ant-Y; **ak**, top to bottom: D vs E; D vs P). One-way ANOVA in (between Pre, Agg$^+$ and Agg$^-$ in **b,c**) in **f,g,h,y,aa,ab,ae–aj**. Data are mean ± s.e.m. Box plots: median (line), interquartile range (box), whiskers, 1.5× IQR. *$P$ < 0.05, **$P$ < 0.01, ***$P$ < 0.001. See Supplementary Table 3 for further details of the statistical analyses.

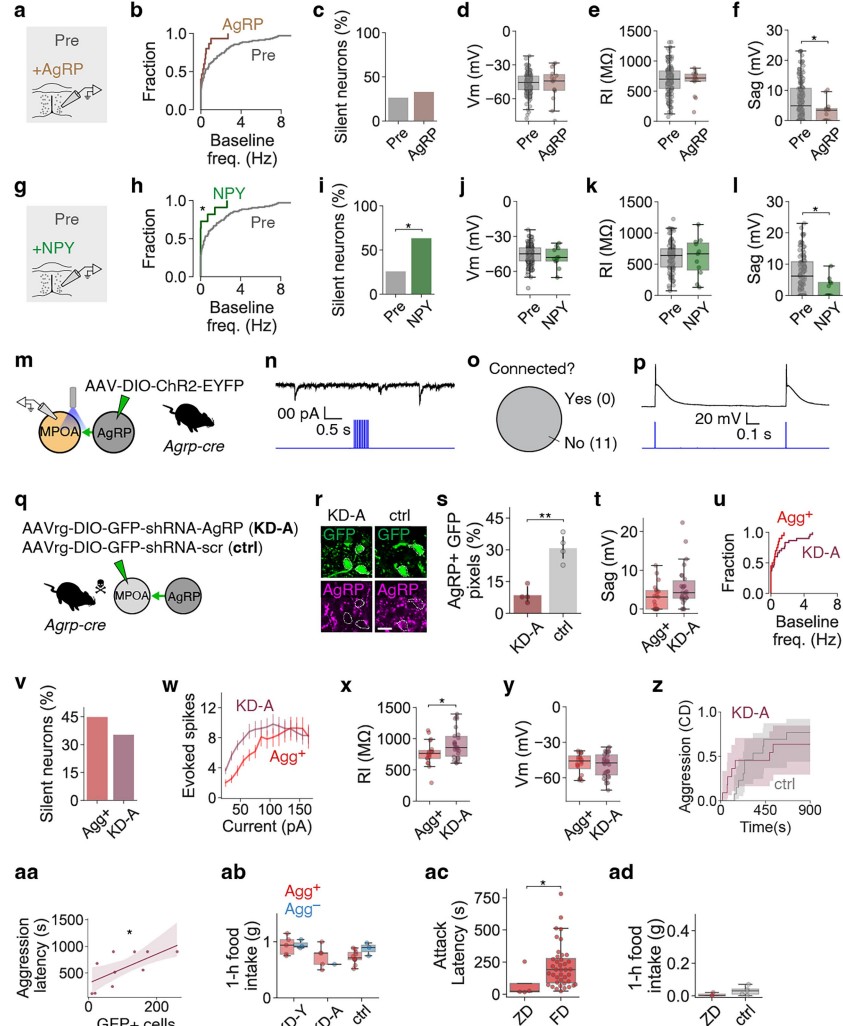

**Extended Data Fig. 6 | Arc^AgRP→MPOA neuromediator identification and behavioural effects of *Npy* knockdown and HCN blockade. a, g,** Recordings from MPOA neurons without (Pre) and with bath application of 100 µM AgRP (**a**) or 100 µM NPY (**g**). **b–f** and **g–l**, biophysical parameters of MPOA neurons at baseline (Pre) and with addition of AgRP or NPY: baseline firing frequency (**b**, *n* = 167 (Pre), 15 (AgRP) cells, *N* = 48 (Pre), 5 (AgRP) mice; **h**, *n* = 167 (Pre), 11 (NPY) cells, *N* = 48 (Pre), 4 (NPY) mice), percentage of silent neurons at resting membrane potential (**c**, *n* = 167 (Pre), 15 (AgRP) cells, *N* = 48 (Pre), 5 (AgRP) mice; **i**, *n* = 144 (Pre), 11 (NPY) cells, *N* = 48 (Pre), 4 (NPY) mice), resting membrane potential (**d**, *n* = 166 (Pre), 15 (AgRP) cells, *N* = 48 (Pre), 5 (AgRP) mice; **j**, *n* = 166 (Pre), 15 (AgRP) cells, *N* = 48 (Pre), 5 (AgRP) mice), input resistance (**e**, *n* = 142 (Pre), 15 (AgRP) cells, *N* = 46 (Pre), 5 (AgRP) mice; **k**, *n* = 142 (Pre), 11 (NPY) cells, *N* = 46 (Pre), 4 (NPY) mice), and voltage sag amplitude (**f**, *n* = 142 (Pre), 15 (AgRP) cells, *N* = 45 (Pre), 5 (AgRP) mice; **l**, *n* = 142 (Pre), 11 (NPY) cells, *N* = 45 (Pre), 4 (NPY) mice). **m**, Channelrhodopsin-assisted circuit mapping (CRACM) between Arc^AgRP and MPOA neurons. Whole-cell recordings from MPOA neurons, and 450 nm widefield stimulation of ChR2+ Arc^AgRP axons (see Methods). **n**, Example recording trace from MPOA neuron with sIPSCs. Note absence of light-evoked IPSCs in response to a train of 9 × 3-ms light pulses. **o**, Synaptic response pattern of MPOA neurons to acute Arc^AgRP terminal activation (*n* = 11 cells, *N* = 3 mice). **p**, Representative example of action potentials in a ChR2-positive Arc^AgRP neuron evoked by single 3-ms light pulses. **q**, *Agrp* knockdown (KD-A) in Arc^AgRP→MPOA projections, and scrambled control (ctrl). **r**, Example images of

*Agrp* KD and control. Scale bar, 20 µm. **s**, *Agrp* KD efficiency (*n* = 4 (KD-A), 4 (ctrl) mice). **t**, Voltage sag amplitude (*n* = 19 (Agg^+), 26 (KD-A) cells, *N* = 6 (Agg^+), 4 (KD-A) mice). **u**, Baseline firing (*n* = 20 (Agg^+), 30 (KD-A) cells, *N* = 6 (Agg^+), 4 (KD-A) mice). **v**, Percentage of silent neurons at resting membrane potential (*n* = 20 (Agg^+), 31 (KD-A) cells, *N* = 6 (Agg^+), 4 (KD-A) mice). **w**, Action potentials per injected current (*n* = 17 (Agg^+), 16 (KD-A) cells, *N* = 6 (Agg^+), 3 (KD-A) mice). **x**, Input resistance (*n* = 19 (Agg^+), 26 (KD-A) cells, *N* = 6 (Agg^+), 4 (KD-A) mice). **y**, Resting membrane potential (*n* = 20 (Agg^+), 30 (KD-A) cells, *N* = 6 (Agg^+), 4 (KD-A) mice). **z**, Cumulative incidence of aggression (*n* = 11 (KD-A), 13 (ctrl) mice). Shaded areas are confidence intervals. **aa**, Correlation between aggression latency and number of Arc^AgRP→MPOA neurons transduced with KD-Y construct (linear regression, R² = 0.450; *P* = 0.034; *n* = 10 mice). **ab**, 1-h food intake of animals with knockdown of *Npy* (KD-Y) or *Agrp* (KD-A) in Arc^AgRP→MPOA projections, and control (*n* (left to right) = 8, 5, 13 mice). **ac**, Attack latency of animals with bilateral infusion of HCN blocker into MPOA (ZD, *n* = 4 mice) or after food deprivation (FD, *n* = 39 mice). **ad**, 1-h food intake of animals with bilateral infusion of ZD or vehicle (ctrl) into MPOA (*n* = 3 (ZD), 5 (ctrl) mice). Statistics: *U* test (two-sided) in **b,d–f,h,j–l,t–u,x,y,ab,ac,ad**. Fisher's exact test (two-sided) in **c,i,v**. Unpaired *t*-test (two-sided) in **s**. Mixed linear model with cell ID as random effect in **w**. Log-rank test (one-sided) in **z**. Data are mean ± s.e.m. Box plots: median (line), interquartile range (box), whiskers, 1.5× IQR. Shaded areas in **z** represent 95% CI. *P < 0.05, **P < 0.01. See Supplementary Table 3 for further details of the statistical analyses.

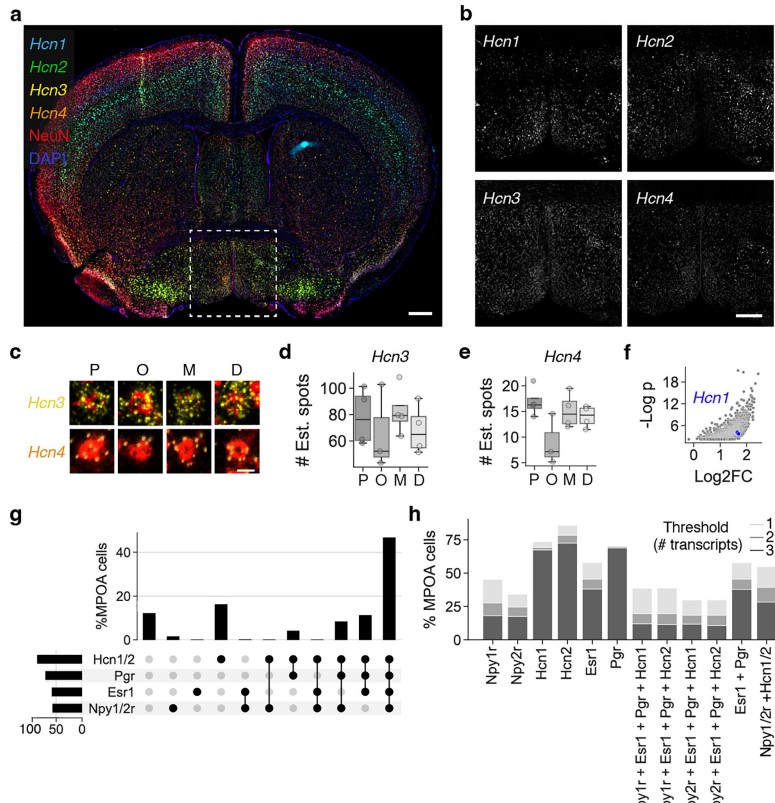

**Extended Data Fig. 7 | Hcn expression in the MPOA and co-expression with Pgr, Esr1 and Npyr. a**, Example coronal brain section with *Hcn* transcripts, after multiplexed in situ hybridisation and counterstaining with NeuN and DAPI. Scale bar, 500 μm. **b**, *Hcn* subunit expression in the MPOA. Scale bar, 300 μm. **c**–**e**, *Hcn3* and *Hcn4* mRNA expression across estrous cycle (**c**, red, NeuN counterstain; scale bars, 10 μm) and quantification (**d,e**, estimated number of spots, see Methods; *n* (left to right) = 4, 3, 4 and 4 mice). **f**, RNA-seq of genes upregulated by E2 treatment in *Esr1*⁺ neurons across the mouse brain (data from[37]). p, *P* value. FC, fold change. **g,h**, Co-expression of indicated transcripts in MPOA neurons (**g**), and percentage of co-expression in MPOA neurons depending on detection threshold (**h**). Based on data from[79] (see Methods). Statistics: One-way ANOVA with Tukey *post hoc* test in **d,e**. Box plots: median (line), interquartile range (box), whiskers, 1.5× IQR. *$P < 0.05$. See Supplementary Table 3 for further details of the statistical analyses.

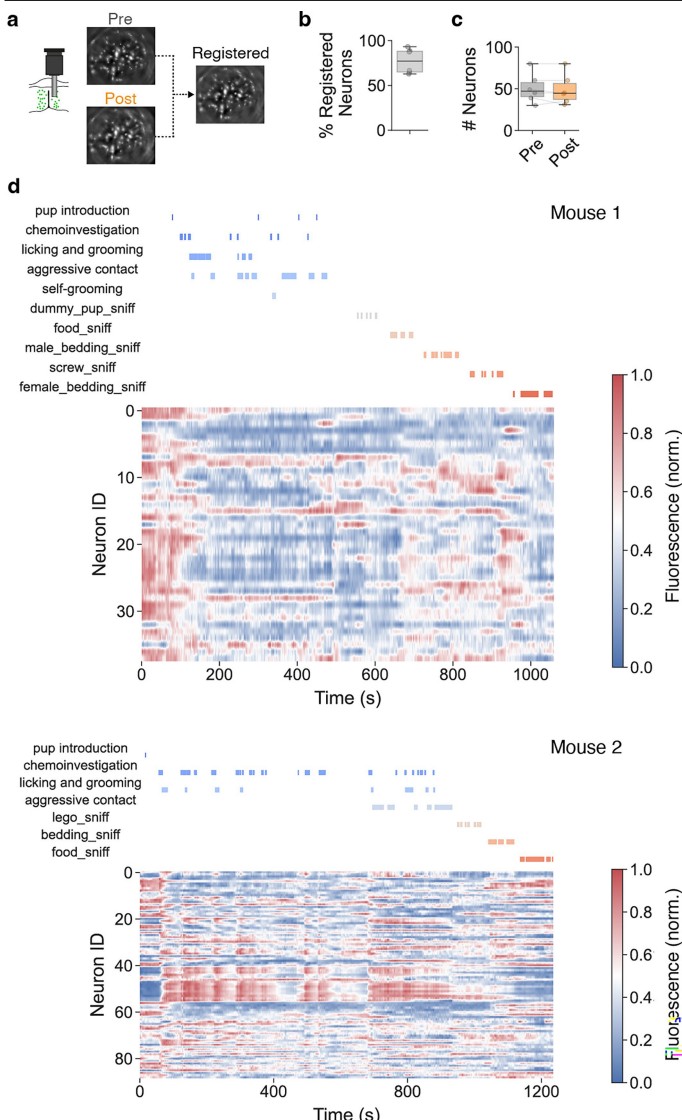

**a**, Pre / Post / Registered

**b**, % Registered Neurons

**c**, # Neurons — Pre, Post

**d**

Mouse 1

pup introduction
chemoinvestigation
licking and grooming
aggressive contact
self-grooming
dummy_pup_sniff
food_sniff
male_bedding_sniff
screw_sniff
female_bedding_sniff

Neuron ID / Time (s) / Fluorescence (norm.)

Mouse 2

pup introduction
chemoinvestigation
licking and grooming
aggressive contact
lego_sniff
bedding_sniff
food_sniff

Neuron ID / Time (s) / Fluorescence (norm.)

**Extended Data Fig. 8 | Longitudinal cell registration of micro-endoscopic images and MPOA neuronal responses to pup stimuli in Agg⁺ mice. a**, Example and miniature microscope recording frames before and after registration (see Methods). **b**, Percentage of successfully registered neurons per animal ($n = 5$ mice). **c**, Number of detected neurons in Pre and Post recording sessions ($n = 6$ mice). **d**, Temporal profile of neuronal responses during interactions with pups and other targets. Full behavioural episodes from two Agg⁺ mice are shown. Statistics: $U$ test (two-sided) in **c**. Box plots: median (line), interquartile range (box), whiskers, 1.5× IQR. See Supplementary Table 3 for further details of the statistical analyses.

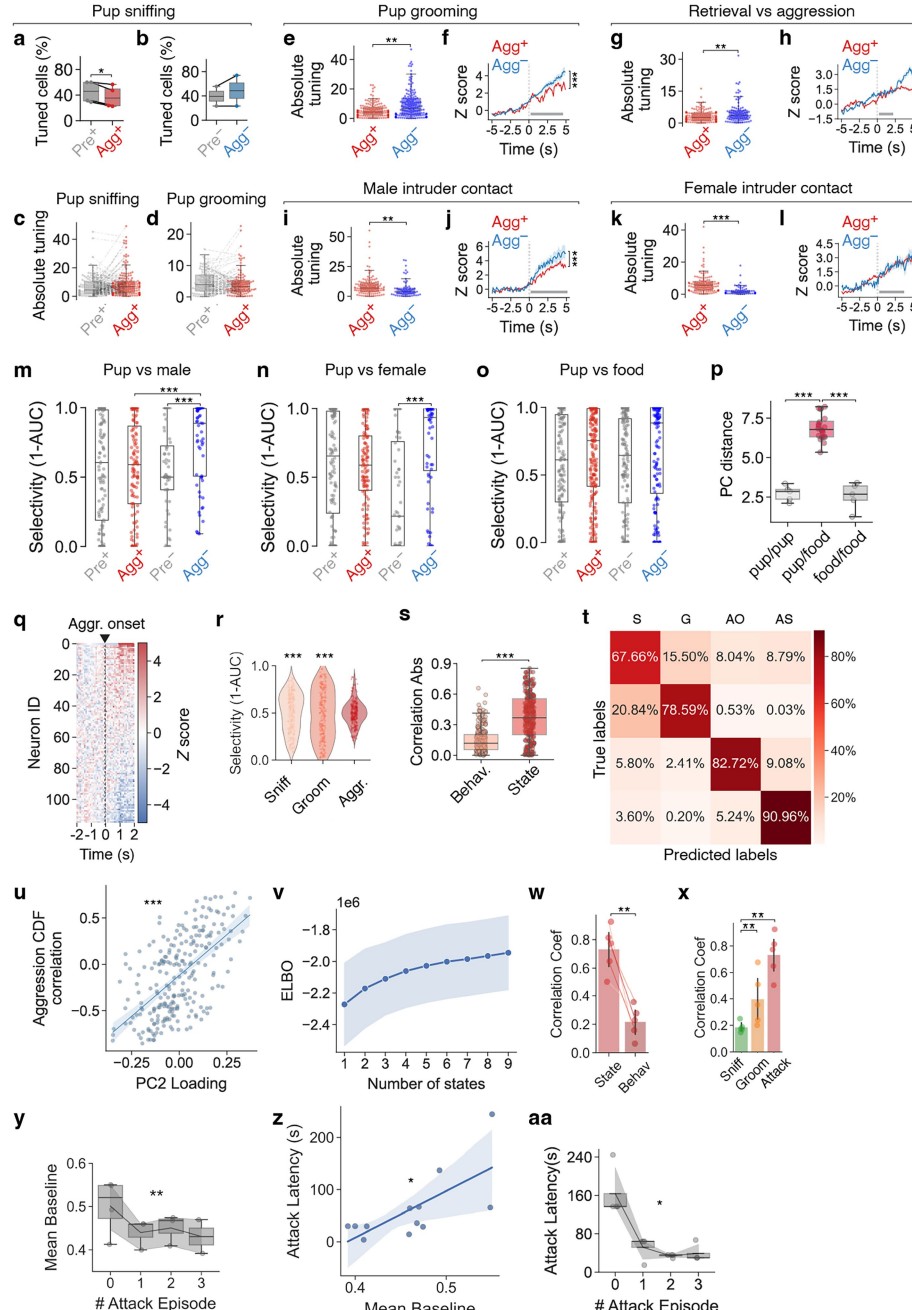

**Extended Data Fig. 9** | See next page for caption.

**Extended Data Fig. 9 | MPOA neuronal responses to pup stimuli in Agg⁻ mice, and aggression state encoding in the MPOA. a, b,** Percentage of MPOA neurons tuned (activated or inhibited) to pup sniffing in Agg⁺ (**a**, $n = 4$) and Agg⁻ (**b**, $n = 2$) mice before (Pre) and after food deprivation. **c, d,** Absolute tuning index of MPOA neurons during pup chemoinvestigation (**c**, $n = 154$ neurons, $N = 4$ mice) and pup grooming (**d**, $n = 99$ neurons, $N = 3$ mice) before and after food deprivation. **e–l,** Absolute tuning index (**e,g,i,k**) and Z-scored neuronal responses (**f,h,j,l**) of MPOA neurons during indicated behaviours ($n = 154$ (Agg⁺), 148 (Agg⁻) neurons, $N = 4$ (Agg⁺), 2 (Agg⁻) mice in **e**; $n = 54$ (Agg⁺), 52 (Agg⁻) neurons, $N = 4$ (Agg⁺), 2 (Agg⁻) mice in **f**; $n = 154$ (Agg⁺), 88 (Agg⁻) neurons, $N = 4$ (Agg⁺), 1 (Agg⁻) mice in **g**; $n = 33$ (Agg⁺), 16 (Agg⁻) neurons, $N = 4$ (Agg⁺), 1 (Agg⁻) mice in **h**; $n = 116$ (Agg⁺), 60 (Agg⁻) neurons, $N = 3$ (Agg⁺), 1 (Agg⁻) mice in **i,k**; $n = 64$ (Agg⁺), 19 (Agg⁻) neurons, $N = 3$ (Agg⁺), 1 (Agg⁻) mice in **j**; $n = 49$ (Agg⁺), 9 (Agg⁻) neurons, $N = 3$ (Agg⁺), 1 (Agg⁻) mice in **l**. **m–o,** Selectivity of chemoinvestigation-associated responses for indicated stimulus pairs compared with pups ($n = 116$, 60 neurons from $n = 3$, 1 mice in **m,n**; $n = 243$ (Agg⁺), 148 (Agg⁻) neurons, $N = 5$ (Agg⁺), 2 (Agg⁻) mice in **o**). Selectivity score 1 = neuron exclusively activated during pup sniffing; score 0 = exclusive activation during sniffing of another stimulus; score 0.5 = nonselective response. **p,** PC distance between Pre and Post episodes during pup vs food investigation ($n$ (left to right) = 5, 20 and 5 episodes, $N = 5$ mice). Dashed lines and grey bars in **f,h,j,l** indicate sniffing onset and mean bout duration (**f**, 4.6 s, **h**, 2.3 s, **j**, 8.2 s, **l**, 3.6 s), respectively. **q,** Z-scored neuronal responses during pup-directed aggression (average of episodes 2–5) with hierarchical clustering based on mean response onset ($n = 243$ neurons, $N = 5$ mice). **r,** Selectivity (see Methods) of MPOA neurons for indicated behaviours relative to no behaviour ($n = 243$ neurons, $N = 5$ mice). **s,** Absolute correlation of MPOA neuronal activity with either aggression ethogram ('Behaviour') or with cumulative distribution function (CDF) of aggression state ('State') ($n = 243$ neurons, $N = 5$ mice). **t,** Averaged confusion matrix from SVMs trained on PC1 and PC2. S, pup sniffing, G, pup grooming, AO, first aggression, AS, later aggression. **u,** Correlation between each neuron's PC2 loading and its correlation coefficient with the CDF of aggression episodes (linear regression, $R^2 = 0.356$, $P = 4.46 \times 10^{-25}$, $n = 243$ neurons, $N = 5$ mice). **v,** ELBO score for HMM fitting. **w,** Absolute correlation coefficients between MPOA neuronal activity and either the aggression ethogram (Behaviour) or the binary aggression state (State), where State is defined as 0 before and 1 after the first occurrence of aggression behaviour ($n = 5$ mice). **x,** Correlation between inferred HMM states associated with pup sniffing, pup grooming, or aggression and their binary behaviour state defined by the onset of the corresponding behaviour (0 before, 1 after first occurrence) ($n = 5$ mice). **y,** Mean baseline MPOA neuronal activity during the 3 s preceding attack onset across successive attack episodes ($n = 4$ mice). **z,** Correlation between mean baseline activity and latency of the subsequent attack episode (12 episodes from $n = 4$ mice; linear regression, $R^2 = 0.441$, $P = 0.011$). **aa,** Attack latency across consecutive attack episodes ($n$ (left to right) = 4, 3, 4 and 2 mice). Statistics: Paired $t$-test (two-sided) in **a,b,s,w,x**. Wilcoxon Signed-Rank test (two-sided) in **c,d**. U test in **e,g,i,k,m–o**. Two-way ANOVA with Tukey *post hoc* test in **f,h,j,l**. One-way ANOVA with Tukey *post hoc* test in **p**. Kurtosis test (two-sided) in **r**. One-way ANOVA in **y**. Repeated-measures ANOVA with Greenhouse-Geisser correction in **aa**. Data are mean ± s.e.m. Box plots: median (line), interquartile range (box), whiskers, 1.5× IQR. Shaded areas represent 95% CI. *$P < 0.05$, **$P < 0.01$, ***$P < 0.001$. See Supplementary Table 3 for further details of the statistical analyses.

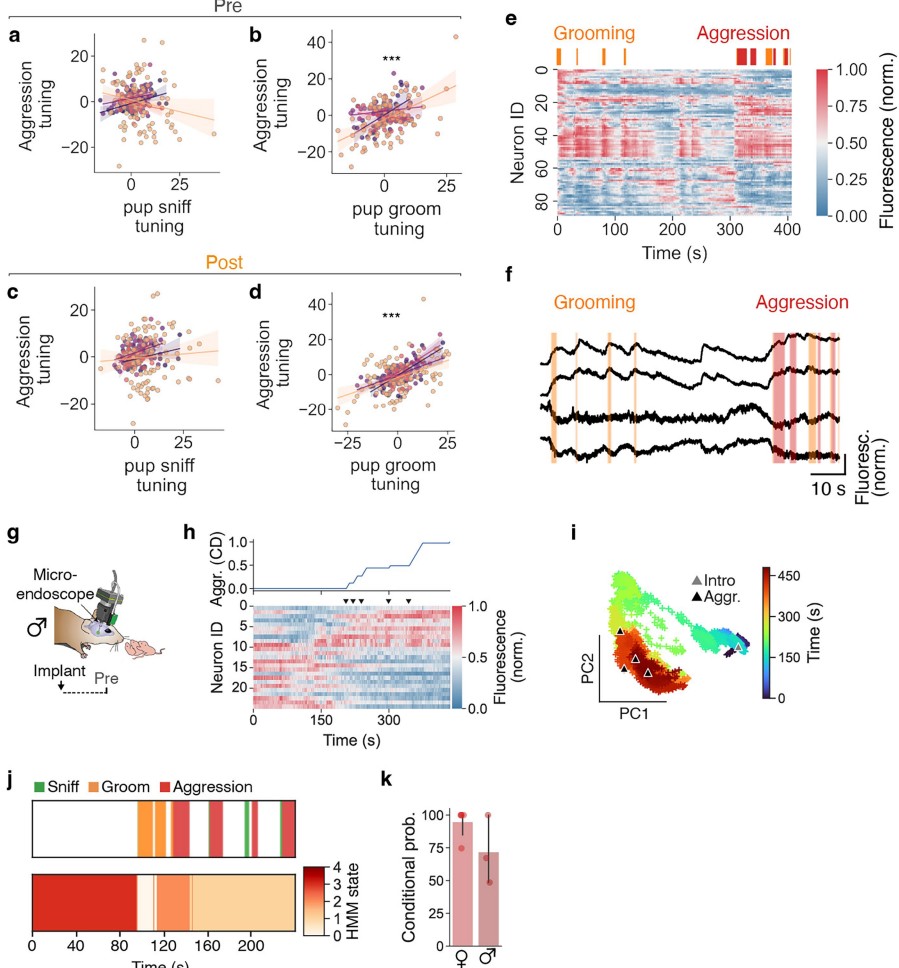

**Extended Data Fig. 10 | Behavioural co-tuning of MPOA neurons, and aggression state encoding in the MPOA of males. a–d**, Correlation between the tuning of MPOA neurons to pup-directed aggression and either pup sniffing (**a, c**, n = 243 neurons, N = 5 mice) or grooming (**b, d**, n = 186 neurons, N = 4 mice), assessed before (**a, b**) or after (**c, d**) food deprivation. Positive values represent activation, negative values inhibition. Each dot represents a single neuron; neurons from the same animal are shown in matching colours. Linear mixed-effects model with mouse ID as random effect (**a**, P = 0.204; **b**, P < 0.0001; **c**, P = 0.077; **d**, P < 0.0001). **e**, Representative heatmap of neuronal responses from an Agg⁺ animal displaying both pup grooming and pup-directed aggression (neurons sorted by hierarchical clustering; n = 89 neurons). **f**, Example traces from two MPOA neurons positively (*top*) or negatively (*bottom*) tuned to both

pup grooming and pup-directed aggression. **g**, Miniature microscope recordings during pup-directed aggression in virgin males. **h**, Example heatmap of neuronal activity sorted based on correlation of activity with cumulative distribution (CD) of aggression (n = 25 neurons). Arrows indicate attack episodes. **i**, Population activity traces projected onto first two PCs. Note the aggression-specific state along PC2. **j**, Example HMM state segmentation with ethogram (*top*) and Agg⁺ neural data *(bottom)*. **k**, Conditional probability of observing the indicated behaviour (aggression) when the system is in the HMM state most frequently aligned with aggression (n (left to right) = 5, 3 mice). Statistics: U test (two-sided) in **k**. Box plots: median (line), interquartile range (box), whiskers, 1.5× IQR. Data are mean ± s.e.m. *P < 0.05, **P < 0.01, ***P < 0.001. See Supplementary Table 3 for further details of the statistical analyses.

# Reporting Summary

## Statistics

For all statistical analyses, confirm that the following items are present in the figure legend, table legend, main text, or Methods section.

| n/a | Confirmed | |
|---|---|---|
| ☐ | ☒ | The exact sample size (*n*) for each experimental group/condition, given as a discrete number and unit of measurement |
| ☐ | ☒ | A statement on whether measurements were taken from distinct samples or whether the same sample was measured repeatedly |
| ☐ | ☒ | The statistical test(s) used AND whether they are one- or two-sided *Only common tests should be described solely by name; describe more complex techniques in the Methods section.* |
| ☐ | ☒ | A description of all covariates tested |
| ☐ | ☒ | A description of any assumptions or corrections, such as tests of normality and adjustment for multiple comparisons |
| ☐ | ☒ | A full description of the statistical parameters including central tendency (e.g. means) or other basic estimates (e.g. regression coefficient) AND variation (e.g. standard deviation) or associated estimates of uncertainty (e.g. confidence intervals) |
| ☐ | ☒ | For null hypothesis testing, the test statistic (e.g. *F*, *t*, *r*) with confidence intervals, effect sizes, degrees of freedom and *P* value noted *Give P values as exact values whenever suitable.* |
| ☒ | ☐ | For Bayesian analysis, information on the choice of priors and Markov chain Monte Carlo settings |
| ☒ | ☐ | For hierarchical and complex designs, identification of the appropriate level for tests and full reporting of outcomes |
| ☐ | ☒ | Estimates of effect sizes (e.g. Cohen's *d*, Pearson's *r*), indicating how they were calculated |

*Our web collection on statistics for biologists contains articles on many of the points above.*

## Software and code

Policy information about availability of computer code

**Data collection**
Ethovision XT 14 software (Noldus) was used for animal tracking. Videos were acquired using custom routines in Bonsai 2.9.0 (NeuroGEARS, https://bonsai-rx.org/) and behaviours were scored using BORIS v. 9.6.4 (https://www.boris.unito.it/). Widefield Images were acquired on a Vectra Polaris Automated Quantitative Pathology Imaging System (Akoya Biosciences) using Phenochart and inForm software (Akoya) for ROI selection and spectral unmixing. Stitching of spectrally unmixed image tiles was performed in QuPath-0.5.1. Confocal images were acquired on a Zeiss LSM 710 via ZEN 2.3 software. Slice electrophysiology data were acquired using pCLAMP 10.6.2 (Scientifica). Fibre photometry was performed on a P3001 fibre photometry system (Neurophotometrics) via Bonsai. Miniature microscopy imaging data were acquired using nVista HD 2.0 software (Inscopix).

**Data analysis**
Analysis of widefield images was performed in QuPath-0.5.1. Analysis of confocal images was performed in ImageJ (2.16.0). The ImageJ plugin ABBA 0.10.4 (https://abba-documentation.readthedocs.io/en/latest/) was used to register coronal brain sections to the Allen Brain Atlas. Mass spec data were analysed using MANIC software version 3.0.20. Analysis of slice electrophysiology data was performed with Clampfit 10 software (Molecular Devices), WinEDR v4, WinWCP v5 (http://spider.science.strath.ac.uk/sipbs/software_ses.htm), and custom routines written in Python 3.7. Analysis of fibre photometry data was performed in Python 3.7. Preprocessing of miniature microscopy data was performed in Inscopix Data Processing Software v1.5.1 (Inscopix) and further analysis performed in Python 3.7. Statistical analyses were performed in Python 3.7. Code created for this study is available at GitHub (https://github.com/FrancisCrickInstitute/negative_parental_switch).

For manuscripts utilizing custom algorithms or software that are central to the research but not yet described in published literature, software must be made available to editors and reviewers. We strongly encourage code deposition in a community repository (e.g. GitHub). See the Nature Portfolio guidelines for submitting code & software for further information.

## Data

Policy information about availability of data

All manuscripts must include a data availability statement. This statement should provide the following information, where applicable:

- Accession codes, unique identifiers, or web links for publicly available datasets
- A description of any restrictions on data availability
- For clinical datasets or third party data, please ensure that the statement adheres to our policy

The data that support the findings of this study are available from the corresponding author upon request. The previously published adult hypothalamus scRNA-seq dataset compiled by Yao et al. (2023) (WMB-10Xv3-HY-log2.h5ad) was downloaded from https://allen-brain-cell-atlas.s3.us-west-2.amazonaws.com/ index.html#expression_matrices/WMB-10Xv3/20230630/. Source data are provided with this paper.

## Research involving human participants, their data, or biological material

Policy information about studies with human participants or human data. See also policy information about sex, gender (identity/presentation), and sexual orientation and race, ethnicity and racism.

| | |
|---|---|
| Reporting on sex and gender | No human participants were used in this study |
| Reporting on race, ethnicity, or other socially relevant groupings | No human participants were used in this study |
| Population characteristics | No human participants were used in this study |
| Recruitment | No human participants were used in this study |
| Ethics oversight | No human participants were used in this study |

Note that full information on the approval of the study protocol must also be provided in the manuscript.

# Field-specific reporting

Please select the one below that is the best fit for your research. If you are not sure, read the appropriate sections before making your selection.

☒ Life sciences          ☐ Behavioural & social sciences          ☐ Ecological, evolutionary & environmental sciences

For a reference copy of the document with all sections, see nature.com/documents/nr-reporting-summary-flat.pdf

# Life sciences study design

All studies must disclose on these points even when the disclosure is negative.

| | |
|---|---|
| Sample size | No statistical methods were used to predetermine sample size. Sample sizes were estimated based on previous experiments performed in our group (Kohl et al., 2018, PMID: 29643503; Ammari et al., 2023, PMID: 37797007), and are consistent with those generally used in the field. |
| Data exclusions | Animals were only excluded if viral transduction was unsuccessful or off-target, or if fibre/cannula/lens tip placement was off-target. For electrophysiological recordings, only cells with a stable series resistance of <30MOhm were analysed. |
| Replication | The following experiments were replicated twice by different experimenters: switch to pup-directed aggression induced by 6h of food deprivation, slice physiology recordings across estrous cycle. All attempts at replication were successful. |
| Randomization | Animals were randomly assigned to treatment and control groups. Experimental groups consisted of multiple cohorts to avoid litter and cage effects. |
| Blinding | Data acquisition was not performed blind. Behavioural data was scored by an individual blind to the experimental design, and analysis of behavioural, histological, electrophysiological and in vivo imaging data was conducted under blind conditions. |

# Reporting for specific materials, systems and methods

We require information from authors about some types of materials, experimental systems and methods used in many studies. Here, indicate whether each material, system or method listed is relevant to your study. If you are not sure if a list item applies to your research, read the appropriate section before selecting a response.

## Materials & experimental systems

| n/a | Involved in the study |
|---|---|
| ☐ | ☒ Antibodies |
| ☒ | ☐ Eukaryotic cell lines |
| ☒ | ☐ Palaeontology and archaeology |
| ☐ | ☒ Animals and other organisms |
| ☒ | ☐ Clinical data |
| ☒ | ☐ Dual use research of concern |
| ☒ | ☐ Plants |

## Methods

| n/a | Involved in the study |
|---|---|
| ☒ | ☐ ChIP-seq |
| ☒ | ☐ Flow cytometry |
| ☒ | ☐ MRI-based neuroimaging |

# Antibodies

| | |
|---|---|
| Antibodies used | Primary antibodies: rabbit anti c-Fos (Synaptic Systems 226003, 1:2,000), rabbit anti-NPY (Abcam ab30914, 1:500), rabbit anti-AgRP (Abcam ab254558, 1:500); secondary antibodies: donkey anti-rabbit Alexa Fluor-568 (Thermo Fisher A-11057, 1:2,000), donkey anti-rabbit Alexa Fluor-647 (Thermo Fisher A-21245, 1:2,000), goat anti-rabbit Alexa Fluor-647 (Thermo Fisher A-21244, 1:1,000). |
| Validation | All antibodies used were commercial and validated in previous publications: rabbit anti c-Fos (PMID: 40702175), rabbit anti-NPY (PMID: PMID: 26946128), rabbit anti-AgRP (PMID: 39479445). |

# Animals and other research organisms

Policy information about [studies involving animals](); [ARRIVE guidelines]() recommended for reporting animal research, and [Sex and Gender in Research]()

| | |
|---|---|
| Laboratory animals | C57BL/6J mice from the Crick breeding colonies were used at age 8–14 weeks for all behavioural experiments. Agrp-Cre mice (JAX #012899) were used to target AgRP neurons. For slice physiology experiments, this line was crossed to Rosa26 tdTomato (Ai9, JAX #007909) reporter mice. For hormone receptor KO experiments, Esr1-loxP (estrogen receptor α conditional knockout, imported from EMMA, EM:11179) or PR-loxP (progesterone receptor conditional knockout, see Ammari et al., 2023, PMID: 37797007) were used. All mice were maintained in a C57BL/6J background. Mice had access to food and water ad libitum and were housed on a 12/12 h light-dark cycle (light on: 22:00–10:00) at 21°C and 32% humidity.<br><br>House crickets (Grillus domesticus) of either sex and 12–20 mm in length (Northampton Reptile Centre) were used as targets in prey hunting experiments. |
| Wild animals | No wild animals were used. |
| Reporting on sex | Experiments were performed in female mice unless indicated in the corresponding figure legends. |
| Field-collected samples | No field-collected samples were used in this study. |
| Ethics oversight | All animal procedures performed in this study were approved by the UK government (Home Office) and by the Crick Institutional Animal Welfare Ethical Review Panel (AWERB). |

Note that full information on the approval of the study protocol must also be provided in the manuscript.

# Plants

| | |
|---|---|
| Seed stocks | N/A |
| Novel plant genotypes | N/A |
| Authentication | N/A |

