## [Peer Review file · Nature]

Integration of hunger and hormonal state gates infant-directed aggression

Corresponding Author: Dr Johannes Kohl

Version 0:

Reviewer comments:

Referee #1

(Remarks to the Author)

In this study, Cao et al. report a novel interaction between the hunger state and estrous cycle that drives pup-directed aggression in female mice. The authors first describe a novel behavioral phenotype, wherein short-term food deprivation drives infanticidal behavior in female mice. NPY released from the arcuate nucleus inhibits MPOA neurons to mediate this behavior. Interestingly, the effectiveness of NPY-mediated inhibition, and thus the effectiveness of hunger to drive infanticide, depends on the animals' estrous state; animals with low levels of progesterone relative to estrogen (P4/E2 ratio; as in estrus) are less likely to show pup-directed aggression following food deprivation, while those with higher P4/E2 ratio (as in metestrus) are likely to attack pups. The authors dive into the physiological mechanisms underlying this estrous state-dependent gating of hunger signals, finding that HCN channels fluctuate as a function of the estrous state, which adjusts the effectiveness of NPY-mediated silencing of MPOA neurons. Finally, the authors use single-cell miniscope imaging to identify a subpopulation of MPOA neurons active during pup-directed aggression, which may signal an aggressive drive following food deprivation.

The findings from this study are quite interesting and pinpoint a compelling physiological mechanism for how hunger and hormonal cues interact. However, some questions remain to be addressed. In particular, the evidence supporting the direct impact of food deprivation on endogenous MPOA cell activity is weak.

Here are some specific comments.

1. The authors should discuss how and why this short-term food deprivation yields quite a dramatic behavioral phenotype. As the authors describe, C57BL/6 female mice very rarely attack pups, and a recent study using longer-term food deprivation did not report any infanticide in females (Alcantara et al., 2024 bioRxiv).
2. The authors should also discuss why food deprivation does not increase predation (attack of prey). Food deprivation clearly increases food intake, as shown in Fig. 1c, suggesting that the animal is hungry. It appears to be counter-intuitive that hunger does not increase predation. Are there any other prior studies supporting this behavioral observation?
3. In arcuate manipulation experiments (Fig 1d-k), do the animals show the same latency to attack as following food deprivation?
4. In the immediate early gene mapping experiment (Extended Data Fig. 2), it was unclear how c-Fos was induced. Is this c-Fos induced by pup interactions (aggression in Agg+ mice and investigation or parenting in Agg- mice)? Were all mice in Agg+ and Agg- groups deprived of food? If c-Fos was induced by pups (aggression vs. non-aggressive interactions), the authors should discuss the activation patterns they observe. Notably, BNST, PA, PeFA, and MeA have all been described as containing infanticide-promoting cell populations but do not differ between Agg+ and Agg- mice. Does food deprivation-induced infanticide significantly differ from naturally occurring infanticide?
5. What is the physiological mechanism of 30 minute pre-stimulation of the Arc to MPOA connection? Why is 15 minutes of stimulation insufficient (Fig. 1k)?
6. The authors make a compelling argument that the ratio of progesterone to estrogen, rather than estrogen or progesterone

concentrations alone, is crucial for the tendency to commit infanticide. However, to concretely make this point, more experiments are needed. The authors should systematically vary the levels of estrogen and progesterone – for example, by injecting hormones into ovariectomized females to artificially induce varying P4/E2 ratios and observe the switching rate following food deprivation.

7. In Fig. 2j, it is unclear why the authors predicted baseline switching rates. It would be preferable to track the estrous cycle of receptor knockout animals and report this.

8. In the *Esr1* KO females, do they show spontaneous infanticide or only attack pups after food deprivation?

9. Why does the excitability (evoked spikes) differ between Pre and Agg- groups (Fig. 3d)? Do those animals differ in the estrous cycle, resulting in different physiological properties? Extended data Fig. 10 suggests that most biophysical properties of MPOA neurons do not differ across the estrous cycle, but the authors do not display the evoked firing rates for these cells. The authors should report this, and ideally test animals from the same estrous state to make this point. This would also clarify the claim that the estrous cycle gates the effect of NPY-mediated inhibition.

10. In Figure 3c, 3d, and 3f, if the Agg- and Agg+ cells are combined, do they differ from Pre group? If not, it questions the claim that food deprivation changes the property of MPOA cells. Instead, the behavior output provides an axis to separate the preexisting heterogeneous responses of MPOA cells across animals.

11. The results presented make a compelling argument that the arcuate-to-MPOA projection is not GABAergic in this context. However, AgRP seems to have largely similar effects as NPY on the properties of MPOA neurons (Extended Data Fig. 7). AgRP seems to reduce the baseline firing rate of MPOA neurons (though there isn't a statistically significant effect). Importantly, AgRP also reduces the sag to the same magnitude as NPY. If you injected a retrograde shRNA against AgRP into the MPOA, would you also expect a reduction in infanticide? The authors use NPY antagonists to rescue the excitability of MPOA cells. To concretely show that this effect is mediated by NPY rather than AgRP, the authors should investigate the effectiveness of MC3R/MC4R antagonism and AgRP knockdown.

12. Please examine the correlation between sag amplitude and maximum firing of the IF curve (basically parameters shown in 3f and 3d). The rationale is that HCN level, which is reflected in the sag, alters the excitability of the MPOA cells and, hence, the tendency of the animal to show infanticide. Given that the cell excitability could be affected by many ion channels, whether the sag and excitability are indeed correlated should be reported.

13. The trends of sag magnitude across the estrous cycle nicely match the behavioral data. However, the HCN transcript levels do not. The authors should discuss this discrepancy – if HCN1 and HCN2 RNA are lowest in diestrus, why is the sag magnitude comparable to when RNA expression is highest (in estrus)?

14. The behavioral effects of HCN blockade are quite dramatic (Fig. 3v). The authors should provide additional behavioral characterization regarding this experiment.

15. The author states (page 9) “this difference was present before food deprivation and therefore likely reflects the influence of estrous state (Pre+ vs Pre-).” Please confirm whether the difference in Agg- and Agg+ animals before food deprivation indeed reflects the difference in estrous state vs. individual difference unrelated to the estrous cycle. Basically, for Pre+ and Pre-, does the animals' estrous state distribution differ?

16. Based on Figure 4b, it appears that food deprivation does not change the response of MPOA cells at all. The difference between Agg+ and Agg- are all preexisting. Again, this questions whether food deprivation directly influences MPOA cell activity.

17. I found the absolute tuning index of each cell's response during behavior (Fig. 4e) confusing. Does this differ at all from the average response during each behavior?

18. The authors claim that the activity of MPOA cells in Agg+ females correlates more closely with an aggressive state, rather than the specific action of aggression itself. I think this would largely depend on how the authors define a state. Presumably, an aggressive state would imply a persistent change in neural dynamics that lasts beyond a specific behavior. Indeed, the authors indicate that the initial aggression episodes shift the neural activity into a distinct representation. To really make this claim concretely, the authors would need to remove the pup following the initial aggressive encounter(s) or prevent the adult from attacking. If the neural activity is indeed state-encoding, the dynamics should persist independent of the consummation of aggression.

19. There needs to be more information about the PCA and classifier training in the methods. For example, exactly how was the PCA matrix constructed, and what percentage of variance was explained by the first two PCs? There is a dearth of information about how the classifiers were trained and validated. In particular, I cannot find any mention of the Hidden Markov Model in the

Referee #2

(Remarks to the Author)

This work identified a dual-gating neural mechanism controlling pup-directed aggression in females in a hunger and estrous state-dependent manner. The authors found that mild starvation induces pup-directed aggression in females that are otherwise non-infanticidal. AgRP neurons in the arcuate nucleus, which are known as hunger-sensing neurons, project to the MPOA, a region well-known for parenting behavior control, and induce pup-directed aggression through NPY release. The estrous cycle defines attack probability and HCN channel expression levels in MPOA neurons, with NPY release reducing HCN currents, potentially through reduction of cAMP that could bind to HCN channels. In vivo miniscope imaging confirmed attenuated average neural activity in the MPOA, while also identifying a population of neurons activated during pup-directed aggression, suggesting a potential novel role of MPOA neurons in controlling pup-directed aggression. Overall, this work is well conceptualized and executed, addressing an important mechanism of how diverse internal states interact to support optimal and flexible behavioral expression.

I have several comments and suggestions to improve clarity for readers:

1. The authors showed that Agg+ mice have more silent neurons than Agg- mice, MPOA neurons in Agg+ mice show low excitability upon current injection, and HCN blocker or NPY infusion increases the percentage of silent neurons in MPOA. However, miniscope data shows that a particular population of neurons exhibits strong activation during pup-directed aggression. Are these neurons silent during non-aggressive trials (Pre-trial in Agg+ mice) or active during different behaviors? This partially contradicts the model presented in Fig. 3w. How do the authors explain this discrepancy?
2. Since pup-directed aggression induced by mild hunger is a novel finding, it would be helpful to describe the behaviors of both Agg+ and Agg- mice in more detail:
 - Do Agg+ mice ever retrieve pups?
 - Do Agg- mice always show retrieval/parenting behavior or sometimes ignore pups?
 - Do Agg+ and Agg- mice show similar levels of non-aggressive pup-directed behaviors (sniffing, grooming etc.)?Adding behavioral quantifications and example ethograms in supplemental materials would be helpful. The interpretation of Extended Figure 2 may vary depending on these behavioral patterns.
3. What accounts for the differences between Agg+ and Agg- mice in the same estrous stages? Does it depend on individual differences in P4/E2 ratio or do other factors contribute?
4. The MPOA contains diverse neural populations. What is the overlap between ER, PR, NPY receptor, and HCN channel expression in MPOA? This is important since the proposed mechanism assumes all these components exist in the same neurons. Such data could be extracted from published scRNA-seq datasets.
5. Please define "aggressive state" on pages 9 and 11 and in Fig. 4. Does it refer to the entire period after aggression onset, the HMM state including aggression, or something else?
6. The methods for miniscope analysis need more detail. Descriptions of HMM analysis and selectivity analysis are missing. Please make sure the methods section fully explains all analyses.
7. The complex miniscope imaging data could benefit from more in-depth analysis to better understand aggression neural representation in MPOA and changes in pup neural representation under different hunger/estrous states. Here are suggestions:
 - Most neural data shown are either cropped around behavior onset or processed (e.g. PCA plot or HMM). Including example neural activity plots with behavior ethograms would help display diverse neural activity during behavior sessions. Activity heatmaps like Fig. 4h and Ex 15b with behavior ethograms (showing all pup interactions) above would be informative.
 - In Fig. 4d and Ex Fig. 13, the Z-scored neural activity shows a gradual increase, with larger differences after 2 sec potentially reflecting other behaviors. How were behavior bouts selected for averaged neural activity? Were bouts close to different behaviors excluded? If not, activity peaks may not correspond to the indicated behaviors. Please indicate mean bout durations in the Z-score plot and whether bouts were preselected.
 - What is the relationship between aggression-activated/inhibited neurons and their activity during other pup interactions like retrieval, grooming, and sniffing? Do neural tuning properties change pre- vs post-food deprivation? Comparing tuning between pre vs post in Agg+ mice would be informative.
 - Fig. 4m compares PC distances pre vs post for pup-directed behavior. While this quantifies neural representation changes, visualizing behavior locations in PC space (Pre-sniff, groom, Post-sniff, groom, attack, etc.) may better illustrate how representation shifts. Measuring distances between behaviors (for example, "pre-sniff vs post-aggression", "post-sniff vs post-aggression") would reveal overall representation changes.
 - How well can pup retrieval be decoded from entire neural data or PC1/2 of Agg- mice? Since MPOA is well known for parenting behavior control, comparing decoding of parenting vs aggression would reveal how well MPOA activity explains each behavior and whether they occupy the same PC space.
 - What is the difference between Fig. 4f and Extended Fig. 14a?

Minor comments/errors:

1. Methods, page 27, Hormone receptor knockout section: The description "AAV2/1-syn-fDIO-EGFP-2A-iCre (250 nl) or AAV2/1-syn-fDIO-GCaMP7s-2A-iCre (250 nl) was injected into the MPOA of Esr1-loxP or PR-loxP mice" appears incorrect as there is no Flpo.
2. Extended Fig. 4g-h: How can KO cell numbers be measured? AAV-GFP-Cre injection in Esr1 or PR loxP mice would show GFP in all infected cells regardless of Esr1/PR expression. The X-axis should indicate GFP+ cells rather than KO cells.

3. Fig. 2j: Please change "-- baseline" to "-- prediction" to match panel h.
4. Extended Fig. 8 e-i are not cited in the text. Consider citing around page 12, lines 8-11.
5. Fig. 3C legend: Please add descriptions for ZD and NPY.
6. Page 9, line 15: Suggested edit: "HCN channels via reducing cAMP through NPY receptor signaling."
7. Page 12, line 24: Reference 54 is cited twice.

Referee #3

(Remarks to the Author)

The manuscript by Cao et al. from the Kohl lab, answers of very important question in the field of animal behavior in general and specifically of maternal behavior - how do animals integrate and prioritize different biological needs to produce behavior. The authors show with high temporal resolution that hunger increases pup aggression in virgin females mice, who normally are tolerant toward pups. They then use chemogenetic approaches to show that activation of AgRP+ neurons from the Arc nucleus is necessary and sufficient for this effect. Furthermore, they show that prolonged activation of AgRP+ neurons projecting to the MPOA are necessary for this behavior. Interestingly they also show that the estrus cycle induces significant variability in pup aggression induced by food deprivation. They elegantly show that the ratio of P4/E2 hormones is the determinant factor. In whole-cell slice recordings from MPOA the authors then show that baseline and evoked firing as well as the sag amplitude were decreased in aggressive females compared to non-aggressive ones following food deprivation. The % of silent MPOA neurons was increased in aggressive virgins. They then show that knocking down NPY, one of the hormones produced by AgRP+ cells rescues these electrophysiological changes in aggressive virgins, and delayed the onset of pup aggression events. Sag amplitude is a measure of HCN channels, that integrate cAMP signal with membrane potential hyperpolarization, and the authors show that the amplitude of sag as well as Hcn transcripts varies with the estrus cycle. Pharmacological blockade of HCN inhibitors turned virtually all virgin females aggressive. Lastly, miniscope recordings in behaving females show less activity in MPOA in aggressive females during pup sniffing. With their evidence and knowledge from prior work, the authors propose a model where hunger releases NPY in MPOA to reduce cAMP and suppress activity of HCN channels, which can result in significant reduction of neuronal firing and pup aggression behavior depending on P4/E2-dependent abundance of HCN channels.

This is carefully designed and executed study, with an impressive number of control experiments. The findings are novel and exciting not only for understanding maternal care (and its failures) but more broadly for understanding the biological substrates by which a hierarchy of needs drives behavior.

I have a few and relatively minor concerns:

1. I am not sure that the pattern of AgRP neuron activity is sufficient to conclude that: 'aggressive females do not perceive pups as food'. Perhaps it would be more informative to know if the virgins cannibalize the pups when hungry or just aggress them.
2. The amplitude of sag in Figure 3q, does not perfectly match the Hcn transcripts in Fig 3s, particularly for diestrus. Moreover, Hcn expression is high in estrus and low in diestrus, but in terms of behavior, these two estrus phases both have relatively low aggressive rates after food deprivation (below 50%, Fig. 2f). What could be the explanation?
3. For the model proposed to be validated, I think the authors need to show that NPY receptors and HCN are co-expressed in MPOA neurons. The authors do mention in the discussion that ~70% of MPOA neurons express NPY receptors, so presumably the co-expression likelihood is high.
4. In Figure 4, the estrus cycle seems to be abandoned – is there less aggressive state in estrus?

Some comments on the Methods:

5. why is 'frantic carrying of pups' considered aggressive behavior? Wouldn't it be explained by higher anxiety?
6. the authors state that aggressive animals targeted all pups in cage (2), despite also stating that the assay was terminated immediately upon observing pup-directed aggression. It seems it can't be both.

Version 1:

Reviewer comments:

Referee #1

(Remarks to the Author)

The authors have thoroughly addressed all the reviewers' comments.

Referee #2

(Remarks to the Author)

The authors have thoroughly and thoughtfully addressed the concerns raised by the reviewers. The revisions have strengthened the manuscript, making the data presentation clearer and the conclusions more convincing. I appreciate the care taken to improve both clarity and rigor. I have no further concerns or suggestions at this stage. I believe this exciting work is now ready for publication.

Referee #3

(Remarks to the Author)

The authors have added an impressive number of experiments and analysis to address the concerns that I and the other reviewers had. This is a very strong manuscript, with appropriate and well described statistical analysis, and I recommend it for publication.

Dear Editor and referees,

we sincerely appreciate the time and effort you dedicated to reviewing our manuscript. Your constructive feedback has significantly improved its clarity and overall quality. We were encouraged by your positive comments and have conducted additional experiments, analyses, and textual revisions to address your suggestions and concerns.

Specifically:

1. In response to referee 1, we have performed additional experiments and analyses to support our claim that food deprivation changes the biophysical properties of MPOA neurons independently of behavioural outcome or estrous cycle stage.
2. In response to referee 1, we have performed projection-specific knockdown of *Agrp* in $Arc^{AgRP} \rightarrow MPOA$ projections. We find that this manipulation does not affect pup-directed aggression, and that it does not affect the voltage sag amplitude, baseline activity, or excitability of MPOA neurons.
3. We have performed extensive additional analysis of our behavioural, electrophysiology and *in vivo* imaging data in response to all referees.
4. In response to referees 2 and 3, we have analysed published scRNA-seq datasets to address the co-expression of *Hcn*, *Npyr*, *Esr1* and *Pgr* in MPOA neurons.
5. In response to referees 1 and 2, we have improved and extended the Methods section.
6. We have also expanded the Discussion in response to all referees and improved the labelling in several figures.

Please note that the numbering of most Extended Data Figures has changed in the revised manuscript; we have referenced the original figure numbers where relevant for clarity.

Point-by-point replies to the referees' comments (*in italics*) are provided below (our responses are shown in blue).

Referee #1 (Remarks to the Author):

In this study, Cao et al. report a novel interaction between the hunger state and estrous cycle that drives pup-directed aggression in female mice. The authors first describe a novel behavioral phenotype, wherein short-term food deprivation drives infanticidal behavior in female mice. NPY released from the arcuate nucleus inhibits MPOA neurons to mediate this behavior. Interestingly, the effectiveness of NPY-mediated inhibition, and thus the effectiveness of hunger to drive infanticide, depends on the animals' estrous state; animals with low levels of progesterone relative to estrogen (P4/E2 ratio; as in estrus) are less likely to show pup-directed aggression following food deprivation, while those with higher P4/E2 ratio (as in metestrous) are likely to attack pups. The authors dive into the physiological mechanisms underlying this estrous state-dependent gating of hunger signals, finding that HCN channels fluctuate as a function of the estrous state, which adjusts the effectiveness of NPY-mediated silencing of MPOA neurons. Finally, the authors use single-cell miniscope imaging to identify a subpopulation of MPOA neurons active during pup-directed aggression, which may signal an aggressive drive following food deprivation. The findings from this study are quite interesting and pinpoint a compelling physiological mechanism for how hunger and hormonal cues interact. However, some questions remain to be addressed. In particular, the evidence supporting the direct impact of food deprivation on endogenous MPOA cell activity is weak.

We thank the referee for their support for our work, and for their very insightful comments and helpful suggestions.

Here are some specific comments.

1. *The authors should discuss how and why this short-term food deprivation yields quite a dramatic behavioral phenotype. As the authors describe, C57BL/6 female mice very rarely attack pups, and a recent study using longer-term food deprivation did not report any infanticide in females (Alcantara et al., 2024 bioRxiv).*

Thank you for highlighting this important point. While both our study and that of Alcantara et al. (2024) used C57BL/6 females, a key difference lies in parental experience. In our study, mice were naïve to pups prior to testing (see Methods > Behavioural profiling), whereas in Alcantara et al., females were pup-sensitised (see e.g., Stolzenberg & Rissman, 2011, PMID 21276101)—they had been repeatedly exposed to pups until they displayed robust parental behaviour. In Alcantara et al., 10 out of 10 virgins retrieved pups (see their Extended Data Fig. 4) and their Methods section states that "to promote alloparenting in virgin females, mice were exposed to pups for 2–3 consecutive days". Sensitisation—which is known to induce lasting pro-parental behavioural changes (e.g., Bridges & Scanlan, 2005, PMID 15690385; Stolzenberg & Rissman, 2011, PMID 21276101)—therefore appears to buffer against competing states such as hunger. We now discuss this in the revised manuscript (page 15, lines 19–22):

“Notably, repeated pup exposure (sensitisation) appears to prevent the hunger-induced switch to pup-directed aggression via an unknown mechanism³². Future work will explore how social experience modulates the Arc^{AgRP}→MPOA circuit to shape infant-directed behaviour.”

2. The authors should also discuss why food deprivation does not increase predation (attack of prey). Food deprivation clearly increases food intake, as shown in Fig. 1c, suggesting that the animal is hungry. It appears to be counter-intuitive that hunger does not increase predation. Are there any other prior studies supporting this behavioral observation?

Thank you for raising this point. The primary aim of these control experiments was to determine whether food deprivation induces indiscriminate aggression towards various targets, including pups, adults and prey. Our initial analysis therefore focused on the percentage of mice initiating hunting behaviour in the prey assay ('% Aggressive', Extended Data Fig. 1e), from which we concluded that food deprivation does not significantly increase the likelihood of initiating a hunt. However, a closer examination of the data revealed that among those mice that did initiate hunting the proportion that consumed the cricket rose from 33% in sated animals to 100% following food deprivation (see Referee Fig. 1, now Extended Data Fig. 1f). This indicates that while food deprivation does not increase the probability of initiating predation, it does enhance prey consumption once hunting is initiated.

Referee Fig. 1. Percentage of mice that consume prey after initiating hunting behaviour (n = 3, 4 mice)

We have now rephrased the Results section (page 2, lines 20–22):

“This aggression was specifically directed at pups, as the proportion of mice attacking prey or adult intruders of either sex was unaffected by food deprivation (Extended Data Fig. 1e).”

We also refer to this in the Discussion (page 15, lines 10–12):

“Food-deprived mice are more likely to consume prey (Extended Data Fig. 1f) [...]”

3. In arcuate manipulation experiments (Fig 1d-k), do the animals show the same latency to attack as following food deprivation?

Yes, we find that attack latencies are comparable between chemogenetic activation (Gq), optogenetic activation (Opto, 30 min stimulation of MPOA projections before pup interactions), and food deprivation. As shown in Referee Fig. 2 (now Extended Data Fig. 4i) below, there is no significant difference in latency between these groups (one-way ANOVA: $F = 0.715$ $P = 0.494$). We now reference these findings in the main text (page 4, lines 34–page 5, line 2):

“Opto- and chemogenetic Arc^{AgRP} manipulations resulted in pup attack latencies comparable to those observed after food deprivation (Extended Data Fig. 4i), suggesting that engaging this circuit is sufficient to replicate the behavioural switch induced by metabolic state changes.”

Referee Fig. 2. Pup attack latency after chemogenetic activation of Arc^{AgRP} neurons (Gq), optogenetic stimulation of $Arc^{AgRP} \rightarrow MPOA$ projections (Opto, 30-min pre stimulation), and food deprivation ($n = 7, 5, 39$).

4. In the immediate early gene mapping experiment (Extended Data Fig. 2), it was unclear how *c-Fos* was induced. Is this *c-Fos* induced by pup interactions (aggression in *Agg+* mice and investigation or parenting in *Agg-* mice)? Were all mice in *Agg+* and *Agg-* groups deprived of food? If *c-Fos* was induced by pups (aggression vs. non-aggressive interactions), the authors should discuss the activation patterns they observe. Notably, BNST, PA, PeFA, and MeA have all been described as containing infanticide-promoting cell populations but do not differ between *Agg+* and *Agg-* mice. Does food deprivation-induced infanticide significantly differ from naturally occurring infanticide?

Apologies for the lack of clarity. *c-Fos* mapping was conducted in food-deprived mice—the observed *c-Fos* labelling thus reflects both the effects of food deprivation and pup interactions. To identify brain regions specifically associated with pup-directed aggression, we compared *c-Fos* expression between food-deprived mice that either attacked (*Agg+*) or did not attack (*Agg-*) pups.

We have now added a dedicated section describing these experiments to the Methods section (Behavioural profiling > c-Fos mapping) and reference it in the main text (page 4, line 11).

We appreciate your important point regarding the absence of c-Fos differences between Agg⁺ and Agg⁻ mice in regions previously implicated in infanticide, including the BNST, PA, PeFA and MeA. While initially surprising, our findings do not contradict the existing literature. For example, Mei et al. (2023, PMID 37286598) reported elevated c-Fos expression in the BNST and MeA of infanticidal Swiss Webster mice compared to *undisturbed*, single-housed females (see their Fig. 1b). In contrast, we compare two groups of mice that are both engaged in (aggressive vs non-aggressive) pup interactions, both of which likely recruit MeA and BNST neurons which are in chemosensory processing pathways.

Similarly, Autry et al. (2021, PMID 34423776) observed increased c-Fos expression in BNST and PeFA of infanticidal males relative to fathers and mothers (see their Fig. 1a, b). However, in the same study, there was no significant difference in PeFA c-Fos levels between pup-retrieving and pup-attacking virgin females (see their Fig. 1h). Therefore, differences in experimental design—including strain, sex, and comparison group—likely explain the discrepancies between our studies.

One possible explanation for the lack of c-Fos differences between Agg⁺ and Agg⁻ mice in these regions is that distinct neuronal subpopulations within each area may show opposing activity changes—some becoming more active, others less—and this heterogeneity could be masked by population-level averaging. Alternatively, given that all five brain areas with significantly different c-Fos densities showed lower ‘activity’ in Agg⁺ mice (Extended Data Fig. 3c; previously Extended Data Fig. 2c), pup-directed aggression—driven by inhibitory Arc^{AgRP} neurons—may operate via disinhibition, with key excitatory neurons located outside the regions assessed here.

We now discuss these points in the revised manuscript (page 4, lines 13–18):

“The absence of c-Fos differences in areas implicated in female infanticide (BNST, PA, PeFA, and MeA)^{28,29} may result from masking of bidirectional activity changes in neuronal subsets by population averaging. Alternatively, pup-directed aggression—driven by inhibitory Arc^{AgRP} neurons—may rely on disinhibition, with key excitatory neurons located elsewhere, consistent with lower c-Fos⁺ densities in Agg⁺ mice (Extended Data Fig. 3c).”

Finally, the behavioural characteristics of food deprivation-induced pup-directed aggression are indistinguishable from naturally occurring infanticide in both females and males (Autry et al., 2021, PMID 34423776; Mei et al., 2023, PMID 37286598; Isogai et al., 2018, PMID 30550786), including rapid, rhythmic head movement, biting, and aggressive carrying of pups around the cage (see Methods > Behavioural profiling > Pup-directed behaviour assay).

5. *What is the physiological mechanism of 30 minute pre-stimulation of the Arc to MPOA connection? Why is 15 minutes of stimulation insufficient (Fig. 1k)?*

Our optogenetic pre-stimulation protocol was inspired by prior studies demonstrating that prolonged Arc^{AgRP} stimulation is required for maximal feeding effects. Aponte et al. (2011, PMID 21209617) and Betley et al. (2013, PMID 24315102) used 1-h optogenetic Arc^{AgRP} stimulation to drive robust feeding responses. More recently, Chen et al., (2019, PMID 31033437) showed that while 5 min of Arc^{AgRP} neuron pre-stimulation can increase food consumption, maximal effects require at least 30 min, a process dependent on NPY release (see their Fig. 1g, m).

Given that the behavioural effects of Arc^{AgRP}→MPOA projections on pup-directed behaviour are also NPY-mediated, it is likely that shorter (e.g., 15-min) stimulation is insufficient to induce significant NPY release. In contrast, 30 min of continuous activation may allow enough dense core vesicle fusion and peptide release to modulate downstream MPOA circuits. Moreover, as NPY likely acts through volume transmission in the MPOA and signals via relatively slow G-protein-coupled receptor pathways, the behavioural effects of NPY may emerge only after sufficient accumulation in the extracellular space and prolonged receptor engagement. This notion is also supported by the observation that attack latency decreases with prolonged food deprivation (Extended Data Fig. 1a). We have now updated the discussion accordingly (page 14, lines 19–26):

“The observation that 30 min of Arc^{AgRP}→MPOA pre-stimulation is necessary for the switch to pup-directed aggression (Fig. 1k) suggests that this behavioural transition requires sustained NPY release and/or slow integration of the neuropeptidergic signal in the MPOA. This aligns with prior work showing that prolonged Arc^{AgRP} activation is needed for maximal NPY-dependent feeding responses^{1–3}. As NPY is released from dense-core vesicles, likely acts via volume transmission and through slow-acting GPCR pathways^{4,5}, extended stimulation may be required to reach effective levels of neuromodulation.”

6. *The authors make a compelling argument that the ratio of progesterone to estrogen, rather than estrogen or progesterone concentrations alone, is crucial for the tendency to commit infanticide. However, to concretely make this point, more experiments are needed. The authors should systematically vary the levels of estrogen and progesterone – for example, by injecting hormones into ovariectomized females to artificially induce varying P4/E2 ratios and observe the switching rate following food deprivation.*

Thank you for this suggestion. We have now attempted to experimentally manipulate P4 and E2 levels in ovariectomised females via (1) subcutaneous hormone pellet implantation, and (2) subcutaneous injection of E2 and P4. In both paradigms, negative control groups received

hormone-free implants or injections. Regardless of treatment group, all animals showed pup-directed aggression after 6 h of food deprivation (see Referee Table 1), including controls. In contrast, no animals displayed aggression prior to food deprivation. We hypothesise that pup-directed aggression in these animals is evoked by a combination of food deprivation and stress and/or inflammation associated with the administration procedures.

While these interventions precluded a test of the hormone ratio hypothesis by direct hormone administration, two lines of evidence strongly support the conclusion that the P4/E2 ratio is a key determinant of switching probability:

(1) The switching probabilities observed in mid- and late-pregnant females—who have much higher absolute levels of both hormones—closely match the values predicted by our model based on virgin females (Fig. 2g and Extended Data Table 2), suggesting that it is indeed the ratio, rather than absolute hormone concentrations, that sets aggression probability.

(2) Genetic ablation of hormone receptors in MPOA neurons—without altering circulating hormone levels—effectively alters the perceived P4/E2 ratio by these cells. These *manipulations* yield switching rates that are in strong agreement with model predictions: *Esr1* deletion (raising the perceived P4/E2 ratio to ~1) results in ~100% switching, while *PR* deletion (lowering the perceived ratio to ~0) yields ~20% switching (Fig. 2f, j).

Together, these results indicate that the P4/E2 ratio—as sensed by MPOA neurons—is a robust determinant of pup-directed aggression following food deprivation, even in the absence of direct experimental titration of hormone levels.

Referee Table 1: Summary of hormone manipulation experiments

Manipulation	Substance	Number of mice	Phenotype before 6-h food deprivation	Phenotype after 6-h food deprivation
Implantation of subcutaneous pellet	E2	12	Agg ⁻ (12/12)	Agg ⁺ (12/12)
Implantation of subcutaneous pellet	P4	9	Agg ⁻ (9/9)	Agg ⁺ (9/9)
Implantation of subcutaneous pellet	blank	9	Agg ⁻ (9/9)	Agg ⁺ (9/9)
Subcutaneous injection (2x injection)	E2	6	Agg ⁻ (6/6)	Agg ⁺ (5/6)
Subcutaneous injection (2x injection)	P4	7	Agg ⁻ (7/7)	Agg ⁺ (7/7)
Subcutaneous injection (2x injection)	Sesame oil	5	Agg ⁻ (5/5)	Agg ⁺ (5/5)
Subcutaneous injection (1x injection)	E2	4	Agg ⁻ (4/4)	Agg ⁺ (4/4)

Methodological details

Ovariectomy: Female C57BL6/J mice 8–12 weeks of age received oral meloxicam (10 mg/kg), were anaesthetised with isoflurane (5% induction, 1.5% maintenance) in oxygen-enriched air, and secured in a stereotaxic frame (Model 940, Kopf Instruments). Meloxicam (10 mg/kg) and buprenorphine (0.1 mg/kg) were administered subcutaneously. A small dorsal incision was made through skin and muscle, and the ovaries were bilaterally exteriorised and removed. Incisions were closed using Vetbond (3M), and mice were returned to a heated chamber for recovery. Oral meloxicam (10 mg/kg) was continued for three days post-operatively. Ovariectomised females were allowed to recover for 3–4 weeks before behavioural testing.

Subcutaneous implantation of hormone pellets: we followed a protocol by Topilko et al. (2022, PMID 35123655). Hormone pellets were prepared as follows: 100 mg of progesterone (Sigma) was thoroughly mixed over 10 min with 300 mg of the dimethyl siloxane-containing multi-purpose sealant (Dowsil 732) using plastic tips. 200 mg of estradiol (Sigma) was mixed with 560 mg of Dowsil and 40 μ l of DMSO (Sigma). Dowsil-only was used for controls. The respective mixtures were pressed between glass slides wrapped in greaseproof paper and left to harden overnight. The following day, tablets were weighed and cut into pieces of 40 mg to obtain 10-mg progesterone or estradiol (or control) pellets, respectively. For implantation, mice were anaesthetised under isoflurane (5% induction, 1.5% maintenance) and analgesic (meloxicam, 10 mg/kg) was administered. Mice were positioned over a heat pad at 37°C and a 3–4 mm incision was made through the skin, dorsally on the neck. A small hormone (or control) pellet was placed in the subcutaneous tissue and the skin sutured using Vicryl 6.0. Mice stayed in a chamber at 37°C until fully recovered. Oral meloxicam (10 mg/kg) was continued for three days post-operatively, and behavioural testing was performed 3 d after surgery.

Subcutaneous hormone injection: ovariectomised females received two subcutaneous 100 μ l (per 20g body weight) injections of either 5mg/ml progesterone (Sigma) or 0.1 mg/ml estradiol (Sigma) in sesame oil (Sigma) on the day of behavioural testing, one immediately after the pup-directed behaviour testing in the Pre phase (before food deprivation), and one 2 h before the behaviour testing in the Post phase ('2x injection'). Control females received identical volumes of vehicle (sesame oil). A separate cohort of 4 mice was injected once each with 150 μ l of a 0.1 mg/ml solution of estradiol in sesame oil and tested 2 d later ('1x injection').

7. In Fig. 2j, it is unclear why the authors predicted baseline switching rates. It would be preferable to track the estrous cycle of receptor knockout animals and report this.

We apologise if this was not clear in the original manuscript. To clarify, the estrous cycle stage of each experimental animal was tracked, and the predicted (expected or baseline) switching rate for each group derived accordingly.

We used this approach because each experimental group contained a variable—but known—composition of animals in different estrous stages, and the expected mean switching rate therefore differed across groups. Knowledge of the average switching rate of each estrous stage (Fig. 2e, f), allowed us to calculate each group’s predicted switching rate using hypothesis testing based on a Poisson binomial distribution, determined by the estrous cycle composition of the experimental group (see Methods > Quantification and Data Analysis > Calculation of predicted baseline switching rates).

For example, the MPOA *Esr1* KO group contained 1 individual in proestrus, 4 in estrus, 2 in metestrus, and 4 in diestrus. Based on this distribution, the Poisson binomial model predicted a switching probability range of 17.1–77.9%, with a mean expected baseline of 40%. The *observed* switching probability of 100% falls outside the 95th percentile of this distribution, indicating a statistically significant difference.

To improve clarity, we have revised the text (page 5, lines 27–29) as follows:

“100% of *Esr1*-ablated mice became aggressive after food deprivation, compared to a predicted 40% baseline rate for a group of mice with intact receptors, based on the measured estrous stage distribution (Fig. 2j, see Methods)”

In addition, we have updated the legend of Fig. 2j:

“Percentage of Agg⁺ mice after *Esr1* or PR KO (n = 10, 13, 8, 10, 11 mice). Dashed lines indicate predicted baseline switching rates of each cohort with intact receptors based on the measured distribution of estrous stages (see Methods)”.

Finally, we revised the Methods section to further clarify our approach and explicitly state that the *poibin* package was used for this analysis.

8. *In the *Esr1* KO females, do they show spontaneous infanticide or only attack pups after food deprivation?*

In our experiments, neither *Esr1* nor PR KO females showed spontaneous infanticide before food deprivation (see Referee Fig. 3). We now state this in the revised text (page 6, lines 4–6).

Referee Fig. 3. Percentage of Agg⁺ mice after KO of Esr1 (n = 0/20 mice) or PR (n = 0/24 mice) in MPOA. or Arc.

9. Why does the excitability (evoked spikes) differ between Pre and Agg⁻ groups (Fig. 3d)? Do those animals differ in the estrous cycle, resulting in different physiological properties? Extended data Fig. 10 suggests that most biophysical properties of MPOA neurons do not differ across the estrous cycle, but the authors do not display the evoked firing rates for these cells. The authors should report this, and ideally test animals from the same estrous state to make this point. This would also clarify the claim that the estrous cycle gates the effect of NPY-mediated inhibition.

Thank you for pointing this out. The difference in excitability between the Pre and Agg⁻ groups appears to be attributable to differences in their estrous cycle composition. When we reanalysed the data using Pre and Agg⁻ groups matched for estrous stage distribution, the difference in excitability was no longer observed (see Referee Fig. 4a), indicating that the original effect was likely driven by hormonal stage.

Referee Fig. 4. a, b, Action potentials per injected somatic current between estrous stage-matched Pre and Agg⁻ groups (**a**, 18, 19 cells from n = 11, 3 mice) and across estrous cycle stages (**b**, P, E, M, D; 10, 13, 16, 14 cells from n = 3, 6, 6, 4 mice). Bars indicate periods of significant difference, and shading indicates significance level (from top to bottom: D vs E; D vs P). Mixed linear model with cell ID as random effect.

We have now examined MPOA neuronal excitability across the estrous cycle. Despite sample sizes comparable to those used in our main analyses, we did not detect consistent differences in evoked firing across stages (Referee Fig. 4b; now Extended Data Fig. 13n). This likely reflects the relatively modest influence of estrous stage on excitability, coupled with substantial within-group variability. A similar pattern is observed for voltage sag amplitude, which shows moderate fluctuations across the cycle and high variability even with large sample sizes (Fig. 3q).

Nonetheless, since sag amplitude is positively correlated with excitability (Referee Fig. 7), these findings indirectly support a modulatory role for the estrous cycle. In our model, excitability is shaped by the combined effects of estrous stage-dependent *Hcn* expression (Fig. 3q–s) and NPY

signalling during food deprivation. While food deprivation alone modestly reduces excitability (Referee Fig. 5), this effect is amplified in the context of estrous cycle variation (Fig. 3d). We now clarify this point in the main text (page 10, lines 10–13):

“Hunger and estrous state thus converge on HCN channels to regulate MPOA neuron activity and excitability. While estrous stage modulates HCN channel abundance, NPY release during food deprivation inhibits available HCN channels. Neither signal alone substantially alters excitability (Extended Data Fig. 9 and 13n); rather, excitability is gated by their integration.”

10. In Figure 3c, 3d, and 3f, if the Agg⁻ and Agg⁺ cells are combined, do they differ from Pre group? If not, it questions the claim that food deprivation changes the property of MPOA cells. Instead, the behavior output provides an axis to separate the preexisting heterogeneous responses of MPOA cells across animals.

We thank the referee for this important point and regret not including these analyses in the original manuscript. While combining all Post–food deprivation neurons (Agg⁺ and Agg⁻) for comparison with the Pre group might seem intuitive, this approach would be confounded by the uneven distribution of estrous cycle stages across subgroups. To better isolate the specific effects of food deprivation—*independent of estrous state*—we have now performed two complementary analyses, which we describe below.

First, given that ~60% of food-deprived animals showed pup-directed aggression (Agg⁺) and ~40% did not (Agg⁻) (Fig. 1b), we constructed a weighted composite Post group by sampling 60% of neurons from Agg⁺ mice and 40% from Agg⁻ mice, and compared this to neurons from a Pre group sampled to match the estrous cycle composition of the Post group. The weighted Post group exhibited decreased voltage sag amplitude and reduced baseline firing frequency, along with a non-significant reduction in excitability, consistent with the direction of our original findings (see Referee Fig. 5a–g). These data have been added to the manuscript as Extended Data Fig. 9a–g.

Referee Fig. 5. Biophysical properties of MPOA neurons were compared between a Pre group (sampled to match the estrous cycle distribution of the Post group) and a weighted Post group (60% Agg⁺, 40% Agg⁻; **a–g**), and between Pre and Post groups restricted to mice in diestrus (**h–n**). Parameters shown include voltage sag amplitude (**b, i**; 108, 24 cells from $n = 39$, 9 mice and 38, 9 cells from $n = 11$, 3 mice), input resistance (**c, j**; 109, 24 cells from $n = 39$, 9 mice and 38, 9 cells from $n = 11$, 3 mice), baseline firing frequency (**d, k** and **e, l**; 132, 25 cells from $n = 43$, 9 mice and 47, 10 cells from $n = 12$, 3 mice), percentage of silent neurons at resting membrane potential (**f, m**; 136, 27 cells from $n = 43$, 9 mice and 47, 12 cells from $n = 12$, 3 mice), and number of action potentials evoked by injected somatic current (**g, n**; 36, 21 cells from $n = 19$, 8 mice and 14, 9 cells from $n = 4$, 3 mice). *U* test in **b–e, i–l**. Fisher’s exact test in **f, m**. Mixed linear model with cell ID as random effect in **g, n**.

Second, we restricted our analysis to animals in the same estrous cycle stage—diestrus—which was the most common stage across groups. Even within diestrus alone, Post neurons showed decreased voltage sag amplitude and increased input resistance relative to Pre neurons. Other changes—such as baseline firing rate, excitability, and proportion of silent neurons—exhibited similar trends, although some comparisons did not reach statistical significance, likely due to reduced sample size (Referee Fig. 5h–n; now Extended Data Fig. 9h–n).

Together, these results indicate that food deprivation alters MPOA neuron physiology independently of estrous cycle stage and behavioural outcome. We now refer to these findings in the main text (page 7, lines 19–26):

“To test whether food deprivation alters MPOA neuron properties independently of behavioural outcome or estrous cycle stage, we compared the Pre group to a weighted Post group (60% Agg⁺, 40% Agg⁻) and to stage-matched animals in diestrus. In both analyses, food deprivation was associated with reduced voltage sag amplitude, increased input resistance, and a trend toward lower neuronal excitability (Extended Data Fig. 9), indicating that food deprivation alone affects MPOA neuronal physiology.”

11. *The results presented make a compelling argument that the arcuate-to-MPOA projection is not GABAergic in this context. However, AgRP seems to have largely similar effects as NPY on*

the properties of MPOA neurons (Extended Data Fig. 7). AgRP seems to reduce the baseline firing rate of MPOA neurons (though there isn't a statistically significant effect). Importantly, AgRP also reduces the sag to the same magnitude as NPY. If you injected a retrograde shRNA against AgRP into the MPOA, would you also expect a reduction in infanticide? The authors use NPY antagonists to rescue the excitability of MPOA cells. To concretely show that this effect is mediated by NPY rather than AgRP, the authors should investigate the effectiveness of MC3R/MC4R antagonism and AgRP knockdown.

Thank you for this suggestion. We have now examined both the biophysical properties of MPOA neurons and pup-directed behaviour following food deprivation in mice with projection-specific *Agrp* knockdown (see Referee Fig. 6; now included in Extended Data Fig. 10, previously Extended Data Fig. 7). This manipulation did not significantly alter voltage sag amplitude (although a trend was observed), baseline firing frequency, the proportion of silent neurons, or neuronal excitability (Extended Data Fig. 10r–w). Furthermore, this manipulation did not significantly affect pup-directed aggression (Extended Data Fig. 10z). These findings support our conclusion that release of NPY—rather than AgRP or GABA—from Arc^{AgRP}→MPOA projections mediates the hunger-induced shift to aggression.

We now refer to these new findings in the main text (page 9, lines 26–31):

“To test whether AgRP contributes to this effect, we performed projection-specific *Agrp* knockdown. This manipulation did not alter pup-directed aggression or key MPOA properties (sag amplitude, baseline activity, or excitability; Extended Data Fig. 10q–z), indicating that NPY—rather than AgRP—release from Arc^{AgRP}→MPOA projections promotes the hunger-evoked switch to aggression.”

We have also updated the Methods section accordingly (Surgical procedures > Gene knockdown).

Further, as the referee points out, application of NPY receptor antagonists increases the excitability of MPOA neurons in Agg⁺ mice (Extended Data Fig. 8i). This result is somewhat surprising, because antagonists were added to acute brain slices of Agg⁺ animals *after* 6-h food deprivation, i.e., *after* NPY release has occurred. We would therefore expect only minor effects of antagonizing NPY receptors at this stage, and this is indeed what we observe for all other tested parameters (Extended Data Fig. 8e–h). We now discuss this observation in the revised manuscript (page 14, lines 16–19):

“[...] Along with shorter aggression latencies after prolonged food deprivation (Extended Data Fig. 1a), this suggests that NPY release progressively increases during food deprivation. Consistent with this, addition of NPY receptor antagonists after food deprivation has variable biophysical effects on MPOA neurons (Extended Data Fig. 8e–i).”

Referee Fig. 6. **a**, *Agrp* knockdown (KD-A) in $Arc^{AgRP} \rightarrow MPOA$ projections. **b**, Example images of *Agrp* KD and control. Scale bar, 20 μm . **c**, *Agrp* KD efficiency ($n = 4$, 4 mice). **d**, Voltage sag amplitude (20, 30 cells from $n = 6$, 4 mice). **e**, Baseline firing (20, 30 cells from $n = 6$, 4 mice). **f**, Percentage of silent neurons at resting membrane potential (20, 30 cells from $n = 6$, 4 mice). **g**, Action potentials per injected current (17, 16 cells from $n = 6$, 3 mice). **h**, Input resistance (19, 26 cells from $n = 6$, 4 mice). **i**, Resting membrane potential (19, 26 cells from $n = 6$, 4 mice). **j**, Cumulative incidence of aggression (KD-A, ctrl; $n = 11$, 13 mice). *U* test in **c–e**, **h**, **i**. Fisher’s exact test in **f**. Mixed linear model with cell ID as random effect in **g**. Log-rank test in **j**.

Since food-deprivation-induced AgRP release is expected to occur even more slowly than NPY release (Krashes et al., 2013, PMID 24093681), this limitation would apply even more strongly to using MC3R/MC4R antagonists on brain slices from Agg⁺ mice. Projection-specific *Agrp* knockdown—which has no effect on excitability (see above)—is therefore a more appropriate test of whether AgRP release contributes to effects of food deprivation on pup interactions.

12. Please examine the correlation between sag amplitude and maximum firing of the IF curve (basically parameters shown in 3f and 3d). The rationale is that HCN level, which is reflected in the sag, alters the excitability of the MPOA cells and, hence, the tendency of the animal to show infanticide. Given that the cell excitability could be affected by many ion channels, whether the sag and excitability are indeed correlated should be reported.

Thank you for this helpful suggestion. We have now conducted the requested analysis and indeed find a significant correlation between the mean voltage sag amplitude and both the mean number of evoked action potentials (Referee Fig. 7a) and the maximal number evoked action potentials (Referee Fig. 7b). These panels have now been added to Extended Data Fig. 6 and we mention this in the revised manuscript (page 7, lines 14–15). These findings support a causal role for HCN channel activity in regulating the excitability of MPOA neurons and further strengthen our mechanistic interpretation.

Referee Fig. 7. a, b, Correlation between mean voltage sag amplitude and mean number of evoked action potentials per animal (**a**, linear regression, $R^2 = 0.272$; $P = 0.026$, $n = 18$) correlation between voltage sag amplitude and number of evoked action potentials at the maximal injected current (**b**, $P = 0.011$, mixed linear model with mouse ID as random effect, 50 cells from $n = 18$ mice).

13. *The trends of sag magnitude across the estrous cycle nicely match the behavioral data. However, the HCN transcript levels do not. The authors should discuss this discrepancy – if HCN1 and HCN2 RNA are lowest in diestrus, why is the sag magnitude comparable to when RNA expression is highest (in estrus)?*

Thanks for pointing this out. Several non-mutually exclusive factors may underlie the apparent discrepancy between *Hcn* transcript levels and sag amplitude during diestrus:

First, diestrus is the longest estrus cycle phase (>48 h), compared to proestrus (~12 h), estrus (~16-24 h) and metestrus (~8-24 h). During diestrus, the P4/E2 ratio gradually declines, reflecting falling P4 levels and a gradual increase in E2 (Miller et al., 2014, PMID 24478756). Since RNAscope *in situ* hybridisations (used to assess *Hcn* transcript levels) and electrophysiological recordings (used to measure sag amplitude) were carried out in separate cohorts at different times, it is possible that the discrepancy may reflect the temporal dynamics within diestrus and a nonlinear or lagged relationship between hormonal state, transcript levels, and functional output.

Second, mRNA levels may not always correspond linearly to protein expression or channel density at the membrane, due to post-transcriptional regulation, trafficking, and turnover. Such mechanisms could contribute to the observed dissociation between transcript abundance and functional readout.

Third, cAMP directly modulates HCN function by binding to the cyclic nucleotide-binding domain and facilitating channel opening (Wainger et al., 2001, PMID 11459060). Importantly, hypothalamic cAMP levels rise during diestrus and peak in late diestrus (Zubin & Taleisnik, 1983,

PMID 6311328). This elevated cAMP could enhance I_h current amplitude even in the face of reduced *Hcn* transcript levels, providing a functional explanation for the maintained sag during diestrus.

We now discuss these possibilities in the revised manuscript (page 14, lines 9–13):

“This discrepancy may arise from differences in the timing of data collection within the prolonged (~2-day) diestrus phase, during which the P4/E2 ratio gradually declines, or from post-transcriptional modulation. For instance, hypothalamic cAMP levels fluctuate across the estrous cycle⁷⁶, and increasing cAMP levels during diestrus may enhance HCN channel function despite reduced *Hcn* expression⁷⁷.”

14. *The behavioral effects of HCN blockade are quite dramatic (Fig. 3v). The authors should provide additional behavioral characterization regarding this experiment.*

We appreciate the referee’s suggestion and have now performed additional behavioural characterisation of animals receiving ZD-7288 (ZD) infusions into the MPOA. Notably, we find that these animals exhibit significantly shorter latencies to attack pups compared to food-deprived Agg⁺ mice (79.5 s vs 219.4 s; Referee Fig. 8a). This increased propensity for pup-directed aggression is also reflected in a steeper cumulative incidence of aggression in ZD-treated mice (Referee Fig. 8b). These new data are now included as Extended Data Fig. 12c, d (previously Extended Data Fig. 9) and referenced in the main text (page 10, lines 19–21).

Referee Fig. 8. a, Attack latency of animals with bilateral infusion of HCN blocker into MPOA (ZD, n = 4 mice) or after food deprivation (FD, n = 39 mice). **b**, Cumulative incidence of aggression after ZD-infusion or food deprivation (FD, n = 4, 39 mice).

15. *The author states (page 9) “this difference was present before food deprivation and therefore likely reflects the influence of estrous state (Pre+ vs Pre-).” Please confirm whether the difference in Agg- and Agg+ animals before food deprivation indeed reflects the difference in estrous state vs. individual difference unrelated to the estrous cycle. Basically, for Pre+ and Pre-, does the animals’ estrous state distribution differ?*

This is an important point which we have addressed in our response to referee 3 (point 4) below: Among the six recorded females, one was in proestrus, two in estrus, two in metestrus, and one in diestrus. Both mice in estrus were non-aggressive (Agg⁻), whereas all others were Agg⁺,

consistent with our model (Fig. 3w) in which estrous stage interacts with food deprivation to shape behavioural outcomes. This pattern suggests that the low $\frac{P4}{E2}$ ratio characteristic of estrus biases animals toward an Agg⁻ phenotype, while the higher ratio during metestrus promotes pup-directed aggression.

16. Based on Figure 4b, it appears that food deprivation does not change the response of MPOA cells at all. The difference between Agg⁺ and Agg⁻ are all preexisting. Again, this questions whether food deprivation directly influences MPOA cell activity.

Thank you for raising this important point, and apologies for the lack of clarity in the original manuscript. This comment also relates to the earlier concern (point 10) regarding the effects of food deprivation on the biophysical properties of MPOA neurons.

To clarify: although food deprivation does influence MPOA neuron physiology, these effects are not readily apparent in the miniscope imaging data shown in Fig. 4c. Specifically, as we show *ex vivo* (Referee Fig. 5, now Extended Data Fig. 9), food deprivation alters intrinsic properties of MPOA neurons: it reduces baseline firing frequency, increases the proportion of silent cells (albeit non-significantly), and decreases neuronal excitability. These findings indicate that food deprivation does directly affect MPOA neuronal function.

However, these baseline changes are not detectable in our *in vivo* 1-photon miniscope recordings (Fig. 4c), likely due to the technical limitations of this method. GCaMP-based calcium imaging, particularly via 1-photon miniscopes, has limited sensitivity for detecting low or tonic activity, and may not reliably capture subtle shifts in *baseline* firing (Zhou et al., 2018, PMID 29469809).

In contrast, when we assess *evoked* activity—specifically, responses to pup sniffing—we do detect a food deprivation effect, but only in the Agg⁺ group. In these mice, the proportion of MPOA neurons tuned (i.e., activated or inhibited, see point 17 below) to pup sniffing is significantly reduced after food deprivation (Pre⁺ vs. Agg⁺, $P = 0.03$, paired *t*-test); this effect is not observed in Agg⁻ mice ($P = 0.50$) (Referee Fig. 9a, b, now Extended Data Fig. 16a, b).

These data suggest that while we cannot detect the effect of food deprivation on baseline MPOA activity *in vivo*, food deprivation modulates *stimulus-evoked responsiveness*—particularly in animals predisposed to pup-directed aggression. This aligns with our broader conclusion that internal state (hunger) interacts with behavioural disposition (aggression) to shape MPOA neuronal responses to social stimuli.

We now reference these data in the text (page 12, lines 2–8):

“While slice electrophysiology confirms that food deprivation decreases baseline activity of MPOA neurons, we did not detect this effect *in vivo* (Pre⁺ vs Agg⁺, Pre⁻ vs Agg⁻; Fig.

4c)—likely due to the limited sensitivity of 1-photon calcium imaging. In Agg⁺ mice, however, MPOA responses to pup chemoinvestigation and grooming were suppressed, potentially reflecting reduced excitability (Fig. 4d and Extended Data Fig. 16a, f). In addition, the absolute tuning of MPOA neurons—defined as the magnitude of activation or inhibition—was diminished in Agg⁺ mice (Fig. 4e and Extended Data Fig. 16e; see Methods).”

Referee Fig. 9. a, b, Percentage of MPOA neurons tuned (activated or inhibited) to pup sniffing in Agg⁺ (**a**, n = 4 mice) and Agg⁻ (**b**, n = 2 mice) mice before (Pre) and after food deprivation. Paired *t*-test in **a**, **b**.

17. I found the absolute tuning index of each cell’s response during behavior (Fig. 4e) confusing. Does this differ at all from the average response during each behavior?

We apologise for this not being clear. The absolute tuning index measures how strongly each detected cell’s activity deviates from baseline during a behavioural event, incorporating both positive and negative activity changes. This index accounts for variability by considering the standard deviation of both the baseline and activity period. We have now improved the explanation of the ‘absolute tuning index’ metric in the Methods section and relabelled the respective section (Quantification > Evoked activity and absolute tuning index).

18. The authors claim that the activity of MPOA cells in Agg⁺ females correlates more closely with an aggressive state, rather than the specific action of aggression itself. I think this would largely depend on how the authors define a state. Presumably, an aggressive state would imply a persistent change in neural dynamics that lasts beyond a specific behavior. Indeed, the authors indicate that the initial aggression episodes shift the neural activity into a distinct representation. To really make this claim concretely, the authors would need to remove the pup following the initial aggressive encounter(s) or prevent the adult from attacking. If the neural activity is indeed state-encoding, the dynamics should persist independent of the consummation of aggression.

We appreciate the referee’s thoughtful comments regarding the definition of an aggressive state and the necessity of testing whether MPOA neuronal activity persists independently of attacking behaviour itself. We indeed conducted these experiments as you suggest: after an initial aggressive encounter, the pup was removed and a new one subsequently introduced to capture additional instances of pup-directed aggression (now clarified in Methods > Behavioural profiling > Pup-directed behaviour assay).

Our existing analysis of neural activity across entire recording sessions found that the activity of most MPOA neurons correlates more strongly with the cumulative distribution function (CDF) of aggression ('State') in general rather than with individual attack events ('Behaviour') (see Fig. 4h and Extended Data Fig. 17c; previously Extended Data Fig. 14b). This suggests that MPOA neurons encode a persistent aggression-associated state rather than discrete aggressive actions. The below heatmap (Referee Fig. 10) shows neuronal responses from an example trial of an Agg⁺ animal aligned with episodes of pup-directed aggression. As you can appreciate, neural activity remains quite stable throughout the trial after the onset of aggression. This can also be seen in Fig. 4h.

Referee Fig. 10. Example heatmap of neuronal responses from Agg⁺ animal, sorted based on correlation of activity with cumulative distribution of aggression. Aggression episodes highlighted.

We have now performed additional analyses to more directly demonstrate the emergence of a distinct neural state after aggression has occurred.

First, we extracted the period of neural activity following the first aggression episode and analysed the signal amplitude distribution to compare neural selectivity between aggression and non-aggression periods. This approach—similar to how we previously calculated the selectivity of MPOA neurons during investigation of pups vs other stimuli (Extended Data Fig. 16m–o; previously Extended Data Fig. 13i–k; and see Methods > Processing and analysis of *in vivo* imaging data > Selectivity analysis)—allows us to quantify the extent to which MPOA neurons preferentially encode aggression: if a neuron participates in the encoding of a persistent aggression state, its signal amplitude distribution should be similar between aggressive and non-aggressive episodes. Across a population of neurons, this would result in a normally distributed selectivity centred around 0.5. Conversely, if a neuron encodes aggressive actions, its signal amplitude should be higher during aggression periods and lower during non-aggression periods. This would result in a skewed selectivity distribution. Our observations were consistent with the first scenario ('Aggression', Referee Fig. 11, now Extended Data Fig. 17b; kurtosis test: $P = 0.138$). In addition, we extracted the period of neural activity prior to any pup-directed aggression and performed the same analysis for pup sniffing vs non-pup-sniffing and pup grooming vs non-pup-

grooming periods. Unlike aggression, these corresponding distributions were skewed towards both extremes (Kurtosis test: $P(\text{sniff}) = 2.7 \times 10^{-10}$, $P(\text{groom}) = 1.8 \times 10^{-21}$), suggesting the presence of neuronal subpopulations that are either activated or inhibited by pup sniffing and -grooming behaviour, respectively.

Referee Fig. 11. Selectivity of MPOA neurons for the indicated behaviours (243 neurons from $n = 5$ mice). Neurons exclusively active during the indicated behaviour have a selectivity index of 1, those inhibited during this behaviour an index of 0, and non-selective neurons an index of 0.5. Kurtosis test.

Second, we had previously used a Hidden Markov Model (HMM) to infer latent neural states associated with aggression and identified an HMM state that captured 95% of aggression-associated neural activity episodes (Fig. 4j, k). To further characterize the relationship between this HMM state and aggression, we examined how its presence correlates with two aggression-related measures: (1) ‘State’—this variable reflects a sustained neural state of aggression and remains at a value of 1 throughout the recording session after the first aggression episode. (2) ‘Behav’—this variable is 1 when aggression occurs (i.e., during an attack) and 0 otherwise. We find that the HMM state correlated more strongly with ‘State’ than ‘Behav’ (paired t -test, $P = 0.0077$), suggesting that it represents a persistent neural state of aggression rather than being tied to individual attack events (Referee Fig. 12a, now Extended Data Fig. 17g).

Furthermore, when the same analysis was applied to other behaviours—pup sniffing and pup grooming—(i.e., calculating the correlation between each behaviour’s state representation and its corresponding inferred HMM state), much weaker correlations were observed (Referee Fig. 12b, now Extended Data Fig. 17h; Sniff vs Attack: $P = 0.002$; Groom vs Attack: $P = 0.0099$). This suggests that in contrast to pup grooming and sniffing, aggression is associated with a distinct neural state in MPOA neurons.

Referee Fig. 12. a, Correlation between aggression-associated HMM state and ‘State’ or ‘Behaviour’ variable (n = 5 mice). **b**, Correlation between pup sniffing-, pup grooming- or aggression-associated inferred HMM states and occurrence of the corresponding behaviour (n = 5 mice). Paired *t*-tests in **a**, **b**.

We have now included this analysis in Extended Data Fig. 17 and have updated the Methods section accordingly (‘Processing and analysis of *in vivo* imaging data > HMM analysis’ and ‘> Selectivity analysis’).

In summary, our data indicate that after an initial aggression episode, MPOA activity shifts into a persistent, distinct, aggression-related neural state during which MPOA activity is not time-locked to further aggressive actions. We have now updated the Results section accordingly (page 12, lines 16–22; page 12, lines 25–29):

“To test whether a persistent neural state emerged following aggression, we quantified aggression selectivity across post-aggression activity epochs using a ROC-based approach (see Methods). In contrast to grooming and sniffing—which showed strongly skewed selectivity distributions consistent with transient, event-linked encoding—aggression-related selectivity values were centred around 0.5. This pattern suggests the presence of a sustained, population-level activity state rather than time-locked responses (Extended Data Fig. 17b).”

[...]

“[...] the inferred HMM state more strongly tracked a sustained aggressive state than discrete attack events (Extended Data Fig. 17g). In contrast, HMM states associated with pup sniffing and grooming showed weaker correspondence to those behaviours (Extended Data Fig. 17h), suggesting that aggression is linked to a distinct and persistent neural state in MPOA neurons.”

19. There needs to be more information about the PCA and classifier training in the methods. For example, exactly how was the PCA matrix constructed, and what percentage of variance was explained by the first two PCs? There is a dearth of information about how the classifiers were trained and validated. In particular, I cannot find any mention of the Hidden Markov Model in [the methods].

Thanks for bringing this to our attention. We have added the following sections to the revised manuscript (Methods > Quantification and data analysis):

Principal Component Analysis (PCA)

PCA was used to reduce the dimensionality of the neural data and identify the primary sources of variance in each recorded pup interaction session. Before applying PCA, the activity of each recorded neuron was standardized to ensure comparability across different neurons. The standardised activity for each neuron in each session was computed as:

$$x'_i = \frac{x_i - \mu_i}{\sigma_i}$$

where x_i represents the activity of the i -th neuron, μ_i is its mean activity across the entire dataset, and σ_i is its standard deviation. To quantify relationships between neurons, we computed the covariance matrix \mathbf{C} of the standardized data:

$$\mathbf{C} = \frac{1}{N-1} \sum_{i=1}^N (x_i - \bar{x})(x_i - \bar{x})^T$$

where N is the total number of neurons, and \bar{x} is the mean activity vector across all neurons. The eigenvalues and eigenvectors of \mathbf{C} were then computed, with each eigenvector v_j representing a principal component (PC) and its corresponding eigenvalue λ_j indicating the variance explained by that component. The principal components were ranked by their eigenvalues, with higher-ranked components capturing more variance. Recording sessions were included in the analysis only if the first two PCs accounted for at least 70% of the total variance. These two PCs were then used to project the neural population activity into a lower-dimensional space for visualisation.

Principal Component distance calculation

For the calculation of principal component (PC) distances, activity episodes of 5 s after behavioural onset were extracted from each neuron. All episodes were standardized prior to analysis. To quantify the similarity between two neural activity episodes, \mathbf{a} and \mathbf{b} , in the k -dimensional principal component (PC) space ($k = 2$), we computed the pointwise Euclidean distance at each time point t :

$$d(\mathbf{a}_t, \mathbf{b}_t) = \sqrt{\sum_{i=1}^k (a_{t,i} - b_{t,i})^2}$$

where:

- $\mathbf{a}_t = (a_{t,1}, a_{t,2}, \dots, a_{t,k})$ and $\mathbf{b}_t = (b_{t,1}, b_{t,2}, \dots, b_{t,k})$ are the projections of the episodes onto the reduced k -dimensional PC space at time t .
- $a_{t,1}$ and $b_{t,1}$ represent the i -th PC of each episode at time t .
- k is the number of retained PCs (chosen to account for 90% of the variance).

To obtain the total distance between the two episodes, the pointwise Euclidean distances were summed across all time points:

$$D(\mathbf{a}, \mathbf{b}) = \sqrt{\sum_{t=1}^T \sum_{i=1}^k (a_{t,i} - b_{t,i})^2}$$

where T is the total number of time points in the episode. This total distance, $D(\mathbf{a}, \mathbf{b})$, serves as a similarity measure of neural population activity between conditions (e.g., Pre and Post), with larger values indicating greater divergence in activity patterns.

Multiclass SVM classification

A Support Vector Machine (SVM) classifier was used to categorize behavioural states (pup sniff, pup groom, lick-aggressive, and late aggression) based on: (1) Raw neural data, (2) PCA-reduced neural data, and (3) shuffled neural data (control). All behavioural episodes of >2 s were selected. Behavioural labels were assigned unique numeric identifiers using LabelEncoder, while the neural activity data served as the feature matrix (\mathbf{X}), and the behavioural labels as the target variable (\mathbf{y}). To address class imbalances, the Synthetic Minority Oversampling Technique (SMOTE) was applied. The k -neighbour parameter was dynamically adjusted based on the size of the smallest class. If a class contained fewer than two samples, SMOTE was not applied, and the dataset was excluded from the analysis. Classifier performance was evaluated over 50 iterations. The dataset was split into training and test sets (split by episode number rather than by frame number) using stratification to preserve class distributions. To avoid data leakage, splits were made based on episode numbers rather than frame numbers. The SVM classifier was implemented using SVC from scikit-learn with default parameters. After training, predictions were generated for the test set, and accuracy was computed as:

$$\text{Accuracy} = \frac{\sum_{i=1}^n \mathbb{I}(y_{pred,i} = y_{test,i})}{n}$$

where:

- y_{pred} and y_{test} are the predicted and true labels, respectively.
- $\mathbb{I}(\cdot)$ is the indicator function, which returns 1 if the predicted and true labels match, and 0 otherwise.
- n is the total number of test samples.

The accuracy of the SVM trained on raw neural data, PCA-reduced neural data, and shuffled data was then compared. To assess classification performance across behavioural states, a confusion matrix was computed. The average accuracy and average confusion matrix were obtained by summing the confusion matrices over all iterations and then normalizing them row-wise as percentages:

$$\text{Normalised } CM_{ij} = \frac{CM_{ij}}{\sum_{j=1}^m CM_{ij}} \times 100$$

where:

- CM_{ij} represents the number of times a sample from class i was classified as class j .
- m is the total number of classes.

Hidden Markov Model (HMM) analysis

To determine the optimal number of states for the Hidden Markov Model (HMM), we first explored a range of state values and assessed model performance using the Evidence Lower Bound (ELBO). ELBO, a standard metric in variational inference, measures model fit, with higher values indicating better performance. The optimal state number was identified by locating the turning point in the ELBO curve, beyond which additional states provided diminishing improvements. This optimal value (here: 5, see Extended Data Fig. 17f) was then used for all subsequent analyses. The HMM was implemented using the `ssm` package (<https://github.com/lindermanlab/ssm>), with a Gaussian observation model. This framework applies Bayesian learning and inference to state-space models and is well suited for analysing sequential neural data (Nair et al., 2023, PMID 36608653). The HMM was trained on neural recordings using the Expectation-Maximization (EM) algorithm for 50 iterations, with initial state assignments determined via k -means clustering. To assess the relationship between neural states and behaviour, we constructed a binary matrix for each behaviour, marking timestamps where the behaviour occurred as 1 and all other timestamps as 0. We then identified the state most frequently associated with each behaviour and computed the conditional probability of that behaviour occurring within the inferred state. This probability quantifies how strongly a given neural state is linked to specific behavioural patterns.

Selectivity analysis

To assess the selectivity of individual neurons for a particular stimulus, social and non-social stimuli were presented sequentially to recorded animals. Selectivity for pups versus other stimuli was quantified using a choice probability approach (Remedios et al., 2017, PMID 29052632; Karigo et al., 2021, PMID 33268894). For each neuron, fluorescence signals ($\Delta F/F_0$) recorded during pairs of chemosensory investigation behaviours (e.g., pup vs intruder investigation) were used to estimate how reliably the two behaviours could be distinguished based on their $\Delta F/F_0$

distributions. Specifically, $\Delta F/F_0$ values were extracted for each neuron's activity during behaviour α (e.g., attack) and behaviour β (e.g., sniff). These distributions were plotted as paired histograms and cumulative distribution functions (CDFs). A Receiver Operating Characteristic (ROC) curve was then generated, with the CDF of pup sniffing on the x-axis. The selectivity index was computed as:

$$\text{Selectivity Index} = 1 - \text{AUC}_{\text{ROC}}$$

Where AUC_{ROC} is the area under the ROC curve. Neurons exclusively active during pup sniffing have a selectivity index of 1, those exclusively active during investigation of another stimulus an index of 0, and non-selective neurons an index of 0.5. To minimize the effects of gradual desensitisation, only the first chemosensory investigation episode for each stimulus was used. For aggression-related analyses (Extended Data Fig. 17b), we adapted this approach to test for persistent neural encoding. Following the first pup-directed aggression episode, we extracted the $\Delta F/F_0$ values for each neuron during aggression periods and non-aggression periods. Selectivity indices were computed as above. To assess population-level structure, we analysed the distribution of selectivity indices across neurons: a normal distribution centred at 0.5 indicates persistent state encoding, while skewed distributions suggest selective responses to discrete behavioural events. For comparison, we applied the same analysis to pup sniffing and pup grooming behaviours using activity recorded prior to the first aggression episode. Selectivity distributions for sniffing and grooming were significantly skewed, suggesting event-related encoding by discrete neuronal subpopulations.

Referee #2 (Remarks to the Author):

This work identified a dual-gating neural mechanism controlling pup-directed aggression in females in a hunger and estrous state-dependent manner. The authors found that mild starvation induces pup-directed aggression in females that are otherwise non-infanticidal. AgRP neurons in the arcuate nucleus, which are known as hunger-sensing neurons, project to the MPOA, a region well-known for parenting behavior control, and induce pup-directed aggression through NPY release. The estrous cycle defines attack probability and HCN channel expression levels in MPOA neurons, with NPY release reducing HCN currents, potentially through reduction of cAMP that could bind to HCN channels. In vivo miniscope imaging confirmed attenuated average neural activity in the MPOA, while also identifying a population of neurons activated during pup-directed aggression, suggesting a potential novel role of MPOA neurons in controlling pup-directed aggression.

Overall, this work is well conceptualized and executed, addressing an important mechanism of how diverse internal states interact to support optimal and flexible behavioral expression.

Many thanks for your positive assessment of our work, and the following, very helpful suggestions.

I have several comments and suggestions to improve clarity for readers:

1. The authors showed that Agg+ mice have more silent neurons than Agg- mice, MPOA neurons in Agg+ mice show low excitability upon current injection, and HCN blocker or NPY infusion increases the percentage of silent neurons in MPOA. However, miniscope data shows that a particular population of neurons exhibits strong activation during pup-directed aggression. Are these neurons silent during non-aggressive trials (Pre-trial in Agg+ mice) or active during different behaviors? This partially contradicts the model presented in Fig. 3w. How do the authors explain this discrepancy?

We appreciate the referee's question regarding the apparent discrepancy between the overall silencing of MPOA neurons in Agg+ mice and the activation of a subset of neurons during pup-directed aggression.

Multiple lines of evidence suggest that pup-directed aggression arises in part from a broad reduction in MPOA excitability and an increase in the proportion of silent neurons: (1) Baseline activity is significantly lower in Agg+ mice compared to Agg- mice (Fig. 4c), consistent with *ex vivo* evidence of decreased excitability (Fig. 3b, c); (2) Behaviourally evoked responses—particularly to pup chemoinvestigation and grooming—are reduced in Agg+ mice (Fig. 4d, Extended Data Fig. 16e, f; previously Extended Data Fig. 13a, b), likely reflecting lowered responsiveness due to reduced excitability (Fig. 3d); (3) The absolute magnitude of tuning (activation or inhibition) is diminished in Agg+ mice (Fig. 4e, Extended Data Fig. 16a, b).

However, this global shift toward quiescence (see Fig. 3w) does not preclude the recruitment of a subset of neurons during specific behaviours. As shown in Fig. 4h, a minority of MPOA neurons—an average of 29.5%—exhibit positive correlations (>0.2) with the aggression state. We have now analysed the behavioural tuning of these neurons before and after food deprivation. This analysis—described in more detail in our response to your point 7 below (Referee Fig. 19), reveals that aggression-tuned neurons are often also responsive to pup grooming in the pre-aggression state.

These findings suggest that a subset of previously active, pup-responsive neurons may be re-engaged in the context of aggression, potentially contributing to the persistent aggression-related state we observe. We have now clarified this point in the main text (page 13, lines 6–9):

“Of note, MPOA neurons tuned (i.e., activated or inhibited) to aggression were often also responsive to pup grooming, both before and after food deprivation (Extended Data Fig. 19), suggesting that affiliative and aggressive behaviours recruit overlapping neuronal populations.”

Further supporting a link between MPOA silencing and pup-directed aggression, we find that lower baseline MPOA activity predicts shorter latencies to attack. Specifically, baseline activity decreases following each aggression episode (Referee Fig. 13a), and the extent of this decline correlates with the latency to the next attack (Referee Fig. 13b). This suggests that the aggression state is accompanied by progressive MPOA inhibition, with neurons becoming increasingly suppressed across repeated episodes. This pattern closely mirrors the behavioural dynamics observed during *in vivo* recordings, in which attack latency decreases with successive bouts of aggression (Referee Fig. 13c). We have added these data as Extended Data Fig. 17i–k and reference them in the main text (page 12, line 29–33).

Referee Fig. 13. a, Mean baseline MPOA neuronal activity during the 3 s preceding attack onset across successive attack episodes ($n = 4$ mice). **b**, Correlation between mean baseline activity and latency of the subsequent attack episode (12 episodes from $n = 4$ mice; linear regression, $R^2 = 0.441$, $P = 0.011$). **c**, Attack latency across consecutive attack episodes ($n = 4, 3, 4, 2$ mice). One-way ANOVA in **a**, **c**.

2. Since pup-directed aggression induced by mild hunger is a novel finding, it would be helpful to describe the behaviors of both Agg+ and Agg- mice in more detail:

- Do Agg+ mice ever retrieve pups?

- Do Agg- mice always show retrieval/parenting behavior or sometimes ignore pups?

- Do Agg+ and Agg- mice show similar levels of non-aggressive pup-directed behaviors (sniffing, grooming etc.)?

Adding behavioral quantifications and example ethograms in supplemental materials would be helpful. The interpretation of Extended Figure 2 may vary depending on these behavioral patterns.

We completely agree with this point and have analysed and described the behaviours of Agg+ and Agg- mice in more detail (see below).

Do Agg+ mice ever retrieve pups?

We never observed pup retrieval in Agg+ mice (the same applies to nest building and crouching above pups). The behaviours of Agg+ and Agg- animals are easily distinguishable: in Agg+ animals, the behavioural sequence begins with chemoinvestigation, followed by pup grooming (Referee Fig. 14b, c). After an extended grooming period, the behaviour transitions to aggression, including biting and aggressive carrying of pups around the cage (Referee Fig. 14b, c; see Methods). This figure has now been added to the revised manuscript as Extended Data Fig. 2. We have also updated the Methods section accordingly (Behavioural profiling > Pup-directed behaviour assay).

Referee Fig. 14. Pup-directed behaviours in Agg⁺ and Agg⁻ mice. a–f, Representative behavioural raster plots (a, c, e) and corresponding behavioural state transition diagrams (b, d, f; see Methods) from individual mice classified as Agg⁺, parental Agg⁻, or pup-ignoring Agg⁻ during pup interactions. Chemo, chemoinvestigation, retri., pup retrieval, in nest: in nest with pups; in empty nest: in next without pups, P_T , transition probability.

Do Agg⁻ mice always show retrieval/parenting behavior or sometimes ignore pups?

After an initial chemoinvestigation and grooming phase, Agg⁻ animals perform non-aggressive behaviours such as pup retrieval, nest building, rearing and digging (Referee Fig. 14d–g). Agg⁻ animals can be further classified as ‘parental’ or ‘ignoring’ based on whether initial chemoinvestigation and grooming are followed by parental behaviour components such as pup retrieval, spending time with the pups in the nest, and crouching. ‘Parental’ animals briefly groom pups, followed by retrieval. Once in the nest, the animals remain with the pups, engaging in crouching above them, grooming, and occasionally nest building (Referee Fig. 14c, d). In contrast, ‘ignoring’ animals only briefly investigate and groom pups, and subsequently engage in other, non-pup-related, behaviours, such as rearing and digging (Referee Fig. 14e, f). This information has now been added to the Methods (Behavioural profiling > Pup-directed behaviour assay).

Do Agg⁺ and Agg⁻ mice show similar levels of non-aggressive pup-directed behaviors (sniffing, grooming etc.)?

Agg⁺ and Agg⁻ animals did not show significant differences in sniffing or grooming of pups (Referee Fig. 15). Note that because behavioural sessions had to be terminated early for Agg⁺ mice, only instances of sniffing and grooming that occurred during the first ~2 min after pup introduction were analysed. This panel has now been added as Extended Data Fig. 1e in the revised manuscript.

3. What accounts for the differences between Agg⁺ and Agg⁻ mice in the same estrous stages? Does it depend on individual differences in P4/E2 ratio or do other factors contribute?

While we can currently only speculate about the underlying causes of behavioural divergence between Agg⁺ and Agg⁻ mice within the same estrous stage, several factors are likely to contribute. First, mice classified within the same estrous stage may differ significantly in circulating hormone levels, including the ratio of progesterone to estradiol (P4/E2), which may influence behavioural output. Notably, hormone concentrations can shift rapidly even within a given stage; for example, P4 levels differ between early and late proestrus, potentially driving divergent behavioural responses. Second, individual variation in hormone receptor expression may also play a role. For example, prior studies have shown that variability in *Esr1* expression in the MPOA correlates with differences in parental behaviour in lactating rats (Champagne et al., 2003, PMID 12959970). It is therefore plausible that similar inter-individual differences in receptor expression or downstream signalling pathways may contribute to the behavioural heterogeneity we observe. We now include a brief discussion of these possibilities in the main text (page 13, line 35–page 14, line 3):

“Divergent behaviour between Agg⁺ and Agg⁻ mice within the same estrous stage may reflect individual differences in hormone levels or receptor expression—variable *Esr1* expression in the MPOA for instance has been linked to parental performance in lactating females⁹.”

4. The MPOA contains diverse neural populations. What is the overlap between ER, PR, NPY receptor, and HCN channel expression in MPOA? This is important since the proposed mechanism assumes all these components exist in the same neurons. Such data could be extracted from published scRNA-seq datasets.

Thank you for this suggestion. We have now addressed this point by analysing a recently published brain-wide scRNA-seq dataset by Zhang et al., (2023, PMID 38092912). This analysis revealed substantial co-expression of the relevant genes within MPOA neurons (Referee Fig. 16):

- 55.0% of MPOA neurons co-express either *Npy1r* or *Npy2r* with either *Hcn1* or *Hcn2*
- 46.7% co-express either *Npy1r* or *Npy2r* with either *Hcn1* or *Hcn2*, and both *Esr1* and *Pgr*

These results are now included as Extended Data Fig. 11g, h and referenced in the revised Results section (page 9, lines 16–18) and Discussion (page 13, lines 32–34). The Methods section has also been updated accordingly (Methods > Quantification and data analysis > Co-expression analysis).

Referee Fig. 16. a, b, Co-expression of indicated transcripts in MPOA neurons (a), and percentage of co-expressing MPOA neurons depending on detection threshold (b). Based on data from Zhang et al., (2023, PMID 38092912) (see Methods).

5. Please define "aggressive state" on pages 9 and 11 and in Fig. 4. Does it refer to the entire period after aggression onset, the HMM state including aggression, or something else?

Indeed, 'aggressive state' refers to the entire period after aggression onset until the end of the pup assay. This clarification has now been added to the revised text (page 12, line 15) and the legend of Fig. 4.

6. The methods for miniscope analysis need more detail. Descriptions of HMM analysis and selectivity analysis are missing. Please make sure the methods section fully explains all analyses.

Many thanks for bringing this to our attention. A similar point was raised by referee 1 (point 19), in response to which we have added additional methodological details to the revised manuscript (Methods > Quantification and data analysis). This includes the following sections on HMM analysis and selectivity analysis:

Hidden Markov Model (HMM) analysis

To determine the optimal number of states for the Hidden Markov Model (HMM), we first explored a range of state values and assessed model performance using the Evidence Lower Bound (ELBO). ELBO, a standard metric in variational inference, measures model fit, with higher values indicating better performance. The optimal state number was identified by locating the turning point in the ELBO curve, beyond which additional states provided diminishing improvements. This optimal value (here: 5) was then used for all subsequent analyses. The HMM was implemented using the *ssm* package (<https://github.com/lindermanlab/ssm>), with a Gaussian observation model. This framework applies Bayesian learning and inference to state-space models and is well suited for analysing sequential neural data (Nair et al., 2023, PMID 36608653). The HMM was trained on neural recordings using the Expectation-Maximization (EM) algorithm for 50 iterations, with initial state assignments determined via *k*-means clustering. To assess the relationship between neural states and behaviour, we constructed a binary matrix for each behaviour, marking timestamps where the behaviour occurred as 1 and all other timestamps as 0. We then identified the state most frequently associated with each behaviour and computed the conditional probability of that behaviour occurring within the inferred state. This probability quantifies how strongly a given neural state is linked to specific behavioural patterns.

Selectivity analysis

To assess the selectivity of individual neurons for a particular stimulus, social and non-social stimuli were presented sequentially. Selectivity for pups versus other stimuli was quantified using a choice probability approach (Remedios et al., 2017, PMID 29052632; Karigo et al., 2021, PMID 33268894). For each neuron, fluorescence signals ($\Delta F/F_0$) recorded during pairs of chemosensory investigation behaviours (e.g., pup vs intruder investigation) were used to estimate how reliably the two behaviours could be distinguished based on their $\Delta F/F_0$ distributions. Specifically, $\Delta F/F_0$ values were extracted for each neuron's activity during behaviour α (e.g., attack) and behaviour β (e.g., sniff). These distributions were plotted as paired histograms and

cumulative distribution functions (CDFs). A Receiver Operating Characteristic (ROC) curve was then generated, with the CDF of pup sniffing on the x-axis. The selectivity index was computed as:

$$\text{Selectivity Index} = 1 - \text{AUC}_{\text{ROC}}$$

Where AUC_{ROC} is the area under the ROC curve. Neurons exclusively active during pup sniffing have a selectivity index of 1, those exclusively active during investigation of another stimulus an index of 0, and non-selective neurons an index of 0.5. To minimize the effects of gradual desensitisation, only the first chemosensory investigation episode for each stimulus was used.

7. The complex miniscope imaging data could benefit from more in-depth analysis to better understand aggression neural representation in MPOA and changes in pup neural representation under different hunger/estrous states.

Here are suggestions:

- Most neural data shown are either cropped around behavior onset or processed (e.g. PCA plot or HMM). Including example neural activity plots with behavior ethograms would help display diverse neural activity during behavior sessions. Activity heatmaps like Fig. 4h and Ex 15b with behavior ethograms (showing all pup interactions) above would be informative.

Thank you for this suggestion. We have now added full example recording sessions with behavioural annotations from two mice as (Referee Fig. 17, now Extended Data Fig. 15) and reference this figure in the revised text (page 12, line 13).

Referee Fig. 17. Enhanced MPOA neuronal responses to pup stimuli in Agg⁺ mice. Temporal profile of neuronal responses during interactions with pups and other targets. Full behavioural episodes from two Agg⁺ mice are shown.

- In Fig. 4d and Ex Fig. 13, the Z-scored neural activity shows a gradual increase, with larger differences after 2 sec potentially reflecting other behaviors. How were behavior bouts selected for averaged neural activity? Were bouts close to different behaviors excluded? If not, activity peaks may not correspond to the indicated behaviors. Please indicate mean bout durations in the Z-score plot and whether bouts were preselected.

Thank you for this helpful suggestion. The Z-scored activity traces and absolute tuning indices shown in Fig. 4d, e and Extended Data Fig. 17 (previously Extended Data Fig. 14) are all based on the *first behavioural bout* of the specified action per session. This approach minimises potential confounds from prior behaviour occurrences or accumulating social experience.

To further ensure that the extracted activity reflects the neural response to the behaviour of interest, we selected bouts in which no other overt behaviours occurred during the 2-s baseline

window (–2 to 0 s before onset). This was straightforward for isolated chemoinvestigation events (e.g., pup or intruder sniffing), but more challenging for pup grooming and aggressive contact, which are embedded in behavioural sequences and typically preceded by pup chemoinvestigation and -grooming, respectively. Only a very small number of episodes was available for these behaviours without any overt preceding chemoinvestigation or grooming, precluding meaningful analysis. We therefore did not exclude grooming- and aggressive contact-associated traces based on this criterion but—as outlined above—only assessed the first bout per session.

We now clarify this in the Methods section (Quantification and data analysis > Processing and analysis of *in vivo* imaging data > Evoked activity and absolute tuning index) as follows:

“For population-averaged neural activity and absolute tuning indices, we analysed the first behavioural bout of each specified action per session. To reduce potential confounds from prior behaviour occurrences or cumulative social experience, and to ensure that neural activity reflected the response to the behaviour of interest, we selected bouts in which no other overt behaviours occurred during the baseline period. This was straightforward for isolated chemoinvestigation events (e.g., pup or intruder sniffing), but more difficult for behaviours typically embedded in behavioural sequences, such as pup grooming and aggressive contact. These behaviours are often preceded by pup-directed sniffing or grooming, and only a very small number of episodes occurred in complete isolation. Grooming- and aggression-related traces were therefore not excluded based on baseline contamination but were still limited to the first bout per session to minimise experience-dependent effects.”

We have also calculated the mean duration of first bouts for each behaviour and now indicate these values directly in the Z-score plots of Fig. 4 and Extended Data Fig. 16, as requested (see Referee Fig. 18 below for an example). The average durations were: pup sniffing, 2.1 s; pup grooming, 4.6 s; pup-directed aggression, 2.3 s; male intruder chemoinvestigation, 8.2 s; female intruder chemoinvestigation: 3.6 s.

We reflect these variable durations in our tuning factor calculations, and now specify this in the Methods section (Quantification and data analysis > Processing and analysis of *in vivo* imaging data > Evoked activity and absolute tuning index):

“The baseline and activity windows used for Z-score and tuning index calculations were defined relative to the onset of each behavioural bout. The pre-event period (–X to 0 s) served as the baseline, and the post-event period (0 to +X s) as the activity window. The duration of these windows ($\pm X$) was tailored to each behaviour based on the average bout length: ± 2 s for pup sniffing and attacks, ± 4 s for pup grooming, ± 5 s for male intruder sniffing, and ± 3 s for female intruder sniffing. Tuning indices were calculated using these

behaviour-specific windows. For visualization, Z-scored responses were plotted over a – 5 to +5 s interval aligned to behavioural onset.”

Referee Fig. 18. Averaged, Z-scored responses of MPOA neurons during pup chemoinvestigation (114, 106 neurons from $n = 4$, 2 mice). Dashed line and grey bar indicate sniffing onset and mean bout duration (2.1 s), respectively. This panel corresponds to Fig. 4h in the manuscript.

What is the relationship between aggression-activated/inhibited neurons and their activity during other pup interactions like retrieval, grooming, and sniffing?

This is an important point. Since pup retrieval and pup-directed aggression never occurred within the same trial, we could not directly compare the responses of individual neurons to these behaviours within a given session. In principle, our longitudinal tracking of the same neurons before and after food deprivation could enable comparison of neural responses to pup retrieval (Pre) and pup-directed aggression (Post). However, only a single animal exhibited both behaviours across the two sessions, precluding any meaningful analysis.

We were, however, able to examine the relationship between aggression tuning (see our response to referee 1’s point 17 for a definition of tuning) and responses to pup sniffing and grooming, which commonly co-occurred with aggression across animals. We therefore assessed how each neuron’s tuning to pup-directed aggression (based on the first aggressive bout per session) relates to its tuning to pup sniffing and grooming, both before and after food deprivation. As shown in Referee Fig. 19, (now Extended Data Fig. 19) aggression tuning was positively correlated with pup grooming tuning before and after food deprivation (Referee Fig. 19b, d) but showed no consistent relationship with sniffing tuning (Referee Fig. 19a, c). This relationship was further illustrated by (1) heatmaps of neuronal responses to pup grooming and aggression highlighted in an individual animal (Referee Fig. 19e), and (2) example traces from neurons exhibiting consistent tuning to both behaviours (Referee Fig. 19f).

Together, these results highlight that aggression-tuned (i.e., -activated or -inhibited) MPOA neurons are also frequently tuned to pup grooming. We now mention this in the main text (page 13, lines 6–9).

Referee Fig. 19. a–d, Correlation between the tuning of MPOA neurons to pup-directed aggression and either pup sniffing (**a, c**) or grooming (**b, d**), assessed before (**a, b**) or after (**c, d**) food deprivation. Each dot represents a single neuron; neurons from the same animal are shown in matching colours. Statistics: linear mixed-effects model with mouse ID as a random effect (**a**, Pearson $r = -0.079$, $P = 0.204$; **b**, $r = 0.517$, $P < 0.001$; **c**, $r = 0.111$, $P = 0.077$; **d**, $r = 0.44$, $P < 0.001$). **e**, Representative heatmap of neuronal responses from an Agg⁺ animal displaying both pup grooming and pup-directed aggression (neurons sorted by hierarchical clustering; $n = 89$ neurons). **f**, Example traces from two MPOA neurons positively (top) or negatively (bottom) tuned to both pup grooming and pup-directed aggression.

Do neural tuning properties change pre- vs post-food deprivation?

Thank you for this suggestion. We have now performed this analysis and did not observe significant changes in MPOA neuron tuning to pup chemoinvestigation ($P = 0.649$) or grooming ($P = 0.655$) in Agg⁺ mice before versus after food deprivation (Referee Fig. 20, now Extended Data Fig. 16c, d).

However, as detailed in our response to referee 1 (point 16), the proportion of MPOA neurons tuned to pup sniffing is significantly reduced after food deprivation in Agg⁺ mice (Referee Fig. 9, now Extended Data Fig. 16a, b).

Together, these findings suggest that while detecting tuning changes with 1-photon calcium imaging and the tuning index metric is challenging, food deprivation nonetheless modulates stimulus-evoked responses in MPOA neurons (see also referee 1, point 16).

We now refer to these findings in the main text (page 12, lines 4–6).

Referee Fig. 20. Absolute tuning index of MPOA neurons during pup chemoinvestigation (**a**, 154 neurons from $n = 4$ mice) and pup grooming (**b**, 99 neurons from $n = 3$ mice) before and after food deprivation. Wilcoxon Signed-Rank test in **a**, **b**.

- Fig. 4m compares PC distances pre vs post for pup-directed behavior. While this quantifies neural representation changes, visualizing behavior locations in PC space (Pre-sniff, groom, Post-sniff, groom, attack, etc.) may better illustrate how representation shifts. Measuring distances between behaviors (for example, "pre-sniff vs post-aggression", "post-sniff vs post-aggression") would reveal overall representation changes.

Thank you for this suggestion. Following your recommendation, we visualised the locations of different behaviours in PC space for an example Agg⁺ animal (Referee Fig. 21a). We then quantified the pairwise distances between specific behaviours (e.g., 'pre-sniff' vs 'post-aggression', 'pre-groom' vs 'post-aggression') but found no significant differences in distances across behaviour pairs (one-way ANOVA, $P = 0.8003$; Referee Fig. 21b).

We believe this is because the neural representations of pup chemoinvestigation, grooming, and aggression are intrinsically distinct (e.g., see Fig. 4i). As a result, none of the behaviour pairs clustered more closely than others in PC space, limiting the interpretability of this approach for assessing representational shifts across states. Nonetheless, as demonstrated in Fig. 4 and Extended Data Fig. 17, the neural representation of pup-directed aggression remains distinct from that of pup-directed sniffing or grooming in MPOA neurons.

Referee Fig. 21. a, Example PC projection of MPOA neural activity from an Agg⁺ mouse during a full behavioural episode. **b**, PC distances between representations of indicated behaviour pairs in PC space (n = 5 mice).

- How well can pup retrieval be decoded from entire neural data or PC1/2 of Agg- mice? Since MPOA is well known for parenting behavior control, comparing decoding of parenting vs aggression would reveal how well MPOA activity explains each behavior and whether they occupy the same PC space.

We appreciate this suggestion. Unfortunately, our ability to directly address this question is limited by the behaviour of the recorded animals: only a single animal retrieved pups in the Pre and Post phase, respectively, and retrieval events were sparse—typically occurring two or fewer times per session. This low frequency results in insufficient data for a robust multiclass SVM analysis of retrieval behaviour.

That said, we would like to more clearly outline the rationale behind our decoding analysis. The primary goal was to evaluate whether neural activity in PC1/2 could recapitulate the classification performance of the full dataset. We found that pup-directed aggression was well-separated in PC space and highly decodable (as also shown in the confusion matrix in Extended Data Fig. 17d, previously Extended Data Fig. 14c), indicating that aggressive states are distinctly represented in MPOA neural dynamics, especially along PC2 (see Extended Data Fig. 17e).

In contrast, retrieval behaviour did not show a similarly distinct structure in PC space (see Fig. 4i), suggesting that although it may be decodable using the full high-dimensional dataset, it is not strongly represented in PC1/2. Thus, even if a decoding analysis were feasible, we would not expect the first two PCs to capture retrieval behaviour as clearly as they do aggression. This may

reflect different encoding strategies or degrees of neural coordination for parenting vs. aggression in the MPOA.

We now further emphasise this in the main text of the revised manuscript (page 12, line 34–page 13, line 2):

“[...] To assess how well this state could be identified from population activity, we trained a linear support vector machine (SVM) on the first two principal components, which successfully decoded aggression at a level comparable to an SVM trained on the full neural dataset (Fig. 4l), indicating that PC1 and PC2 capture a robust and low-dimensional signature of pup-directed aggression.”

- *What is the difference between Fig. 4f and Extended Fig. 14a?*

We apologise for this not being clear: Fig. 4f shows the first aggression episode while Extended Data Fig. 17a (previously Extended Data Fig. 14a) shows the average of all subsequent aggression episodes. We have now made this clearer in the legend of Fig. 4f (“The first aggression episode is shown.”), and the legend of Extended Data Fig. 17a indicates that the “average of aggression episodes 2–5” is shown.

Minor comments/errors:

1. *Methods, page 27, Hormone receptor knockout section: The description "AAV2/1-syn-fDIO-EGFP-2A-iCre (250 nl) or AAV2/1-syn-fDIO-GCaMP7s-2A-iCre (250 nl) was injected into the MPOA of Esr1-loxP or PR-loxP mice" appears incorrect as there is no Flpo.*

Many thanks for pointing this out. We have now corrected these mistakes in the ‘Hormone receptor knockout’ section. We have also standardised the formatting of Addgene numbers and viral titres in the Methods.

2. *Extended Fig. 4g-h: How can KO cell numbers be measured? AAV-GFP-Cre injection in Esr1 or PR loxP mice would show GFP in all infected cells regardless of Esr1/PR expression. The X-axis should indicate GFP+ cells rather than KO cells.*

Thank you - we have changed the x axis labels of Extended Data Fig. 5g, h (previously Extended Data Fig. 4g, h) to ‘GFP+ cells’ and changed the legend to “[...] number of GFP-labelled MPOA neurons in PR^{loxP} (or $Esr1^{loxP}$) mice injected with an AAV co-expressing GFP and Cre”.

3. *Fig. 2j: Please change "-- baseline" to "-- prediction" to match panel h.*

This has now been changed.

4. *Extended Fig. 8 e-i are not cited in the text. Consider citing around page 12, lines 8-11.*

Thank you for pointing this out—we have now cited these figure panels in the revised manuscript (page 14, line 19).

5. *Fig. 3C legend: Please add descriptions for ZD and NPY.*

This has now been added.

6. *Page 9, line 15: Suggested edit: "HCN channels via reducing cAMP through NPY receptor signaling."*

Thank you for this suggestion. We considered this but would prefer to keep the original statement ("food deprivation inhibits available HCN channels via NPY signalling") because our current study does not directly address the cellular pathways linking NPY receptor signalling with HCN channels. However, we discuss that cAMP very likely constitutes this link (page 14, lines 5–8):

"The specific NPY receptor subtypes and downstream signalling pathways remain to be identified, but approximately 57% of MPOA neurons express either NPY receptor Y1 or Y2, both of which inhibit HCN channels by reducing cyclic adenosine monophosphate (cAMP) levels through $G_{i/o}$ -protein-coupled mechanisms^{10–13}".

7. *Page 12, line 24: Reference 54 is cited twice.*

This has now been changed in the revised manuscript.

Referee #3 (Remarks to the Author):

The manuscript by Cao et al. from the Kohl lab, answers of very important question in the field of animal behavior in general and specifically of maternal behavior - how do animals integrate and prioritize different biological needs to produce behavior. The authors show with high temporal resolution that hunger increases pup aggression in virgin females mice, who normally are tolerant toward pups. They then use chemogenetic approaches to show that activation of AgRP+ neurons from the Arc nucleus is necessary and sufficient for this effect. Furthermore, they show that prolonged activation of AgRP+ neurons projecting to the MPOA are necessary for this behavior. Interestingly they also show that the estrus cycle induces significant variability in pup aggression induced by food deprivation. They elegantly show that the ratio of P4/E2 hormones is the determinant factor. In whole-cell slice recordings from MPOA the authors then show that baseline and evoked firing as well as the sag amplitude were decreased in aggressive females compared to non-aggressive ones following food deprivation. The % of silent MPOA neurons was increased in aggressive virgins. They then show that knocking down NPY, one of the hormones produced by AgRP+ cells rescues these electrophysiological changes in aggressive virgins, and delayed the onset of pup aggression events. Sag amplitude is a measure of HCN channels, that integrate cAMP signal with membrane potential hyperpolarization, and the authors show that the amplitude of sag as well as Hcn transcripts varies with the estrus cycle. Pharmacological blockade of HCN inhibitors turned virtually all virgin females aggressive. Lastly, miniscope recordings in behaving females show less activity in MPOA in aggressive females during pup sniffing. With their evidence and knowledge from prior work, the authors propose a model where hunger releases NPY in MPOA to reduce cAMP and suppress activity of HCN channels, which can result in significant reduction of neuronal firing and pup aggression behavior depending on P4/E2-dependent abundance of HCN channels. This is carefully designed and executed study, with an impressive number of control experiments. The findings are novel and exciting not only for understanding maternal care (and its failures) but more broadly for understanding the biological substrates by which a hierarchy of needs drives behavior.

We very much thank the referee for their enthusiasm and support for our work, and for the following, extremely helpful suggestions to strengthen this manuscript. We have now changed the text to address these points.

I have a few and relatively minor concerns:

1. I am not sure that the pattern of AgRP neuron activity is sufficient to conclude that: 'aggressive females do not perceive pups as food'. Perhaps it would be more informative to know if the virgins cannibalize the pups when hungry or just aggress them.

For ethical and legal reasons, we are required to immediately remove pups that are attacked by adults, and to euthanize pups that are injured as a result of being attacked. We could therefore not directly address whether virgins cannibalize pups, and our conclusion relies on the reported Arc^{AgRP} activity phenotype (Extended Data Fig. 1j–l). As we discuss, this notion is supported by the observation that pup and food representations in the MPOA differ after food deprivation (Extended Data Fig. 16p; previously Extended Data Fig. 13l), and that a similar state is present in (non-food-deprived) males during pup-directed aggression (Extended Data Fig. 18; previously Extended Data Fig. 15). We agree that more direct evidence would strengthen this conclusion. We have removed the statement ‘This suggests that aggressive females do not perceive pups as food’ from the results section and have modified the Discussion as follows (page 15, lines 10–16):

“Food-deprived mice are more likely to consume prey (Extended Data Fig. 1g), but do not appear to perceive pups as food, as pup interactions increase Arc^{AgRP} activity (Extended Data Fig. 1k–m), in contrast to the suppression of Arc^{AgRP} activity observed in response to food cues^{21,22}. While ethical and legal constraints prevent us from assessing whether Agg+ females cannibalize pups (see Methods), this interpretation is further supported by two observations: first, pup and food representations in the MPOA differ after food deprivation (Extended Data Fig. 16p); and second, a similar state occurs in males during pup-directed aggression (Extended Data Fig. 18).”

2. The amplitude of sag in Figure 3q, does not perfectly match the Hcn transcripts in Fig 3s, particularly for diestrus. Moreover, Hcn expression is high in estrus and low in diestrus, but in terms of behavior, these two estrus phases both have relatively low aggressive rates after food deprivation (below 50%, Fig. 2f). What could be the explanation?

Thank you - please refer to our above response to referee 1 (point 13).

3. For the model proposed to be validated, I think the authors need to show that NPY receptors and HCN are co-expressed in MPOA neurons. The authors do mention in the discussion that ~70% of MPOA neurons express NPY receptors, so presumably the co-expression likelihood is high.

Thank you for this suggestion —please refer to our above response to referee 2 (point 4). Briefly, analysis of an existing scRNA-Seq dataset indicates that ~55% of MPOA neurons co-express *Npy1r* or *Npy2r* with either *Hcn1* or *Hcn2* (see Referee Fig. 16, now Extended Data Fig. 11g, h).

4. In Figure 4, the estrus cycle seems to be abandoned – is there less aggressive state in estrus?

We regret not including information about the estrous stage of the recorded animals in the original submission. Of the six recorded mice, one was in proestrus, two in estrus, two in metestrus, and one in diestrus. Notably, both mice in estrus were classified as Agg⁻, while all others were Agg⁺.

This pattern is consistent with our proposed model (see Fig. 3w) in which pup-directed aggression arises from an interaction between estrous stage and food deprivation. Specifically, a low P4/E2 ratio, characteristic of estrus, appears to bias animals toward an Agg⁻ phenotype, whereas a higher ratio, as seen in metestrus, promotes an Agg⁺ phenotype.

We have now added this information to the legend of Fig. 4 and referenced it in the main text (page 10, lines 28–31):

“Among the six recorded females, one was in proestrus, two in estrus, two in metestrus, and one in diestrus. Both mice in estrus were non-aggressive (Agg⁻), whereas all others were Agg⁺, consistent with our model (Fig. 3w) in which estrous stage interacts with food deprivation to shape behavioural outcomes. This pattern suggests that the low P4/E2 ratio characteristic of estrus biases animals toward an Agg⁻ phenotype, while the higher ratio during metestrus promotes pup-directed aggression.”

Some comments on the Methods:

5. why is ‘frantic carrying of pups’ considered aggressive behavior? Wouldn’t it be explained by higher anxiety?

Aggressive pup carrying is a characteristic element of pup-directed aggression in virgin male and female mice as described in Isogai et al. (2018, PMID 30550786) and Autry et al. (2021, PMID 34423776). Our behavioural scoring follows the criteria established in these studies. This aggressive carrying, often accompanied by behaviours such as biting, is illustrated in Video S2 of Isogai et al. and Supplementary Video 4 of Autry et al. To align with this terminology, we have replaced ‘frantic’ with ‘aggressive’ in the Methods section (Behavioural profiling > Pup-directed behaviour assay) and updated the references accordingly.

6. the authors state that aggressive animals targeted all pups in cage (2), despite also stating that the assay was terminated immediately upon observing pup-directed aggression. It seems it can’t be both.

We apologise for this not being clearer in our original manuscript. If a pup was attacked in our behavioural experiments, all pups were immediately removed and the trial terminated. During *in vivo* imaging experiments, attacked pups were promptly replaced with new pups to allow the observation of multiple aggression episodes. In the rare event of injury, affected pups were

immediately euthanized. This section has been revised accordingly in the updated manuscript (Methods > Behavioural profiling > Pup-directed behaviour assay).